# CURRICULUM REINFORCEMENT LEARNING FOR QUANTUM ARCHITECTURE SEARCH UNDER HARDWARE ERRORS

**Yash J. Patel**[‡,1,6], **Akash Kundu**[‡,2,3], **Mateusz Ostaszewski**[4], **Xavier Bonet-Monroig**[1,5], **Vedran Dunjko**[1,6], **and Onur Danaci**[1,5]

[1]$\langle aQa^L \rangle$ Applied Quantum Algorithms, Leiden University
[2]Institute of Theoretical and Applied Informatics, Polish Academy of Sciences
[3]Joint Doctoral School, Silesian University of Technology
[4]Warsaw University of Technology, Institute of Computer Science
[5]Lorentz Institute, Leiden University
[6]LIACS, Leiden University
[‡]Equal contribution

## ABSTRACT

The key challenge in the noisy intermediate-scale quantum era is finding useful circuits compatible with current device limitations. Variational quantum algorithms (VQAs) offer a potential solution by fixing the circuit architecture and optimizing individual gate parameters in an external loop. However, parameter optimization can become intractable, and the overall performance of the algorithm depends heavily on the initially chosen circuit architecture. Several quantum architecture search (QAS) algorithms have been developed to design useful circuit architectures automatically. In the case of parameter optimization alone, noise effects have been observed to dramatically influence the performance of the optimizer and final outcomes, which is a key line of study. However, the effects of noise on the architecture search, which could be just as critical, are poorly understood. This work addresses this gap by introducing a curriculum-based reinforcement learning QAS (CRLQAS) algorithm designed to tackle challenges in realistic VQA deployment. The algorithm incorporates (i) a 3D architecture encoding and restrictions on environment dynamics to explore the search space of possible circuits efficiently, (ii) an episode halting scheme to steer the agent to find shorter circuits, and (iii) a novel variant of simultaneous perturbation stochastic approximation as an optimizer for faster convergence. To facilitate studies, we developed an optimized simulator for our algorithm, significantly improving computational efficiency in simulating noisy quantum circuits by employing the Pauli-transfer matrix formalism in the Pauli-Liouville basis. Numerical experiments focusing on quantum chemistry tasks demonstrate that CRLQAS outperforms existing QAS algorithms across several metrics in both noiseless and noisy environments.

## 1 INTRODUCTION

The past decade has witnessed dramatic progress in the study and development of quantum processing units, prompting extensive exploration of the capabilities of Noisy Intermediate-Scale Quantum (NISQ) hardware (Preskill, 2018). To account for the stringent limitations of NISQ devices, variational quantum algorithms (VQAs) (Peruzzo et al., 2014; Farhi et al., 2014; McClean et al., 2016; Cerezo et al., 2021) were developed as a suitable way to exploit them.

In essence, VQAs consist of three building blocks: a parameterized quantum circuit (PQC) or ansatz, a quantum observable allowing the definition of a cost function, and a classical optimization routine that tunes the parameters of the PQC to minimize the cost function. Each of the building blocks corresponds to an active area of research to understand the capabilities of VQAs.

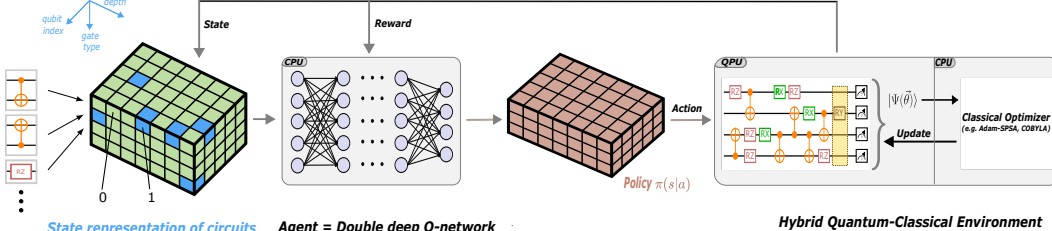

Figure 1: Illustration of the architecture of the double deep-Q network utilized by the reinforcement learning (RL) agent. The RL state $s$ here describes the quantum circuit encoded as a tensor-based 3D grid whose axes correspond to the qubit index, depth (moment) and gate type. This information is processed through a multi-layer perceptron. For the output, the agent computes the policy, according to which the actions $a$ (as gates) in state $s$ are probabilistically chosen. A classical optimizer optimizes the circuit and, upon completion, provides a reward that guides the agent to select subsequent actions.

One such promising VQA with an application in quantum chemistry is variational quantum eigensolver (VQE). In VQE, the objective is to find the ground state energy of a chemical Hamiltonian $H$ by minimizing the energy

$$E(\vec{\theta}) = \min_{\vec{\theta}} \left( \langle \psi(\vec{\theta})|H|\psi(\vec{\theta}) \rangle \right). \tag{1}$$

The trial state $|\psi(\vec{\theta})\rangle$ is prepared by applying a PQC $U(\vec{\theta})$ to the initial state $|\psi_{\text{initial}}\rangle$, where $\vec{\theta}$ specify the rotation angles of the local unitary operators in the circuit. The structure of this circuit significantly impacts the success of the algorithm. Traditionally, in VQAs, the structure of the PQC is fixed before initiating the algorithm and is often motivated by physical (Peruzzo et al., 2014) or hardware (Kandala et al., 2017) considerations. However, fixing the structure of the PQC within VQA may impose a substantial limitation on exploring the relevant parts of the Hilbert space. To circumvent such limitations, recent attention has turned towards automatically constructing PQC through quantum architecture search (QAS) (Grimsley et al., 2019; Tang et al., 2021; Anastasiou et al., 2022; Zhang et al., 2022). This approach removes the necessity for domain-specific knowledge and often yields superior PQCs tailored for specific VQAs. Given a finite pool of quantum gates, the objective of QAS is to find an optimal arrangement of quantum gates to construct a PQC (and its corresponding unitary $U(\vec{\theta}_{opt})$, where $\vec{\theta}_{opt}$ are optimal parameters) which minimizes the cost function (see Eq. 1).

One such proposal to tackle the QAS problem is to employ reinforcement learning (RL) (Ostaszewski et al., 2021b; Kuo et al., 2021), where the PQCs are defined as a sequence of actions generated by a trainable policy. The value of the quantum cost function (optimized independently via a classical optimizer) serves as an intermittent signal for the final reward function. This reward function then guides policy updates to maximize expected returns and select optimal actions for subsequent steps.

Up to the completion of this work, most algorithms for QAS have been formulated under the assumption of an idealized quantum computer, free from physical noise and endowed with all-to-all qubit connectivity. However, it is essential to recognize that the QAS problem becomes even more daunting in light of the constraints imposed by current NISQ devices (Du et al., 2022). According to (Wang et al., 2021), the impact of noise on the trainability of VQAs in the context of QAS is multifaceted.

First and foremost, noise induces the phenomenon of barren plateaus, causing gradients within the cost function landscapes to vanish, thereby impeding optimization. Moreover, quantum noise introduces additional complexity by transforming certain instances of exponentially occurring global minima into local optima, posing a substantial challenge to trainability (Fontana et al., 2022). Additionally, noise alters the cost landscape to an extent that the optimal parameters for the noisy landscape may no longer align with the optimal parameters for the noiseless landscape (Sharma et al., 2020). In addition to the influence of physical noise on the landscape, the finite sampling noise hinders the performance of classical optimizers working on that landscape, raising the necessity of optimizers robust to such conditions (Bonet-Monroig et al., 2023). Overall, the adverse effects in-

duced by the quantum noise demand swift simulation of numerous queries from noisy landscapes and robust classical optimization strategies operating under these conditions.

Performing experiments that assess the performance of QAS algorithms in the presence of noise, though computationally demanding, is a critical step toward understanding and ultimately overcoming the challenges presented by real-world quantum hardware.

The contribution of our work is two-fold. Firstly, we introduce the curriculum-based reinforcement learning QAS (CRLQAS) method depicted in Fig. 1. Secondly, we provide an optimized machinery for CRLQAS that effectively simulates realistic noise—a crucial element for testing and enhancing our method for quantum chemistry problems on larger system sizes. The machinery uses offline computation of Pauli-transfer matrices (PTM) and GPU based JAX framework to accelerate computations by up to six-fold. To further improve the learning process of CRLQAS, we introduce several key components:

(1) A mechanism, namely, illegal actions preventing invalid sequences of gates.
(2) A random halting procedure to steer the agent to learn gate-efficient circuits.
(3) A tensor-based binary circuit encoding that captures the essence of the depth of the PQC and enables all-to-all qubit connectivity.
(4) Two variants of simultaneous perturbation stochastic approximation (SPSA) algorithm that use adaptive momentum (Adam) and a variable measurement sample budget for faster convergence rate and robustness.

By leveraging strategies (1)–(4), we achieve chemical accuracy for a class of chemical Hamiltonians with better accuracy by maintaining both gate and depth efficiency in noisy and noiseless scenarios. Our numerical demonstrations often establish CRLQAS as the superior approach compared to existing QAS methods within the context of VQE across a spectrum of metrics.

## 2 RELATED WORK

There is a growing body of research in the field of QAS aimed at enhancing the efficiency and performance of quantum algorithms. Several key themes emerge from related works, encompassing QAS, optimization strategies, and the application of RL to quantum computing.

**Evolutionary and Genetic algorithms** Existing literature has explored various strategies to automate the design of quantum circuits. Evolutionary algorithms (EAs), particularly genetic algorithms (GAs), have been utilized to evolve quantum circuits (Williams & Gray, 1998). While GAs demonstrate the capability to evolve simple quantum circuits, such as quantum error correction codes or quantum adders (Bang & Yoo, 2014; Potoček et al., 2018; Chivilikhin et al., 2020; Bilkis et al., 2021), they face limitations in handling parameterized rotation gates. They are also known to be sensitive to gene length and candidate gate set size.

**Sampling-based algorithms** As a step forward, sampling-based learning algorithms have been introduced to sample circuits from candidate sets, addressing the QAS problem for ground state energy estimation (Grimsley et al., 2019; Tang et al., 2021).In (Zhang et al., 2022), the authors utilize Monte Carlo sampling to search for PQCs on QFT and Max-Cut problems. In (Du et al., 2022), a QCAS method based on supernet and weight sharing strategy was introduced to better estimate energy for quantum chemistry tasks. However, in the presence of hardware noise, this method fails to estimate the ground state energy within a precision to make realistic chemistry predictions. Meanwhile, (Wu et al., 2023) uses the Gumbel-Softmax technique (Gumbel, 1948) to sample quantum circuits and benchmark their algorithm on VQAs, including quantum chemistry tasks.

**Reinforcement-learning-based algorithms** RL techniques have also been applied to tackle the QAS problem for VQAs. Typically, such approaches employ a carefully designed reward function to train the agent to choose suitable gates. (Ostaszewski et al., 2021a) employed double deep Q-network (DDQN) to estimate the ground state of the 4- and 6-qubit LiH molecules. RL has also been used to address the qubit routing problem (Li et al., 2019; Sinha et al., 2022; Pozzi et al., 2022). These works aim to minimize circuit depth through qubit routing using RL techniques. Additionally, (Kuo et al., 2021) have employed RL to test their algorithms on 2-qubit Bell state and 3-qubit GHZ state in the presence of hardware noise. We summarize the relevant QAS methods for the ground state energy estimation task in Appendix K.

# 3   Curriculum Reinforcement Learning Algorithm

We give an overview of the CRLQAS algorithm to construct PQCs for a VQA task, wherein we present state and action representations and the reward function used in this work. Later, we also describe the features of CRLQAS, which yields better performance across several metrics.

In the CRLQAS environment, the agent starts every episode with an empty circuit. It then sequentially appends the gates to the intermediate circuit until the maximum number of actions has been reached. At every time step of the episode, the state corresponds to a circuit and an action to append a gate to that circuit. As we employ deep RL methods, we encode states and actions in a way that is amenable to neural networks. Thus, each state is represented by a newly introduced *tensor-based binary encoding* of the gate structure of the PQC, which we describe in Sec. 3.3. For our simulations to construct the circuits, we consider the gate set consisting of Controlled-NOT (`CNOT`) and 1-qubit parameterized rotation gates (`RX`, `RY`, `RZ`). We consciously omit the use of the continuous parameters of rotation gates and only utilize the estimated energy of the circuit in the state representation of the RL agent.

To encode the actions, we use a one-hot encoding scheme as in (Ostaszewski et al., 2021b) with `CNOT` and rotation gates represented by two integers. For `CNOT`, these values specify the positions of control and target qubits, respectively. In contrast, for the rotation gates, the first value specifies the qubit register, and the second indicates the rotation axis (the enumeration of the position starts from 0). For an $N$-qubit quantum state, the total number of actions is $3N + 2\binom{N}{2}$, where the first term comes from the choice of selecting the rotation gates and the latter from choosing `CNOT`s.

To steer the agent towards the target, we use the same reward function $R$ at every time step $t$ of an episode, as in (Ostaszewski et al., 2021b). The reward function $R$ defined as,

$$R = \begin{cases} 5 & \text{if } C_t < \xi, \\ -5 & \text{if } t \geq T_s^e \text{ and } C_t \geq \xi, \\ \max\left(\frac{C_{t-1} - C_t}{C_{t-1} - C_{\min}}, -1\right) & \text{otherwise} \end{cases} \tag{2}$$

where $C_t$ refers to the value of the cost function $C$ at each step, $\xi$ is a user-defined threshold value and $T_s^e$ denotes the total number of steps $s$ allowed for an episode $e$. $T_s^e$ can also be understood as the maximum number of actions allowed per episode. Note that the extreme reward values ($\pm 5$) signal the end of an episode, leading to two stopping conditions: exceeding the threshold $\xi$ or reaching the maximum number of actions. For quantum chemistry tasks, the threshold $\xi$ is typically set to $1.6 \times 10^{-3}$ as it defines the precision such that realistic chemical predictions can be made. The goal of the agent is to obtain an estimated value of $C_{\min}$ within such precision.

The continuous parameters $\vec{\theta}$ of the quantum circuit that describes the cost function $C$ are optimized separately using a classical optimizer to obtain the reward $R$. The choice of the classical optimizer is critical for the RL agent's success in the environment. The nature of the CRLQAS reward function depends on the cost function evaluation. The cost function evaluation can be deterministic or stochastic based on both the classical optimizer and quantum noise. Stochastic optimizers, like SPSA, tend to converge toward different parameters, resulting in different function values across multiple trials, even when initialized identically. Moreover, the quantum noise may also lead to different function values even with the same parameters.

We consider both environment types (details discussed in Sec. 4) and successfully validate the effectiveness of novel features introduced in this work for the CRLQAS method. The results of our ablation study to identify the features that standout in CRLQAS can be found in Appendix E. In the next section, we describe the features of this method. We adopt the "feedback-driven curriculum learning" approach from the (Ostaszewski et al., 2021b) which is elaborated in the Appendix B.1.

## 3.1   Illegal actions for the RL agent

As QAS is a hard combinatorial problem with a large search space, pruning the search space is beneficial for the agent to find circuits with different structures. Hence, we introduce a simple mechanism, namely, *illegal actions* to narrow down the search space significantly. The mechanism uses the property that quantum gates are unitary, and hence, when two similar gates act on the same qubit(s), they cancel out. An elaborate discussion on the implementation of the illegal actions mechanism is provided in Appendix B.2.

## 3.2 RANDOM HALTING OF THE RL ENVIRONMENT

In the CRLQAS algorithm, the total number of actions executed by the agent within an episode, denoted as $T_s^e$, is set using multiple hyperparameters. The hyperparameter, $n_{\text{act}}$, determines an upper limit on the total actions in an episode, meaning $T_s^e \leq n_{\text{act}}$. If the agent is not interrupted by any of the schemes mentioned in the paper, it selects a maximum of $T_s^e = n_{\text{act}}$ actions (gates).

If the RL agent finds a circuit that estimates an energy value (upon optimizing parameters) lower than this threshold, the episode is halted abruptly leading to $T_s^e < n_{\text{act}}$. When employing the *random halting* (RH) scheme, both the curricula and a stochastic sampling procedure then influence the episode length. We use samples from the negative binomial distribution to determine the cap on the total number of actions per episode, $T_s^e$, at the beginning of each episode (Dougherty, 1990). The probability mass function of this distribution is

$$\Pr\left(X = n_{\text{fail}}\right) = \binom{n_{\text{act}} - 1}{n_{\text{fail}}} p^{n_{\text{fail}}} (1 - p)^{n_{\text{act}} - n_{\text{fail}}}, \tag{3}$$

where $n_{\text{act}}$ represents the hyperparameter for the total number of allowed actions per episode, and in this context, it is the total number of Bernoulli trials as well. Also, $n_{\text{fail}}$ denotes the number of failed Bernoulli trials, and the $p$ denotes the probability of a Bernoulli trial to fail, which we provide as another hyperparameter. The probability mass function given above yields the probability of having $n_{\text{fail}}$ failed Bernoulli trials given $n_{\text{act}}$ total trials and $p$ failure probability. In practice, we sample $T_s^e \sim n_{\text{fail}}$ as a random number based on the failure probability, and the total number of experiments is determined via inverse transform sampling implemented in NumPy (Harris et al., 2020). This inverse sampling generates a number within the range $T_s^e \sim [0, n_{\text{act}}]$, and we obtain this number at the outset of each episode.

The primary motivation for integrating RH into CRLQAS is to enable the agent to adapt to shorter-length episodes, thereby facilitating the agent's ability to discover more compact circuits in early successful episodes, even if it occasionally leads to a delay in achieving the first successful episode.

## 3.3 TENSOR-BASED BINARY CIRCUIT ENCODING

Several QAS algorithms often require the compact representation of the circuit, also commonly known as encoding, as it allows for modification, comparison, and exploration of quantum circuits. Hence, the choice of encoding plays a vital role in efficiently maneuvering the search space and uncovering efficient and novel quantum circuits.

We provide the agent with a complete description of the circuit by employing a binary encoding scheme that captures the structural aspects of the PQC, specifically, the ordering of the gates. To keep the dimension of the input tensor constant throughout an episode, the tensor must be prepared for the deepest quantum circuit that can be generated.

For constructing a tensor representation of the circuit, we initially specify the hyperparameter $n_{\text{act}}$, which determines the maximum number of allowed actions (i.e., gates) in all episodes. We now define the moment of a PQC, which is crucial for understanding the tensor representation. The moment or layer of a quantum circuit represents all gates that can execute simultaneously; essentially, it is a set of operations acting on different qubits in parallel. The number of these moments determines the depth of the circuit. We represent PQCs with 3D tensors such that each matrix (2D "slice") represents a different moment of the circuit, and the other dimension represents the depth (see Fig. 1). We use the maximum number of actions parameter, $n_{\text{act}}$, at a given episode as an upper bound on the depth of the circuit. We give this upper bound for the extreme case where all the actions are implemented as 1-qubit gates appended to the same qubit. As a result, at each episode, we initialize an empty circuit of depth $n_{\text{act}}$ by defining a $[n_{\text{act}} \times ((N + 3) \times N)]$ dimensional tensor of all zeros. Here $N$ is the number of qubits. Each moment or layer in the circuit is depicted through a matrix of dimensions $((N + 3) \times N)$. In this matrix, the initial $N$ rows showcase the locations of the control and target qubits for the CNOT gates applied during that specific moment. Following these, the subsequent three rows indicate the positions of 1-qubit rotation gates RX, RY, and RZ, respectively. The visualization of such an encoding for a toy circuit is depicted in Appendix B.3.

## 3.4 ADAM-SPSA ALGORITHM WITH VARYING SAMPLES

In the realm of VQE within a limited measurement sample budget, several variants of simultaneous perturbation stochastic approximation (SPSA) have displayed robustness towards finite sample

(shot) noise (Cade et al., 2020; Bonet-Monroig et al., 2023). One such family of variants, multi-stage SPSA, reset the decaying parameters while tuning the allowed measurement sample (shot) budget between stages. Implementing a moment adaptation subroutine in classical ML, such as Adam (Kingma & Ba, 2014), alongside standard gradient descent, increases robustness and convergence rates. This combination has also shown promise in the domain of quantum optimal control for pulse optimization (Leng et al., 2019).

We investigate multi-stage variants of such an Adam-SPSA algorithm, exploring different shot budgets and continuous versus reset of decay parameters (after every stage). In doing so, we empirically observed increased robustness and faster convergence rates of a 3-stage Adam-SPSA with continuously decaying parameters. We note that this particular observation is novel and was not discovered before to the best of our knowledge. In Appendix F, we present empirical results demonstrating the convergence of this Adam-SPSA variant across VQE tasks involving various qubit numbers and shot budgets. Leveraging these enhancements, we managed to halve the number of function evaluations, thereby significantly reducing the evaluation time for RL training under physical noise.

### 3.5 FAST GPU SIMULATION OF NOISY ENVIRONMENTS

Most QAS algorithms require a considerable amount of noisy function evaluations unless a training-free algorithm is used. However, computing these evaluations becomes challenging within modern-day simulation frameworks, particularly due to the method of noise simulation known as the Kraus operator sum formalism. This formalism hinders the use of fast simulation methods tailored for noiseless scenarios. Consequently, with an increase in qubits and gates, simulations are hindered not only by the curse of dimensionality associated with dense matrix multiplications but also by the exponential increase in the number of noise channels and their corresponding Kraus operators.

Most importantly, this computational overhead needs to be paid online (during training and parameter optimization within episodes) at each step (see Appendix H). To mitigate this, the Pauli-transfer matrix (PTM) formalism is applied, allowing the fusion of noise channels with their respective gates to be precomputed offline, eliminating the need for recompilation at each step. In conjunction with PTMs, we employ GPU computation coupled with just-in-time (JIT) compiled functions in JAX (Bradbury et al., 2018), yielding up to a six-fold improvement in RL training while simulating noisy quantum circuits (see Appendix I).

## 4 EXPERIMENTS

Our objective is to assess the performance of CRLQAS described in Sec. 3 to find circuits that are accurate and can overcome hardware noise. We benchmark CRLQAS for the task of finding ground-state energies via variational quantum eigensolvers (VQE) of three molecules, Hydrogen ($H_2$), Lithium Hydride (LiH) and Water ($H_2O$). For all three molecules, we use their representation in the minimal STO-3G basis, mapped to qubits through Jordan-Wigner and Parity transformations (Ostaszewski et al., 2021b; Wu et al., 2023). To simplify the computational task, we use symmetries of the problem to taper off qubits, thus calculating the ground-state energies of 2-qubit $H_2$ ($H_2 - 2$), 3-qubit $H_2$ ($H_2 - 3$), and 4-qubit $H_2$ ($H_2 - 4$), 4-qubit LiH (LiH $- 4$) and 6-qubit LiH (LiH $- 6$) and 8-qubit $H_2O$ ($H_2O - 8$) (see Appendix L for description of molecules). Additionally, we use hardware noise profiles from publicly available resources IBM (Corporation, 2023) to implement noisy models of varying complexity and run experiments on the first three qubit systems. The relevant details about the implementation of the CRLQAS method and its hyperparameters are outlined in Appendix C.

In the subsequent subsection outlining noisy simulations, the RL agent consistently receives signals as noisy energies during training, guiding its action choices. However, while post-processing the data of the trained agent, we only assess energies in a noiseless scenario to determine the success or failure of an episode. An ablation study to identify the features that standout within the CRLQAS method in both noiseless and noisy settings can be found in Table 3 of Appendix E. Our analysis reveals that incorporating illegal actions without random halting prompts the agent to achieve a positive signal (a successful episode) early on, albeit resulting in larger circuits. Conversely, introducing random halting encourages the agent to discover shorter circuits, there is a trade-off as the agent receives the positive signal at a later stage.

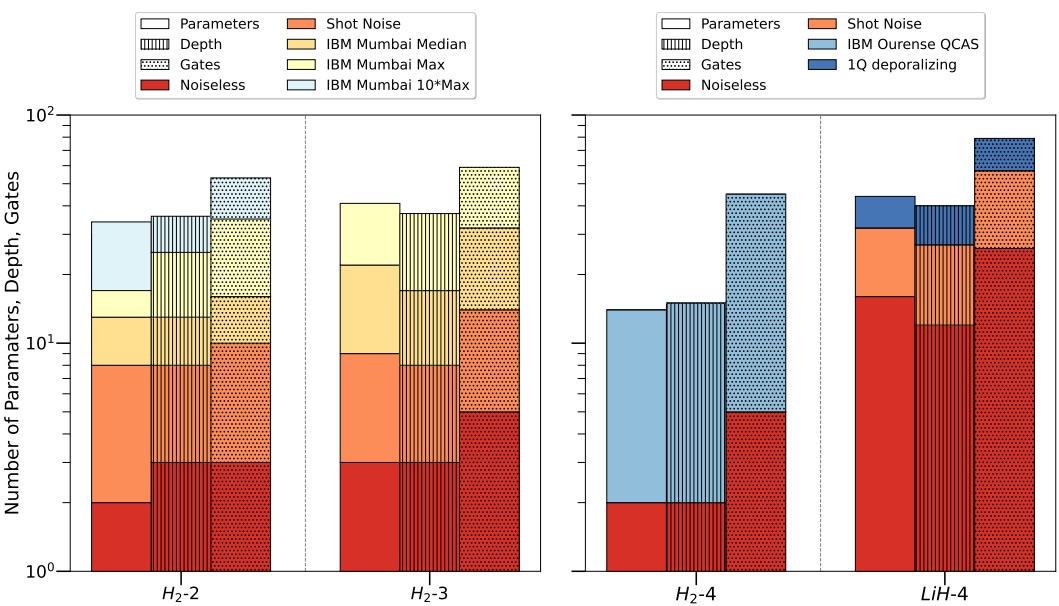

Figure 2: **Achieving the chemical accuracy for** $H_2$ **(with** 2**-,** 3**- and** 4**-qubits), and** LiH **(with** 4**-qubits) molecules via a systematic study under realistic physical noise where the noise model mimics the** `IBM Quantum` **devices.** In the initial episodes, the probability of choosing random actions is very high, and to avoid this, we consider the statistics from 2000 episodes onward and plot the median of the minimum over 3 different seeds. The different colours denote the different levels of noise (increasing from bottom to top), and the patterns (from left to right) denote the number of parameters, the depth and the number of gates, respectively. We reach the chemical accuracy for $H_2 - 2$ and $H_2 - 3$ (except the 10 times max noise of `IBM Mumbai` device) molecule for all levels of noise. Meanwhile, $H_2 - 4$ molecule reaches the chemical accuracy with the noise profile of `IBM Ourense` even with qubit-connectivity constraints. Finally, with LiH $- 4$, we achieve the chemical accuracy with shot and 1-qubit depolarizing noise. Note that, for $H_2$, we decreased the threshold (usually set to chemical accuracy) to $2.2 \times 10^{-4}$ because the problem is straightforward to solve.

## 4.1 NOISY SIMULATION

To simulate molecules, we consider a realistic noisy setting without drift, employing the noise profile of the `IBM Mumbai` (see Appendix H ) and `IBM Ourense` (Du et al., 2022) quantum devices. Despite not considering drift time-scales of quantum computers, our experiments on a real quantum device (see Appendix J) corroborate a hypothesis previously noted in (Rehm et al., 2023), that classical simulations of noisy quantum circuit evaluations closely resemble actual hardware behavior, particularly under low noise levels. When noise is present, estimated cost function values for given parameters differ from those in noiseless scenarios. This discrepancy challenges leveraging prior domain knowledge (like ground state energies) for configuring rewards and curriculum mechanisms in RL training. Notably, our CRLQAS algorithm does not rely on prior knowledge of the true ground state energy value. Instead, it employs a curriculum-based learning method that dynamically adapts to the problem difficulty based on the performance of the agent. This approach makes the agent self-sufficient and allows it to accumulate knowledge at a pace determined by its performance.

We first simulate all the molecules in the noiseless scenario and then in the presence of shot noise. Subsequently, we incorporate the noise profile from the `IBM Mumbai` device, setting the 1-, and 2-qubit depolarizing noise to (i) the median, (ii) the maximum value, and (iii) 10 times the maximum noise value. Our findings, illustrated in Fig. 2, demonstrate the impact of noise levels on the quantum circuit statistics (like depth, number of gates, and parameter count) to solve the ground state energy estimation problem via VQE for $H_2 - 2$, $H_2 - 3$, $H_2 - 4$ and LiH $- 4$. Our results empirically verify a commonly expressed hypothesis: an increase in noise levels corresponds to an increase in the number of gates and parameters in the circuit (Sharma et al., 2020; Fontana et al., 2021).

Moreover, we conduct a comparative analysis between CRLQAS and the QCAS method (Du et al., 2022). With equivalent noise settings and connectivity constraints of `IBM Ourense`, our findings indicate that CRLQAS efficiently determines the ground state energy of $H_2 - 4$, yielding $-1.136$Ha,

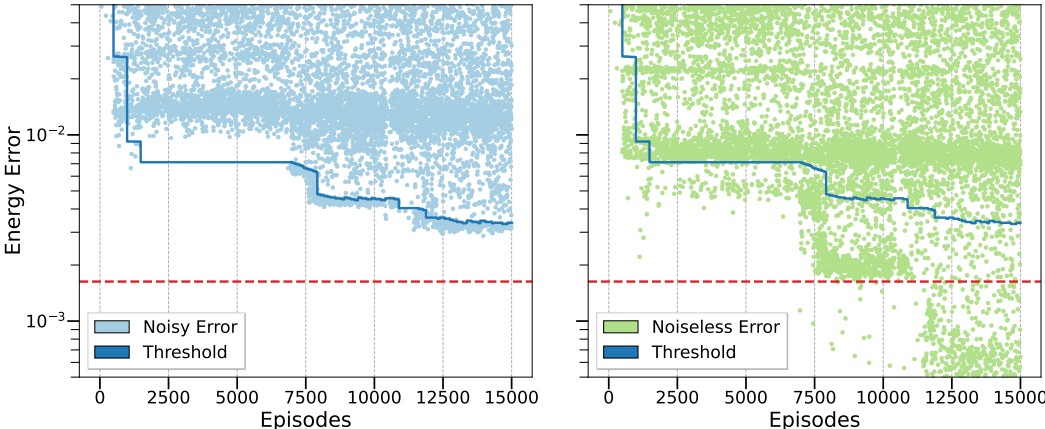

Figure 3: **Learning curve of the** LiH **(with 4-qubits) experiment.** The agent is trained with a noise model with 1-qubit depolarizing noise of strength $0.1 \times 10^{-2}$, and sampling of the expectation values of $10^6$ repetitions. The left panel shows the training curve under noise, the right panel is the evaluation of the points on the left panel but without noise. The red dashed line indicates the chemical accuracy.

in contrast to the reported minimum energy of $-0.963$Ha in (Du et al., 2022). Detailed data in Table 2 outlines the minimum energy and the number of gates for $H_2-4$. We also perform a noiseless simulation of the circuit presented in (Du et al., 2022), yielding an energy error of $1.9 \times 10^{-2}$ (with 16 gates). In contrast, CRLQAS achieves significantly superior energy errors of $8 \times 10^{-5}$ (with RH, 32 gates) and $1.5 \times 10^{-5}$ (without RH, 40 gates), demonstrating improvements by three orders of magnitude. Upon closer inspection of all the successful episodes during post-processing of the trained agent's data (i.e., searching for an intermediate step where the energy error is just below chemical accuracy) for $H_2 - 4$, we observed that CRLQAS indeed achieves an energy error of $2.8 \times 10^{-4}$ (with RH) and $5.5 \times 10^{-4}$ (without RH) respectively, while utilizing only 10 gates.

In Fig. 3, we present the learning curves of the agent from our simulations for $LiH - 4$ with 1-qubit depolarizing noise strength of $0.1 \times 10^{-2}$ and $10^6$ sampling noise. The illustration tracks two crucial values using the optimal parameters discovered in the noisy setting: the noisy and noiseless energies, obtained by evaluating their respective cost functions for those parameters. The left panel depicts that in the presence of noise, the energy error closely aligns with the threshold curve and decreases with it. This trend indicates the learning trajectory of the agent, where it learns to construct circuits that minimize the noisy energy error. Conversely, in the right panel, the energy behaviour does not mimic the threshold curve for the noiseless scenario. Notably, it even passes chemical accuracy despite the threshold being well above it. This divergence suggests that minimizing the noisy energy does not necessarily result in minimizing the noiseless energy, and vice versa.

We also trained RL agents to solve the ground state energy estimation problem via VQE for $LiH - 4$ in two additional noisy settings. In the first setting, we employ 1-, and 2-qubit depolarizing channels of strength $0.1 \times 10^{-2}$ and $0.5 \times 10^{-2}$, respectively. In the second, we utilized the median noise profile of IBM Mumbai device. Unfortunately, the agent fails to achieve chemical accuracy in these noisy settings. In the first setting, the agent attained a minimum noiseless error of $3.4 \times 10^{-3}$ (trained for only 5000 episodes). In the latter scenario, it reached $3.3 \times 10^{-3}$ (trained for 15000 episodes).

## 4.2 Noiseless simulation

We now present the analyses for noiseless settings. This is due to the considerable challenge of scaling noisy QAS beyond four qubits within the scope of this work. The exponential increase in computational cost with the number of qubits makes it exceptionally difficult to handle billions of queries to the noisy cost functions.

To analyze the scaling performance of CRLQAS method, we assess its performance for $H_2 - 4$, $LiH - 4$, $LiH - 6$ and $H_2O - 8$ molecules in comparison to quantumDARTS (Wu et al., 2023) and modified variant of qubit-ADAPT-VQE (Tang et al., 2021). The results are presented in Table 1. Our results demonstrate that for $H_2 - 4$, $LiH - 4$, and $LiH - 6$, CRLQAS surpasses these other QAS methods, producing not only more compact circuits but also achieving lower absolute energy

Table 1: A tabular representation of noiseless simulation for $H_2-4$, $LiH-4$, $LiH-6$, and $H_2O-8$ molecules. We simulate them in the noiseless scenario for the modified variant of qubit-ADAPT-VQE (Tang et al., 2021), quantumDARTS (Wu et al., 2023) and compare it with our CRLQAS. (★) denotes that the simulation is done using 1-, 2- and 3-qubit parameterized gates, which can be further decomposed into {RX, RY, RZ, CNOT}. N.A. is the abbreviation for Not Applicable, implying that the algorithm failed to improve over the Hartree Fock state energy.

| Molecule | Modified qubit-ADAPT-VQE (★) | | | | CRLQAS | | | | quantumDARTS | | | |
|---|---|---|---|---|---|---|---|---|---|---|---|---|
| | Energy Error | # Params | Depth | # Gates | Energy Error | # Params | Depth | # Gates | Energy Error | # Params | Depth | # Gates |
| $H_2-4$ | $1.9 \times 10^{-2}$ | 38 | 29 | 38 | $\mathbf{7.2 \times 10^{-8}}$ | **7** | **17** | **21** | $4.3 \times 10^{-6}$ | 26 | 18 | 34 |
| $LiH-4$ | $4.6 \times 10^{-6}$ | 47 | 38 | 47 | $\mathbf{2.6 \times 10^{-6}}$ | **29** | **22** | **40** | $1.7 \times 10^{-4}$ | 50 | 34 | 68 |
| $LiH-6$ | $3.7 \times 10^{-2}$ | N.A. | N.A. | N.A. | $6.7 \times 10^{-4}$ | **29** | **40** | **67** | $\mathbf{2.9 \times 10^{-4}}$ | 80 | 54 | 132 |
| $H_2O-8$ | $2.6 \times 10^{-3}$ | N.A. | N.A. | N.A. | $\mathbf{1.8 \times 10^{-4}}$ | **35** | 75 | **140** | $3.1 \times 10^{-4}$ | 151 | **64** | 219 |

errors. Furthermore, in (Ostaszewski et al., 2021b), for the $LiH-6$ molecule, their RL algorithm achieves chemical accuracy only in 7 out of 10 independent seeds. Conversely, utilizing CRLQAS with the same molecule, we achieve solutions across all seeds, showcasing the enhanced stability of CRLQAS in contrast to the RL method of (Ostaszewski et al., 2021b).

It should be noted that we utilized a modified qubit-ADAPT-VQE in our simulations for comparative analysis. Specifically, we replace the typically large fermionic pool of operators with a parameterized gate set consisting of

$$\{RX, RY, RZ, RZZ, RYXXY, RXXYY, Controlled-RYXXY, Controlled-RXXYY\}. \quad (4)$$

The energy errors reported in Table 1 were computed by simulating qubit-ADAPT-VQE in Hartree Fock state (Slater, 1951) for 100 ADAPT-VQE iterations with Adam optimizer (learning rate = $0.1 \times 10^{-2}$, and 500 optimization iterations). Notably, for $LiH-6$ and $H_2O-8$, it fails to improve over the Hartree Fock state, repetitively applying the same gate in all iterations, thus resulting in zero values for parameters, depth, and gates. Finally, we emphasize that we exempt ourselves from doing a fine hyperparameter tuning of qubit-ADAPT-VQE, which might improve its performance.

## 5 CONCLUSION

We developed a curriculum-based reinforcement learning QAS (CRLQAS) algorithm, specifically optimized to tackle the unique challenges of deploying VQE in realistic noisy quantum environments. Our main contribution lies in introducing CRLQAS and analyzing its performance across different quantum noise profiles sourced from real IBM quantum devices like IBM Ourense and IBM Mumbai. Our method achieved the ground-state energy within the precision defined by chemical accuracy while suggesting circuits characterized by minimal gate counts and depths, thereby establishing state-of-the-art performance in the sub-field of quantum architecture search (QAS).

We introduced a depth-aware tensor-based 3D encoding for the agent's state description of the quantum circuit, illegal actions to reduce the agent's search space and to avoid consecutive application of similar quantum gate, a random halting mechanism steering the agent to find shorter circuits, and a novel variant of the simultaneous perturbation stochastic approximation (SPSA) algorithm to reduce the energy function evaluations in the presence of noise.

Our numerical experiments focused on quantum chemistry tasks and demonstrated that CRLQAS outperforms existing QAS algorithms across noiseless and noisy environments for $H_2$ (2-, 3-, and 4-qubit), LiH (4-, and 6-qubit) and $H_2O$ (8-qubit) molecule. In pursuit of these experiments, we significantly enhanced the efficiency of simulating realistic noisy quantum circuits by employing the PTM formalism in the Pauli-Liouville basis, thereby fusing gates with their respective noise models and values.

In essence, owing to the notable adaptability of our approach and a significant six-fold speed-up due to PTM formalism, our approach holds potential for applications in QAS for combinatorial optimization, quantum machine learning, reinforcement learning for quantum computing and quantum reinforcement learning. We have outlined limitations and future work in Appendix A.

## ACKNOWLEDGEMENTS

YJP and OD would like to thank Stefano Polla and Hao Wang for the helpful discussions. YJP and VD acknowledge support from TotalEnergies. AK would like to acknowledge support from the Polish National Science Center under the grant agreement 2019/33/B/ST6/02011, and MO would like to acknowledge support from the Polish National Science Center under the grant agreement 2020/39/B/ST6/01511 and from Warsaw University of Technology within the Excellence Initiative: Research University (IDUB) programme. XBM acknowledges funding from the Quantum Software Consortium. VD and OD were supported by the Dutch Research Council (NWO/OCW), as part of the Quantum Software Consortium programmes (project number 024.003.037 and NGF.1582.22.031). This work was also supported by the Dutch National Growth Fund (NGF), as part of the Quantum Delta NL programme. The computational results presented here have been achieved in part using the ALICE HPC infrastructure of Leiden University and SURF Snellius HPC infrastructure (SURF Cooperative grant no. EINF-6793).

## REPRODUCIBILITY

To ensure the reproducibility of our work, we provide detailed descriptions of the experimental configurations and hyperparameters for the CRLQAS method and Adam-SPSA optimizer in Appendix C and Appendix F, respectively. Additionally, information about the noise profiles of the IBMQ device and molecular systems is available in Appendix G and Appendix L, respectively. A comprehensive discussion of the noise models examined in our study and practical aspects of their software implementation can be found in Appendix H and Appendix I. The source code for all experiments conducted in this manuscript is accessible here: https://anonymous.4open.science/r/CRLQAS/.

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

## A    LIMITATIONS AND FUTURE WORK

**Computational Demands**    The training process for the agent is computationally demanding, posing challenges both in terms of evaluating quantum circuits on a quantum computer and training the algorithm on classical devices. This limitation warrants further exploration for more efficient computational strategies.

**Evolution of RL Methods**    Reinforcement learning (RL) methods are continually evolving, and while promising, they face challenges related to sample efficiency, stability, and sensitivity. Recognizing these evolving aspects is crucial for refining the proposed algorithm and addressing its limitations.

**Validation on Real Quantum Hardware**    A limitation of this work is the absence of validation on real quantum hardware due to current cost constraints. Future research should include experimentation on practical quantum devices to assess the algorithm's performance in real-world scenarios.

**Scalability Challenges**    The proposed algorithm's scalability to larger quantum circuits, more complex quantum chemistry problems, or different noise models is a potential limitation that requires thorough investigation. Current experiments train the agent from scratch, necessitating exploration for scalability improvements.

**Transfer Learning Exploration**    Investigate the feasibility of transfer learning for the proposed algorithm, particularly in the context of different molecules and various noise scenarios. This exploration aims to enhance the algorithm's adaptability and generalization across diverse quantum tasks.

**Application Scenarios Enhancement**    Explore more applicable scenarios, such as pre-training the algorithm on simulations and fine-tuning on real quantum devices. This approach can potentially improve the algorithm's efficiency and performance in practical quantum computing applications.

## B    DESCRIPTION OF CRLQAS COMPONENTS

### B.1    FEEDBACK-DRIVEN CURRICULUM LEARNING

The moving threshold technique (see Fig. 4) is a feedback-driven curriculum learning method introduced in (Ostaszewski et al., 2021b). During the learning process the agent pursues a parameter $\xi_2$ that marks the lowest energy known by the agent so far and updates a threshold parameter with respect to this parameter based on some rules. In the beginning, the $\xi_2$ parameter is set to a hyperparameter $\xi_1$. If the agent finds an energy value lower than the current one, it updates $\xi_2$ to this new energy value. Another hyperparameter "fake minimum energy" $\mu$, a proxy to the lower bound of attainable ground state energy is set as a target for the agent[1]. We compute this proxy by taking the summation of absolute values of Pauli string coefficients stemming from the Hamiltonian. In the absence of amortization, the algorithm shifts the threshold to $|\mu - \xi_2|$ for the new $\xi_2$. In the presence of amortization, however, it adds a parameter to that threshold as $|\mu - \xi_2| + \delta$, where $\delta$ is the amortization hyperparameter. In the meantime, the agent continues its exploration with subsequent actions and episodes and records the number of successful actions. Here, there are two rules at play. The first rule greedily shifts the threshold to $|\mu - \xi_2|$ after $G$ episodes. Here $G$ is a hyperparameter as well. The second rule slowly decreases the threshold parameter each time there is a successful episode by subtracting a factor of $\delta/\kappa$. Here $\kappa$ is the radius of shifts, also a hyperparameter. Upon setting the threshold to $|\mu - \xi_2|$, if the agent fails to improve the energy value in consecutive episodes the threshold is increased back to $|\mu - \xi_2| + \delta$, as demonstrated in Fig. 4. This way the agent is given an opportunity to trace its steps back if it was stuck in a local minimum.

Notably, this method does not require any prior knowledge regarding the true value of the ground state energy and does not impose any specific constraints on the initial threshold value, unlike existing QAS methods in the literature.

---

[1]One can set the target of the agent to such a value for VQE because from Rayleigh's variational principle, the agent theoretically can never attain an energy below the true ground state energy.

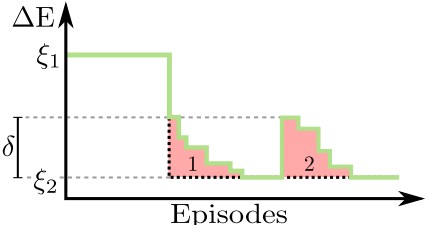

Figure 4: Demonstration of the feedback-driven (green) process, depicting the impact of two amortization occurrences (pink), denoted by $\delta$. The initial occurrence corresponds to a non-zero adjustment in the threshold, transitioning from $\xi_1$ to $\xi_2$, indicating the agent's success in enhancing the energy estimate during training. The subsequent amortization event illustrates the scenario where the agent falls short of improving upon the current threshold $\xi_2$ or the improvement is marginal compared to the amortization value. Consequently, the threshold undergoes a sudden increase due to the reset of the amortization value. It's noteworthy that the ultimate threshold, subsequent to the second amortization reaching zero, may also be less than $\xi_2$.

## B.2    Illegal actions for RL agent

The illegal actions (IA) scheme is an adaptation we developed in this work to prevent the RL agent from choosing actions that either revert or add redundancy to the effect of the previous actions. In our context, each action involves appending a gate to a qubit wire within the circuit. This scheme relies on two heuristics.

The first heuristic centers on the nature of unitary matrices within quantum gates. When adding a unitary (gate) to a qubit wire at a certain circuit moment (layer), appending the same unitary to the same wire in the subsequent moment effectively negates the former's effect. This multiplication results in an identity matrix or an idling operation. Our CRLQAS algorithm is designed to progress forward, consistently increasing the total number of gates in the circuit by appending gates. It does not retract or prune gates dynamically. To restrict redundant idling operations and enhance the RL agent's exploration efficiency, the first rule of IA prohibits adding a CNOT gate on specific wires if the same gate was appended to those wires in the previous layer.

The second heuristic focuses on 1-qubit rotations. When optimizing the parameter of a 1-qubit rotation gate to a certain value $\theta^*$ at a given moment, appending the same rotation gate to the same wire in the next moment introduces redundancy from an optimization perspective. As our CRLQAS continuously optimizes at each step, subsequent rotation gates with the same angle will yield redundant values. Thus, the second rule of IA prevents adding a 1-qubit rotation gate (e.g., RX, RY, RZ) if the same gate was appended to that wire in the previous layer. The RL agent must remain informed about the disallowed actions via a subroutine when it is about to choose an action (gate). Below, we provide implementation details for this IA subroutine.

When the agent is determining its next action, the subroutine scans the three-dimensional tensor representing the circuit to identify the previously added gates. It then translates this information into action numbers based on the number of qubits $N$, presenting it in a format understandable by the RL agent. An example of such a list can be exemplified as the following.

$$A_{\text{illegal}} = [\text{CNOT}\,(i, j, N), \text{RX}\,(k, N)\ldots] \tag{5}$$

Here $i$ and $j$ denote the ctrl and targ qubit wires of a CNOT gate for $N$ qubits, and the $k$ denotes the wire where RX gate was appended. For example, when $N = 4$, $i = 0$, $j = 1$ and $k = 2$ the list takes the following form.

$$A_{\text{illegal}} = [\text{CNOT}\,(0, 1, 4), \text{RX}\,(2, 4)\ldots] \quad \rightarrow [12, 6\ldots] \tag{6}$$

Since the first $N \times 3$ actions are reserved for three 1-qubit rotation directions acting on $N$ qubits in our numbering convention, the action number for RX $(2, 4)$ is 6 in the Eq. 6. Similarly, we use a ctrl major numbering convention, the action number for the CNOT $(0, 1, 4)$ gate is given as 12 as the first action after an array of 1-qubit rotation gate actions. During the selection of the next action, the RL agent updates the Q-table of Q-values corresponding to these action numbers based on the current RL state (quantum circuit) by utilizing a DQN. Without the IA scheme, the agent would typically choose the action with the highest Q-value. However, with the IA scheme, the agent identifies illegal actions using the subroutine's provided list and updates their Q-values to $-\infty$ in the

Q-table. Consequently, when the agent selects actions based on the highest Q-values, those actions with $-\infty$ Q-values (i.e., illegal actions) are effectively discarded.

## B.3 ILLUSTRATION OF TENSOR BASED ENCODING

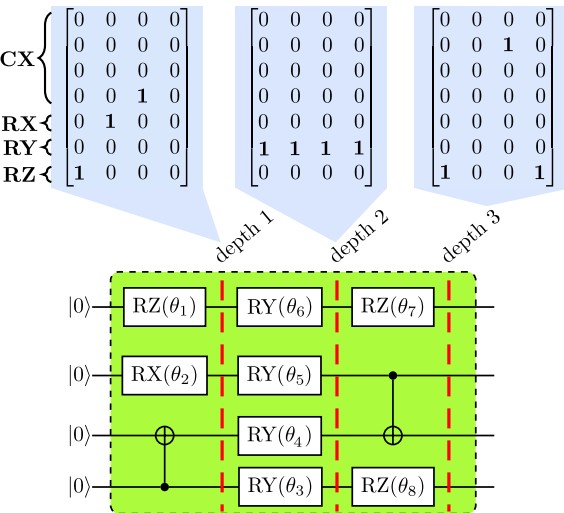

Figure 5: **Illustration of tensor-based encoding for a** $4$**-qubit (i.e.,** $N = 4$**) toy circuit with** $n_{\text{act}} = 3$**.** We initialize a tensor of zeros of dimension $[n_{\text{act}} \times ((N + 3) \times N)]$, equating to $[3 \times ((4 + 3) \times 4)]$ for this circuit. Each blue-colored matrix of size $((4 + 3) \times 4)$, represents a different moment (depth). Within this matrix, the first $(4 \times 4)$ block is reserved for CNOT (CX), with rows and columns encoding target and control qubits, respectively. The remaining $(3 \times 4)$ block of the blue-colored matrix then encodes rotation gates. The columns mark the position of the qubit wire (index) and the rows mark the rotation direction $m$. Here, $m = 1, 2$, and $3$ yields the rotations RX, RY and RZ, respectively.

## C CRLQAS IMPLEMENTATION & HYPERPARAMETERS

In our experiments, we employed the Double deep-Q network (DDQN) algorithm with variable step sizes in the $n$-step trajectory roll-out update (Sutton & Barto, 2018). For noiseless experiments involving simple molecules with low qubit numbers, such as $H_2 - 2$, $H_2 - 3$, $H_2 - 4$, and LiH $- 4$, we used a single step, $n = 1$. For noisy simulations of simple molecules, such as $H_2 - 2$, $H_2 - 3$ with the IBM Mumbai noise model, and $H_2 - 4$ with the IBM Ourense noise model, we used $n = 5$ steps. Conversely, for noiseless simulations of complex molecules LiH $- 6$ and $H_2O - 8$, and for the relatively challenging task of simulating LiH $- 4$ with the IBM Mumbai noise model, we employed $n = 6$ steps.

In these configurations, we set the discount factor ($\gamma$) to $0.88$. We implemented an $\epsilon$-greedy policy for selecting random actions, with $\epsilon$ decaying by a factor of $0.99995$ per step from an initial value of $\epsilon = 1$ until it reached a minimum value of $\epsilon = 0.05$. The memory replay buffer size was fixed at $20000$, and the target network in the DQN training process[2] was updated after every $500$ actions. In the curriculum learning strategy, we implemented a testing phase after every $100$ training episodes. In this testing phase we set the randomness factor to $\epsilon = 0$ to halt the random exploration, and ensure a set of deterministic actions. We exclude the experiences acquired in this phase from the memory replay buffer. We greedily adjusted the threshold after $G = 500$ episodes for both noiseless and noisy 2-, 3-, and 4-qubit problems. Conversely, for 6-, and 8-qubit problems, the threshold underwent adjustments every $G = 2000$ episodes, with an amortization radius set at $\delta = 0.0001$. This amortization radius decreased by $\delta/\kappa = 0.00001$ after every $50$ successfully solved episodes, beginning from an initial threshold value of $\xi_1 = 0.005$.

We conducted simulations of noiseless quantum circuits using the Qulacs library on CPU (Suzuki et al., 2021). For the simulations of noisy quantum circuits, we utilized the JAX library on two computing clusters equipped with NVIDIA-A100 GPUs (Bradbury et al., 2018). In experiments without

---

[2]The neural network was trained using Adam optimizer (Kingma & Ba, 2014).

finite-sampling (shot) noise, we employed the gradient-free COBYLA optimizer (Powell, 1994) with default hyperparameter settings from Scipy (Virtanen et al., 2020) and 1000 iterations to optimize circuit parameters at each step of the RL episode. In the presence of shot noise, we employed the $m$-stage Adam-SPSA (where $m$ is an integer) developed in this work. Specifically, we used $m = 1$ stage Adam-SPSA for two and three qubit problems, and $m = 3$ stages for four qubit problems. Unlike the local optimization approach for circuit parameters, where only the angles of the latest appended parametrized gate are optimized, we adopted a global optimization approach (Ostaszewski et al., 2021b). In each step, we used the circuit parameters from the previous step as initial values, but we optimized all the parameters.

The hyperparameters of the Double deep-Q network algorithm were selected through coarse grain search. The employed network architecture consisted of fully connected network with 5 hidden layers, each with 1000 neurons for the noiseless 4-qubit case, 2000 neurons for the 6-qubit case, and 5000 neurons for the 8-qubit case. In the noisy case, however, we employed 1000 neurons at each layer for the 2-, and 3-qubit problems, while the number of neurons per layer ranged upto 2000 neurons for the 4-qubit problem depending on the complexity of the problem. Simulating 1-qubit depolarizing noise required only 1000 neurons per layer, but the noise model of IBM-Mumbai devices required up to 2000 neurons for the 4-qubit problem. Similarly, in the noiseless case, we capped the maximum number of gates at 40 for 4-qubit problems, 70 for 6-qubit problem, and 250 for 8-qubit problem. In the noisy case, we capped the maximum number of gates at 40 for 2-, and 3-qubit problems, but it ranged up to 60 gates for the 4-qubit problems depending on the complexity of the noise model.

## D    COMPARISON WITH QCAS (DU ET AL., 2022)

Table 2: Comparison summary between our CRLQAS and QCAS for the 4-qubit $H_2$ molecule. The average is taken over 3 seeds. The bold numbers highlight the optimal performance of the CRLQAS algorithm. For both CRLQAS (RH) and CRLQAS (wo-RH), the first row represents the minimum achievable error and the corresponding number of gates required for these settings. Additionally, the second row demonstrates the minimum number of gates needed to achieve errors just below the chemical accuracy threshold. Notably, using only 10 gates in both settings allows for better accuracy to the target. Each approach, RH and wo-RH, presents its unique advantages and disadvantages. The wo-RH setting achieves energy estimates significantly below chemical accuracy. Conversely, using RH yields slightly less accurate energy estimates but with shallower circuits.

| Method | Minimum Noiseless Err. | Avg. Noiseless Minimum Err. | # Gates |
|---|---|---|---|
| CRLQAS (RH) | $\mathbf{8 \times 10^{-5}}$ | $1.7 \times 10^{-4}$ | 32 |
| | $\mathbf{2.8 \times 10^{-4}}$ | | **10** |
| CRLQAS (wo-RH) | $\mathbf{1.5 \times 10^{-5}}$ | $8.7 \times 10^{-4}$ | 40 |
| | $\mathbf{5.5 \times 10^{-5}}$ | | **10** |
| QCAS (Du et al., 2022) | $1.9 \times 10^{-2}$ | | 16 |

# E   ABLATION STUDY OF DIFFERENT COMPONENTS OF CRLQAS

Table 3: Results of ablation study for CRLQAS method. We conducted a thorough investigation to identify the features that standout within the CRLQAS method in both noisy (for 4-qubit LiH and $H_2$) and noiseless (for 6-qubit LiH) settings. Initially, we find episode(s) with the best noiseless error(s), gathering circuit statistics (depth, gate count, etc.) and then acquire the noisy error(s) for these episode(s). The last two columns provide insights into the learning performance of the CRLQAS method. We present median statistics over three random seeds for noisy experiments and five seeds for noiseless settings. Notably, we emphasize noiseless errors (in both noisy and noiseless settings) achieving chemical accuracy and denote in bold. wo-X denotes the deactivation of feature X, where $X \in \{IA, RH\}$. Additionally, N.A. denotes Not Applicable, implying that none of the agents over three seeds achieved chemical accuracy.

| Molecule | Environment | Noisy Err. | Noiseless Err. | Depth | # Gates | # Params | Act. to 1st Succ. Ep. | Succ. Ep. |
|---|---|---|---|---|---|---|---|---|
| LiH − 4 | (wo-IA, wo-RH, `Mumbai` Median) | 0.0581 | 0.0026 | 24 | 40 | 33 | N.A. | N.A. |
| LiH − 4 | (IA, wo-RH, `Mumbai` Median) | 0.0818 | **0.0015** | 23 | 40 | 30 | 66221 | 1 |
| LiH − 4 | (wo-IA, RH, `Mumbai` Median) | 0.0535 | 0.0029 | 15 | 29 | 21 | N.A. | N.A. |
| LiH − 4 | (IA, RH, `Mumbai` Median) | 0.0670 | 0.0024 | 15 | 29 | 23 | N.A. | N.A. |
| LiH − 4 | (IA, RH, `Mumbai` Median & shot noise) | 0.0735 | 0.0038 | 17 | 28 | 18 | N.A. | N.A. |
| $H_2$ − 4 | (IA, wo-RH, `Ourense`) | 0.2382 | **0.0001** | 26 | 40 | 14 | 59311 | 1209 |
| $H_2$ − 4 | (wo-IA, wo-RH, `Ourense`) | 0.3118 | **0.0004** | 29 | 40 | 23 | 58157 | 10 |
| $H_2$ − 4 | (IA, RH, `Ourense`) | 0.1372 | **0.0002** | 27 | 32 | 5 | 42331 | 217 |
| $H_2$ − 4 | (wo-IA, RH, `Ourense`) | 0.1522 | **0.0003** | 19 | 27 | 22 | 35593 | 6 |
| LiH − 6 | ((Ostaszewski et al., 2021b), noiseless) | – | 0.0024 | 24 | 55 | 29 | N.A. | N.A. |
| LiH − 6 | (IA, wo-RH, noiseless) | – | **0.0008** | 27 | 56 | 26 | 356604 | 30 |
| LiH − 6 | (IA, RH, noiseless) | – | **0.0012** | 30 | 59 | 37 | 641593 | 3 |
| LiH − 6 | (wo-IA, RH, noiseless) | – | 0.0037 | 23 | 45 | 25 | N.A. | N.A. |
| LiH − 6 | (wo-IA, wo-RH, noiseless) | – | 0.0016 | 31 | 62 | 31 | N.A. | N.A. |

# F   SPECIFICS FOR ADAM-SPSA

We implement a version of simultaneous perturbation stochastic approximation (SPSA) that was not implemented in the context of VQE before. It combines stochastic gradient estimates of SPSA (Spall, 1987) with adaptive moment estimation (Adam) (Kingma & Ba, 2014) leading to more stability and faster convergence while retaining the noise robustness of SPSA. To estimate a gradient term at a given iteration, the SPSA algorithm randomly samples a number of angles amount of binary directions from the Rademacher distribution, denoted by $\Delta_k$. By adding and subtracting these shifts from the current set of angles $\theta_\pm = \theta \pm \Delta_k$, we acquire two sets of angles such that we evaluate the cost function twice (two function evaluations per iteration) there to acquire the stochastic gradient approximation $\nabla J_k = \frac{g_k^+ - g_k^-}{2c_k\Delta_k}$. Then the algorithm proceeds similarly to standard gradient descent $\theta_{k+1} = \theta_k - a_k\nabla J_k$, with the exception of both parameter shift scaling parameters $c_k$ and the learning rates $a_k$ decay at each iteration $k$ slowly with respect to hyperparameters $\alpha$, $a$, $c$, $\gamma_{sp}$.

Similar to incorporating adaptive moment terms (parameter adaptation and momentum) in gradient descent optimization, we integrate these terms with the stochastic gradient estimate from SPSA. In this integration, three additional hyperparameters, namely $\beta_1$, $\beta_2$, and $\lambda$, are introduced to govern the adaptation and momentum terms. In the quantum context, Adam enables the utilization of gradient information from previous iterations in classical post-processing, without relying on additional quantum queries, thereby enhancing robustness and convergence rates. After testing multiple variants of SPSA in a well-known problem of VQE through hardware efficient ansatz (Agliardi & Prati, 2022; Arouri & Sayyafzadeh, 2020; Kandala et al., 2017), we found the variant used in (Leng et al., 2019) to be the most stable. Throughout this work, we refer to this specific variant as Adam-SPSA (Leng et al., 2019). In this version of Adam-SPSA, the momentum term $\beta_1$ undergoes updates

Table 4: The hyperparameters of Adam-SPSA optimizer used during the noisy simulations. In the noisy simulation of 2-, and 3-qubit problems we used 1-stage version of the algorithm, therefore only single maximum function evaluation hyperparameters are given. The parameters within the curly brackets denote the maximum number of function evaluations in the 3-stage version of the algorithm. We provide Max fevals both for 1-stage and their 3-stage equivalents.

| Molecule | $a$ | $\alpha$ | $\beta_1$ | $\beta_2$ | $c$ | $\gamma_{sp}$ | $\lambda$ | Max fevals | Shots |
|---|---|---|---|---|---|---|---|---|---|
| $H_2$-2 | 1.2104 | 0.9531 | 0.9414 | 0.9983 | 0.1039 | 0.0984 | 0.9277 | 500 | $10^3$ |
| $H_2$-3 | 0.5188 | 0.9859 | 0.716 | 0.6265 | 0.0938 | 0.0974 | 0.6483 | 500 | $10^4$ |
| LiH-4 | 1.2324 | 0.9709 | 0.6114 | 0.9326 | 0.2215 | 0.1485 | 0.9772 | 1600      3300 
 {1191, 357, 119} {2383, 715, 238} | $10^6$ |
| LiH-6 | 1.7564 | 0.8365 | 0.6841 | 0.9048 | 0.1068 | 0.1549 | 0.1223 | 2000 
 {1430, 429, 143} | $10^8$ |

at each iteration using the $\lambda$ scaling hyperparameter, ensuring numerical stability. Conversely, the other momentum term $\beta_2$ remains constant. The pseudo-code for Adam-SPSA is outlined in Alg. 12, with the newly introduced Adam momentum components highlighted in green.

In the presence of finite sampling (or shot) noise, keeping the number of measurement samples, $N_{shots}$, constant throughout the training is named 1-stage optimization (Cade et al., 2020; Bonet-Monroig et al., 2023). Similarly, a 3-stage version of vanilla SPSA is proposed in (Cade et al., 2020; Bonet-Monroig et al., 2023) such that the number of measurement samples is increased at each phase. In the 3-stage optimizers, the first phase is implemented with a shot budget of $N_{shots}^{(1)} = N_{shots}/10$ for a function evaluation budget of $n_f^{(1)}$, the second with a shot budget of $N_{shots}^{(2)} = N_{shots}$ for a function evaluation budget of $n_f^{(2)}$, and the third with a shot budget of $N_{shots}^{(3)} = 10N_{shots}$ for a function evaluation budget of $n_f^{(3)}$. The 3-stage SPSA algorithm introduced in these papers resets the decaying hyperparameters to their default values during transitions between stages.

In this work, we propose a 3-stage Adam-SPSA where the decaying hyperparameters are continuously evolving (i.e., not reset to defaults) while changing between stages such that the momentum of the iterations from higher measurement samples can be utilized in the later stages. Our proposed algorithm also reports the latest function evaluation as the best function evaluation, unlike the others. After examining various versions of SPSA, considering factors like the inclusion of Adam, and experimenting with or without parameter reset, we observed that the SPSA variants without parameter reset are constrained to use the best function evaluation because they significantly diverge from the solution after a certain number of iterations. Our simulations empirically show that the continuity of hyperparameters between stages leads the algorithm to converge towards the optimum while Adam increases the rate of such convergence.

We conducted experiments involving various variants of 1-, and 3-stage SPSA, both with and without Adam, and with and without parameter reset. The evaluation was performed on 2-, 3-, 4-, and 6-qubit systems utilizing the VQE approach with a hardware-efficient ansatz of depth 10. For the 2-qubit problem addressing $H_2 - 2$, we employ $10^3$ samples. Similarly, the 3-qubit problem targeting $H_2 - 3$ utilizes $10^4$ samples, and the 4-qubit problem for LiH-4 employs $10^6$ samples. Lastly, the 6-qubit problem dealing with LiH-6 utilizes $10^8$ samples in each function evaluation. We fine-tuned the hyperparameters of the SPSA variants for these problems using an evolutionary algorithm-based stochastic hyperparameter optimizer library called IRACE in R (López-Ibáñez et al., 2016). The final hyperparameters used in the experiments are outlined in Table 4. Specifically, we tuned the hyperparameters within the following ranges for each hyperparameter: $a \in [0.01, 2]$, $\alpha \in [0, 1]$, $c \in [0.01, 2]$, $\gamma_{sp} \in [0, 1/6]$, $\lambda \in [0.01, 0.999]$, and $\beta_1, \beta_2 \in [0.6, 0.999]$. For running the IRACE algorithm, we allocated 2500 evaluations of the hyperparameters as the total budget. We implemented a single run of IRACE using this budget, and utilized the F-test to eliminate worse configurations.

After the hyperparameter optimization, we executed each combination (for each SPSA variant and each problem) in 100 independent runs for the qubit systems described above. Both Fig. 6 and Fig. 7 illustrate the optimization traces of SPSA variants with 1-, and 3-stage sampling strategy, respec-

**Algorithm 1:** Simultaneous Perturbation Stochastic Approximation with Adam (Adam-SPSA)

| | |
|---|---|
| **Input** | : Initial parameter vector $\theta$, Objective function $f(\theta)$, Number of iterations $K$ |
| **Output** | : Optimal parameter vector $\theta^*$ |
| **Hyperparameters:** | $a$, $\alpha$, $c$, $\gamma_{sp}$, $\lambda$, $\beta_1$, $\beta_2$ |

**1** Initialize momentum parameters to zero: $m, v \leftarrow 0$

**2 for** $k = 1$ **to** $K$ **do**

**3**     Compute the scaling parameters $a_k \leftarrow \frac{a}{(k+1)^\alpha}$,     $c_k \leftarrow \frac{c}{(k+1)^{\gamma_{sp}}}$

**4**     Compute the hyperparameter $\beta_{1,k} \leftarrow \frac{\beta_1}{(k+1)^\lambda}$

**5**     Randomly choose a perturbation vector $\Delta_k$ with elements $\pm 1$

**6**     Evaluate objective function gradients: $g_k^+ \leftarrow f(\theta + c_k \Delta_k)$ and $g_k^- \leftarrow f(\theta - c_k \Delta_k)$

**7**     Compute gradient estimate: $\nabla J_k \leftarrow \frac{g_k^+ - g_k^-}{2 c_k \Delta_k}$

**8**     Biased update of moment parameters $m$ and $v$: $m \leftarrow \beta_{1,k} m + (1 - \beta_{1,k}) \nabla J_k$, $v \leftarrow \beta_2 v + (1 - \beta_2)(\nabla J_k)^2$

**9**     Unbiased computation of moment parameters $\hat{m}$ and $\hat{v}$: $\hat{m} \leftarrow \frac{m}{1 - \beta_{1,k}^{k+1}}$,     $\hat{v} \leftarrow \frac{v}{1 - \beta_2^{k+1}}$

**10**     Update gradient estimate: $\nabla J_k \leftarrow \frac{\hat{m}}{\sqrt{\hat{v}} + k}$

**11**     Update parameters: $\theta \leftarrow \theta - a_k \nabla J_k$

**12 return** $\theta^* = \theta$

tively. The colors show the results of the vanilla SPSA (brown and black) and Adam-SPSA (orange and grey). The thick lines (black and grey) on top of individual optimization traces indicates the median of 100 independent runs. The markers (brown and orange) refer to the final candidates of

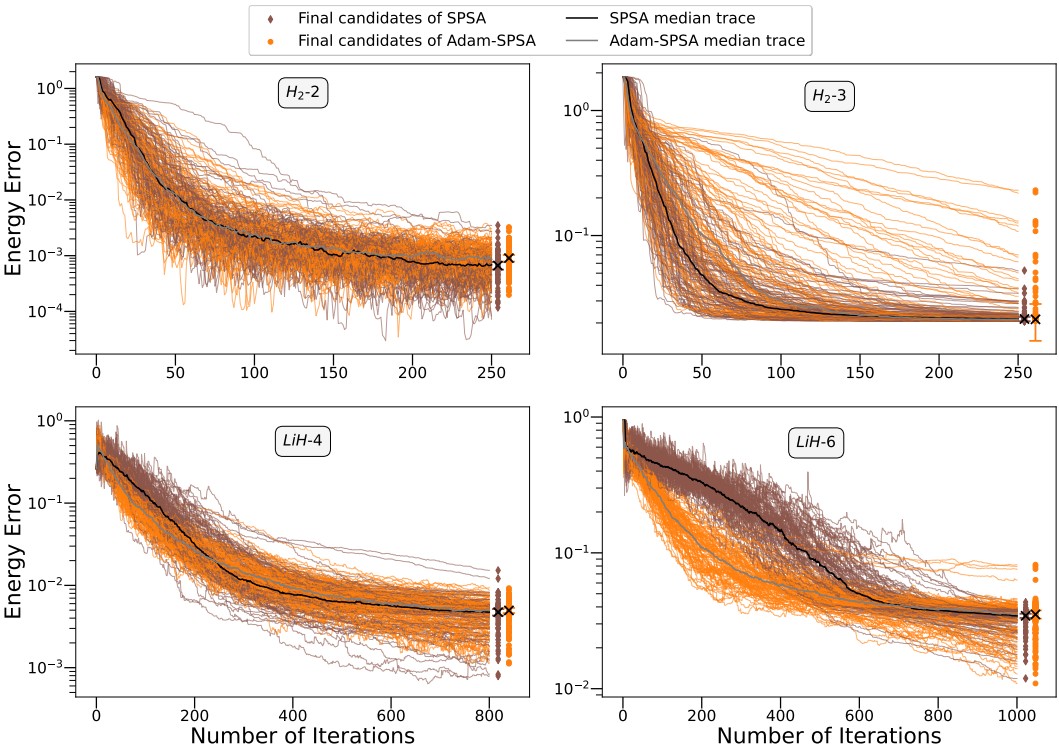

Figure 6: **Optimization traces of the one-stage sampling strategy of SPSA (brown and black) and Adam-SPSA (orange and grey) on the** 2-, 3-**qubit H$_2$ and** 4-, 6-**qubit LiH molecules (whitesmoke text-box), using the hyperparameters outlined in Table 4.** The individual traces are represented by thin lines, while the thick line on top indicates the median of 100 independent runs.

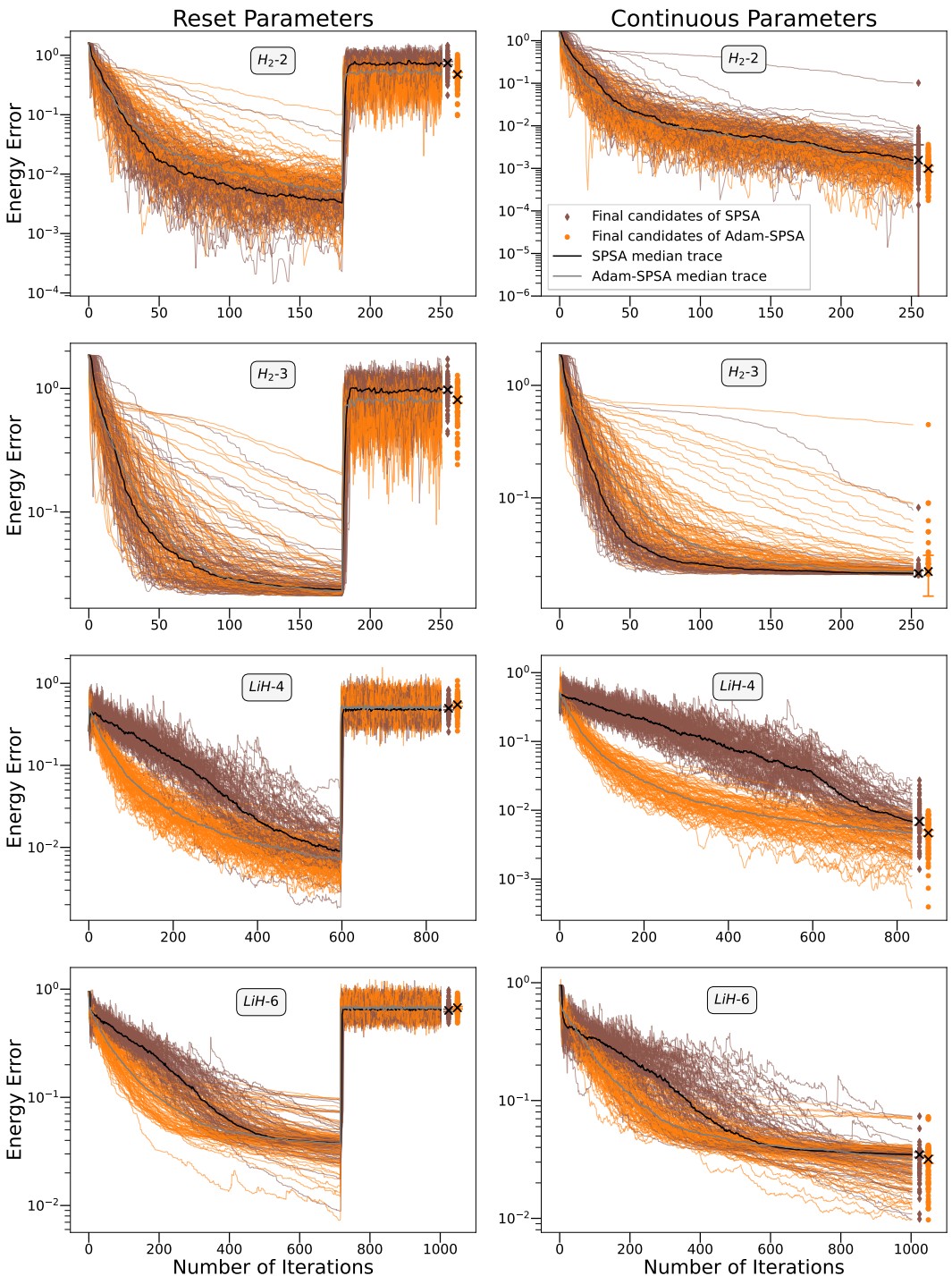

Figure 7: **Optimization traces of the three-stage sampling strategy of SPSA (brown and black) and Adam-SPSA (orange and grey) on the** 2**-,** 3**-qubit** $H_2$ **and** 4**-,** 6**-qubit LiH molecules (whitesmoke text-box), using the hyperparameters outlined in Table** 4**.** The individual traces are represented by thin lines, while the thick line on top indicates the median of 100 independent runs. The left and right panels showcase the resetting and continuous evolution of SPSA (Adam-SPSA) hyperparameters, respectively.

SPSA (Adam-SPSA) after every optimization run. The error bars are one-sigma standard-error of 100 independent runs. The y-axis in both the figures is given in log-scale. In Fig. 7, the left and right panels showcase the resetting and continuous evolution of 3-stage SPSA (Adam-SPSA) hyperpa-

rameters, respectively. Note that the number of iterations (maximum function evaluations) outlined in Table 4 are different than the hyperparameters generated for this systematic benchmarking study.

In both 1-, and 3-stage Adam-SPSA, unlike the vanilla SPSA without Adam momentum, the convergence towards the minima is qualitatively much faster. In 3-stage variants, as seen in Fig. 7, this convergence behaviour is also apparent quantitatively (especially in $LiH - 4$ and $LiH - 6$), though the differences are marginal and inconclusive for the 1-stage in Fig. 6 (except for $LiH - 6$ until 700 iterations). Moreover, for 3-stage variants, we observe that the variant with *resetting of hyperparameters* (left panel in Fig. 7) tend to diverge from the optimal cost function values after a certain number of iterations only to converge back to the optima after training for longer. We noted this convergence behavior is at odds with the fast convergence rates we are looking for in our RL training. In contrast, the variant with *continuous evolution of hyperparameters* (right panel in Fig. 7) do not suffer from this draw back as the cost function values consistently went lower with the number of iterations without making a sharp turn away from the optima. Utilizing these insights from our analysis of various SPSA variants, we employ 1- and 3-stage Adam-SPSA (with continuous evolution of hyperparameters) in our noisy experiments. This helps cut down the total number of function evaluations by half, thereby doubling the speed of our RL training. This improvement at the algorithm level helped us simulate noisy systems that suffer from computational complexity and large run times.

## G  IBM MUMBAI DEVICE NOISE CALIBRATION DATA

Table 5: Tabular representation of the median, max and $10\times$max noise of IBM Mumbai device. Additionally, the qubit frequency and the anharmonicity are the same for median, max and $10\times$max settings and are set to 4.896 GHz and $-0.33$ GHz, respectively.

| Noise Profile | 1-Qubit Dep. Noise | 2-Qubit Dep. Noise | Readout Error | Thermal Relaxation Noise ($\mu$s) | 1-Qubit Gate Time (s) | 2-Qubit Gate Time (s) |
|---|---|---|---|---|---|---|
| Median | $2.44 \times 10^{-4}$ | $8.25 \times 10^{-3}$ | $2.25 \times 10^{-2}$ | $T_1 = 122.28\mu s,$ $T_2 = 167.2\mu s$ | $35 \times 10^{-9}$ | $416 \times 10^{-9}$ |
| Max | $1.45 \times 10^{-3}$ | $2.30 \times 10^{-2}$ | $8.7 \times 10^{-2}$ | $T_1 = 122.28\mu s,$ $T_2 = 167.2\mu s$ | $35 \times 10^{-9}$ | $739.55 \times 10^{-8}$ |
| $10\times$Max | $1.45 \times 10^{-2}$ | $2.30 \times 10^{-1}$ | $8.7 \times 10^{-1}$ | $T_1 = 122.28\mu s,$ $T_2 = 167.2\mu s$ | $35 \times 10^{-8}$ | $739.55 \times 10^{-8}$ |

## H  DETAILS OF QUANTUM NOISE MODELS

We implemented multiple noise models of varying complexity to serve as a testbed for evaluating the proposed CRLQAS method. Initially, we modeled sampling (or shot) noise as independently and identically distributed random variables sampled from a Gaussian distribution with zero mean (Bonet-Monroig et al., 2023; Liu et al., 2022). Secondly, we modeled gate infidelities of 1-, and 2-qubit gates as 1-, and 2-qubit depolarizing channels (Nielsen & Chuang, 2010). Thirdly, we incorporated two physical models for state preparation and measurement (SPAM) noise. State preparation errors were modeled by considering initial states as thermal states due to the residual thermal population of transmons. Read-out errors were modeled as bit-flip channels applied at the end of the circuit. Lastly, we modeled thermal relaxation and decoherence using the thermal relaxation channel, especially when relaxation times $T_1$ were smaller than the coherence time $T_2$ (Blank et al., 2020).

We manually specified parameters for the first two noise models. Subsequently, we obtained noise parameters from the benchmarks of the IBM Mumbai device on August 8, 2023, to create more realistic noise models. By obtaining Kraus matrices for the gates and noise channels, we computed Pauli-transfer matrices (PTMs) offline, eliminating the need for online computations during experiments. We combined the PTMs of gate implementations with those of subsequent noise channels, effectively obtaining noisy gate PTMs.

### H.1  SAMPLING NOISE

The VQE cost function, denoted as $C(\theta)$ in Eq. 1, can be expressed in terms of random variables $X_i = c_i P_i$. In this scenario, with given parameters $\vec{\theta}$ and the observable (chemical Hamiltonian)

$H = \sum_i c_i P_i$, then the cost represents the mean of the sum of $n$ such random variables:

$$C(\theta) = \sum_{i=1}^{n} \langle X_i \rangle = \sum_{i=1}^{n} c_i \langle P_i \rangle \tag{7}$$

In real quantum devices, users have access to a finite sample estimator. The expectation value of each Pauli string is estimated as $\overline{P}_i$ through multiple state preparations, basis transformations, and measurements. For each instance, the initial state is reset, and the quantum state undergoes a basis transformation into the computational basis of the given Pauli string to compute a bit-string sample. These bit-strings are sampled $M$ times (shots) to estimate the eigenvalues $\overline{P}_i$ with a variance of $\text{Var}(P_i) = 1/M$. Using these estimates $\overline{X}_i = c_i \overline{P}_i$, and given parameters $\theta$ and shots per observable $M$, the estimator for the cost function is as follows:

$$\bar{C}(\theta, M) = \sum_{i=1}^{n} \bar{X}_i = \sum_{i=1}^{n} c_i \bar{P}_i \tag{8}$$

Here, the difference between each Pauli-string estimate and the expectation value is denoted by a random variable $\varepsilon_i(M) = \overline{P}_i - \langle P_i \rangle$, drawn from the binomial distribution with variance $\text{Var}(P_i)$ (Bonet-Monroig et al., 2023). Assuming Pauli string observables are measured independently, they can be modeled as independent and identically distributed (i.i.d.) random variables (Bonet-Monroig et al., 2023). In that case, the variance of $\overline{C}(\vec{\theta}, M)$ can be propagated directly as follows:

$$s_n^2 = \text{Var}\left[\bar{C}\right] = \sum_{i=1}^{n} c_i^2 \sigma_i^2 = \frac{1}{M} \sum_{i=1}^{n} c_i^2 \tag{9}$$

According to the Central Limit Theorem, in the limit $n \to \infty$, the difference between the true cost function and its finite-sample-based estimator converges to a normal distribution centered around zero mean (Billingsley, 2017):

$$\lim_{n \to \infty} \sum_{i=1}^{n} c_i \left( \bar{P}_i - \langle P_i \rangle \right) = \lim_{n \to \infty} \sum_{i=1}^{n} c_i \varepsilon_i(M) \sim \mathcal{N}(0, s_n) \tag{10}$$

This expression illustrates that, in the limit where the chemical Hamiltonian has numerous Pauli-string terms ($n \gg 1$), sampling Pauli-string estimation errors from the normal distribution $\varepsilon(M) \sim \mathcal{N}(0, \frac{1}{\sqrt{M}})$ is a reasonable approximation compared to the binomial distribution. This approximation serves as a computationally efficient model for sampling (shot) noise and is consistent with other literature (Liu et al., 2022).

## H.2 Pauli-transfer Matrices

A density matrix $\rho$, a complex-valued object of dimension $2^N \times 2^N$ for $N$ qubits, can represent both the quantum statistics of a single quantum state $\rho = |\psi\rangle\langle\psi|$ (known as a pure state) and the statistics of a classical ensemble of multiple quantum states $\rho = \sum_i p_i |\psi_i\rangle\langle\psi_i|$ (known as a mixed state). In the case of mixed states, each constituent quantum state $|\psi_i\rangle$ occurs with a probability $p_i$, which sums up to unity. This happens because coupling with external processes like measurement or thermal relaxation, applies a (non)unitary process to the quantum state with some probability (Breuer & Petruccione, 2002). Both the unitary processes, normally acting in closed quantum systems, and the above-mentioned processes can be represented by quantum channels $\Lambda$. These channels are completely positive trace-preserving (CPTP) operators in the $2^N$ dimensional Hilbert space, mapping a density matrix to another, $\Lambda : \mathbb{C}^{2^N \times 2^N} \mapsto \mathbb{C}^{2^N \times 2^N}$. A quantum channel $\Lambda$ acting on $\rho$ is conventionally represented using Kraus matrices $K_i$:

$$\Lambda(\rho) = \sum_i K_i \rho K_i^\dagger \tag{11}$$

The Kraus matrix usually has the form $K_i = \sqrt{p_i} A_i$. In the case of a unitary channel (e.g., a quantum gate in a closed system), only a single unitary matrix $A_i$ is applied with a unit probability ($p_i = 1$), keeping the quantum state pure. In open, noisy systems, (non)unitary operations $A_i$ are applied with probabilities $p_i < 1$, resulting in a classical mixture of possible outcomes (i.e., a mixed state). In digital quantum computers, physical noise is typically represented by applying the unitary

channel of a gate on a qubit, followed by the application of various quantum channels representing gate noise on the same qubit. It is computationally advantageous to represent each of these channel applications as a matrix product, requiring vectorization of the density matrix through algebraic manipulation.

$$\rho = \sum_{i,j} \rho_{i,j} \, |i\rangle \langle j| \quad \rightarrow \quad |\rho\rangle_{\text{Choi}} = \Phi^{-1}(\rho) = \sum_{i,j} \rho_{i,j} \, |i\rangle \otimes |j\rangle \tag{12}$$

Here, representing the computational basis states given in outer-product format $|i\rangle \langle j|$ instead of the tensor product $|i\rangle \otimes |j\rangle$, allows for vectorization in the column-major order of the density matrix. This unrolling operation of the matrix, denoted as "vec" ($\Phi^{-1}(\cdot)$), results in the Choi representation of the density matrix. Conversely, the rolling operation back in the column-major order is known as "unvec" ($\Phi(\cdot)$) (Wood et al., 2011). The isomorphism $\Phi^{-1}$ is a mapping from $\mathbb{C}^{2^N \times 2^N}$ to $\mathbb{C}^{2^{2N}}$ in column-major ordering, and vice versa (Blank et al., 2020). In this isomorphism, the application of Kraus matrices can be expressed as a single matrix product:

$$\varepsilon(\rho) = \text{Tr}_1 \left\{ \Lambda \left( \rho^T \otimes I \right) \right\} \tag{13}$$

Here, the Choi matrix (or super-operator) $\varepsilon$ of a channel $\Lambda$ is obtained by tracing out the subspace of identity $I$. This matrix then can be used to evolve quantum states under the influence of a quantum channel by matrix-vector multiplications, $|\rho'\rangle_{\text{Choi}} = \varepsilon_\Lambda \cdot |\rho\rangle_{\text{Choi}}$. In the context of processes characterization (Chow et al., 2012) and classical simulation of variational quantum algorithm landscapes (Rall et al., 2019; Fontana et al., 2023; Rudolph et al., 2023) that involve measurements using the Pauli-Liouville bases instead of the computational bases for the vectorization and super-operator representation (Pauli-transfer Matrix, PTM) have an extra computational advantage. In this formalism, the state, the observable, and the channel super-operators are written either using the Pauli basis set $B_{pauli} = \frac{1}{\sqrt{2}} \{I, X, Y, Z\}$ or the $0xy1$ basis set $B_{0xy1} = \left\{ (I+Z)/2, X/\sqrt{2}, Y\sqrt{2}, (I-Z)/2 \right\}$ (O'Brien et al., 2017; Greenbaum, 2015). A standard quaternary notation of these bases is used to construct multi-qubit Pauli strings, $P_i$. To acquire the expansion coefficients in this basis, the Hilbert-Schmidt norm, $\langle A|B\rangle = \text{Tr}\left( A^\dagger B \right)$, is utilized.

$$\rho = \sum_i c_i P_i \quad \rightarrow \quad c_i = |\rho\rangle_i = \text{Tr}\left( P_i \rho \right)$$

$$O = \sum_i d_i P_i \quad \rightarrow \quad d_i = |O\rangle_i = \text{Tr}\left( P_i O \right)$$

$$R_\Lambda[i,j] = \text{Tr}\left\{ P_i \Lambda \left( P_j \right) \right\} \quad \rightarrow \quad \Lambda\left( P_j \right) = \sum_i K_i P_j K_i^\dagger \tag{14}$$

Here, $|\rho\rangle$ and $|O\rangle$ represent the vector representation of the state and a physical observer (e.g., Hamiltonian) in Pauli-Liouville basis. The $i$-th element of these vectors is denoted by $c_i$ and $d_i$, which are scalars. While $R_\Lambda[i,j]$ represents the PTM element at the $i$-th row and $j$-th column. In this formalism, the channel PTMs can both propagate the initial state forward (Schrodinger propagation) or propagate the observable backward (Heisenberg propagation) to acquire the final expectation via matrix-vector multiplications (Rall et al., 2019; O'Brien et al., 2017; Fontana et al., 2022; 2023). These vectorization and super-operator schemes enable fusing gate channels with the subsequent noise channels offline (ahead of the simulations for RL training). The PTM of a noisy gate is given by fusing the PTMs of $K$ channels acting on the qubit after the PTM of the gate $G$ as $\tilde{R}_G = R_{\Lambda_{K-1}} \circ R_{\Lambda_{K-2}} \cdots \circ R_{\Lambda_0} \circ R_G$. In our simulations, we use $0xy1$ bases for $N$-qubit PTMs, due to their similarity with noiseless $2N$-qubit state-vector simulation, and we apply the noise models of the real devices with actual parameters we describe in the following sub-sections.

## H.3 Depolarizing Channel

We model the gate infidelities as 1-qubit or 2-qubit depolarizing channels (Nielsen & Chuang, 2010). The depolarizing channels can be represented using density matrix formalism, $\rho$, and Kraus matrices, $K_i$. Specifically, a 1-qubit depolarizing channel, $\Lambda_{p,q}^{(dep)}$, acting on qubit $q$ with an error probability (or the noise strength, gate infidelity, etc.) $p$, is defined as follows:

$$\Lambda_{p,q}^{(dep)}(\rho) = \left( 1 - \frac{3p}{4} \right) \rho + \frac{p}{4} \left( X_q \rho X_q + Y_q \rho Y_q + Z_q \rho Z_q \right) \tag{15}$$

In the expression above, $\Lambda_{p,q}^{(dep)}(\rho)$ represents the channel's application on the density matrix, and $X_q, Y_q, Z_q$ denote the Pauli operators acting on qubit $q$. The qubit is left unaffected with a probability of $\left(1 - \frac{3p}{4}\right)$, and with a probability of $\frac{p}{4}$, it undergoes an amplitude bit-flip ($X_q \rho X_q$), a phase-flip ($Z_q \rho Z_q$), or a combination thereof ($Y_q \rho Y_q$).

Initially, we apply this noise model with the probability parameter $p_{\text{one}} = 0.1 \times 10^{-2}$ upon implementation of each 1-qubit gate. In other words, a 1-qubit depolarizing channel of the strength $p_{\text{one}} = 0.1 \times 10^{-2}$ is initially utilized to model the rotation gate infidelities. Subsequently, we apply this model to both the control and target qubits, $q_{\texttt{ctrl}}$ and $q_{\texttt{targ}}$, during the implementation of the 2-qubit control NOT gate $\texttt{CNOT}(q_{\texttt{ctrl}}, q_{\texttt{targ}})$. In this case, a 2-qubit depolarizing channel is used to model the $\texttt{CNOT}$ gate infidelity, applying this 2-qubit noise model with a probability (or strength) parameter $p_{\text{two}} = 0.5 \times 10^{-2}$.

To model gate infidelities in various $\texttt{IBM}$ devices, we employed the following parameters. For $\texttt{IBM Mumbai}$ device (see Table 5), the median 1-qubit gate infidelity was represented by $p_{\text{one}}^{\text{median}} = 2.44 \times 10^{-4}$, and the median 2-qubit gate infidelity was represented by $p_{\text{two}}^{\text{median}} = 8.25 \times 10^{-3}$. Additionally, the maximum 1-qubit gate infidelity was denoted as $p_{\text{one}}^{\text{max}} = 1.45 \times 10^{-3}$, and the maximum 2-qubit gate infidelity was denoted as $p_{\text{two}}^{\text{max}} = 2.30 \times 10^{-2}$. We also created a more noisy scenario by scaling these maximum infidelities by ten folds, resulting in $p_{\text{one}}^{\text{10max}} = 10 \times p_{\text{one}}^{\text{max}}$ and $p_{\text{two}}^{\text{10max}} = 10 \times p_{\text{two}}^{\text{max}}$. These parameters, representing *median*, *max*, and *$10 \times$ max* infidelity strengths, were applied as depolarizing channel strengths uniformly across all qubits and gates in their respective experiments, assuming full device connectivity. The data for the $\texttt{IBM Mumbai}$ device was collected on August 8, 2023.

In contrast, for the $\texttt{IBM Ourense}$ device, we considered individual gate infidelity information while also factoring the qubit connectivity of the device. The qubits $0 - 1 - 2 - 3$ with 1-qubit infidelities were set as follows: $p_{\text{one}}^0 = 5.22 \times 10^{-4}$, $p_{\text{one}}^1 = 4.14 \times 10^{-4}$, $p_{\text{one}}^2 = 1.84 \times 10^{-4}$, and $p_{\text{one}}^3 = 4.3 \times 10^{-4}$. For 2-qubit gates involving qubit 1 and its neighbours, the corresponding 2-qubit depolarizing channel strengths were determined as $p_{\text{two}}^{01} = 9.55 \times 10^{-3}$, $p_{\text{two}}^{12} = 9.44 \times 10^{-3}$, and $p_{\text{two}}^{13} = 1.25 \times 10^{-2}$. The device data used for $\texttt{IBM Ourense}$ was sourced from (Du et al., 2022).

### H.4 READOUT ERRORS AS BIT-FLIP CHANNEL

We incorporate readout errors into our model, inspired by (Blank et al., 2020), treating them as bit-flip (amplitude flip) operations applied just before the measurement (Nielsen & Chuang, 2010). The bit-flip channel, representing the readout error with strength $p$ on qubit $q$, is characterized by the following Kraus representation.

$$\Lambda_{p,q}^{(bf)}(\rho) = (1 - p)\,\rho + p X_q \rho X_q \tag{16}$$

In this scenario, the qubit $q$ has a probability of $1 - p$ to retain its current state, and with a probability $p$, it flips its amplitude, akin to a $\pi$ rotation around the x-axis in the Bloch sphere. Again, noise profiles of two different $\texttt{IBM}$ devices were implemented.

Firstly, we uniformly apply readout noise levels obtained from $\texttt{IBM Mumbai}$ devices, categorized as *median* and *max*, to all qubits in our experiments. The readout error strengths for $\texttt{IBM Mumbai}$ device are denoted as $p_{\text{ro}}^{\text{median}} = 2.25 \times 10^{-2}$ and $p_{\text{ro}}^{\text{max}} = 8.7 \times 10^{-2}$, acquired on August 8, 2023, as listed in Table 5. For the *$10 \times$ max* experiments, we use $87\%$ readout errors, as ten times the *max* readout errors rendered VQE experiments infeasible due to signal loss.

Secondly, we account for readout errors of qubits $0 - 1 - 2 - 3$ from the $\texttt{IBM Ourense}$ device (Blank et al., 2020). The corresponding error values are $p_{\text{ro}}^0 = 1.65 \times 10^{-2}$, $p_{\text{ro}}^1 = 2.38 \times 10^{-2}$, $p_{\text{ro}}^2 = 1.57 \times 10^{-2}$, and $p_{\text{ro}}^3 = 3.95 \times 10^{-2}$.

### H.4.1 STATE PREPARATION ERRORS AND THERMAL RELAXATION CHANNEL

The state preparation errors in transmon qubits are due to residue thermal populations that can be modelled as thermal states (Krantz et al., 2019). A transmon qubit, a quantum anharmonic oscillator in the truncated Hilbert space, can be modeled as a $k$-level quantum system (a qudit) with unequal level spacing such that the subspace of the first two levels is reserved for computation. A widely used approximate model is a qutrit (where the infinite-dimensional Hilbert space is truncated to $k = 3$) where the ground state energy is set to be zero, $E_0 = 0$, the first excited state is determined by the qubit frequency ($\omega$) parameter, $E_0 = \hbar\omega$, and the second excited state is determined by

the anharmonicity parameter ($\delta$, typically a negative number), $E_2 = \hbar \left(2\omega + \delta\right)$. Given the fridge temperature $T$, the initial quantum state of a qubit with residual thermal populations is the following.

$$\rho_0 Z = |0\rangle \langle 0| + \exp\left(-\frac{E_1}{k_B T}\right) |1\rangle \langle 1| + \exp\left(-\frac{E_2}{k_B T}\right) |2\rangle \langle 2|$$

Here $\rho_0$ denotes the initial thermal state, $k_B$ and $\hbar$ denote the Boltzmann and reduced Planck constants, and $Z$ denotes the equipartition function $Z = 1 + \exp\left(-\frac{E_1}{k_B T}\right) + \exp\left(-\frac{E_2}{k_B T}\right)$. The probability of the quantum state being initially in the excited state is given as $p_e = \exp\left(-\frac{E_1}{k_B T}\right)/Z$. In our experiments, we used $\omega_{\mathrm{mumbai}} = 4.894 \mathrm{GHz}$, $\delta_{\mathrm{mumbai}} = -0.33 \mathrm{GHz}$ for the IBM Mumbai device and we used $\omega_{\mathrm{ourense}} = 5 \mathrm{GHz}$, $\delta_{\mathrm{ourense}} = -0.33 \mathrm{GHz}$ for the IBM Ourense device to acquire these parameters.

Each time qubits are reset, the decoherence, which is caused by coupling of the quantum system to external degrees of freedom such as stray fields, allows computations to run for a certain amount of time (coherence time) before the strictly quantum mechanical properties are lost. The thermal relaxation time, $T_1$, measures how long it takes for a qubit that is initially prepared in the excited state $|1\rangle$ (the North pole of the Bloch sphere) to decay back to the ground state $|0\rangle$ (the South pole of the Bloch sphere) (Krantz et al., 2019). The phase-coherence time, $T_2$, measures how long it takes to lose the phase information. When a qubit is initially prepared on the equator of the Bloch sphere, such as a $|+\rangle$ state, it can not be distinguished from other states on the equator, such as a $|-\rangle$ state, due to being end up in a classical mixture of these states after $T_2$. These thermal relaxation and decoherence processes do not happen instantly after $T_1$ and $T_2$ amount of time, but the qubits experience them rather gradually as 1-, and 2-qubit gate implementations also take a finite amount of time, $\Delta t_{\mathrm{one}}$ and $\Delta t_{\mathrm{two}}$, hindering the perfect implementation of these gates beside the gate infidelities. These gradual processes can be modeled as quantum channels that follow the application of 1-, and 2-qubit gates (Blank et al., 2020). The effect of the thermal relaxation channels depends on how good the initial state is prepared or the excited state probability, $p_e$, decoherence times $T_1$ and $T_2$ of the quantum hardware in hand, and how fast the gates are implemented $\Delta t_{\mathrm{one}}$ and $\Delta t_{\mathrm{two}}$ (for values see Table 5). For these gate durations, we can define thermal relaxation and dephasing gate error rates $\epsilon_{T_1} = \exp\left\{-\frac{\Delta t}{T_1}\right\}$ and $\epsilon_{T_2} = \exp\left\{-\frac{\Delta t}{T_2}\right\}$. Using these, we can define the qubit reset probability $p_{\mathrm{reset}} = 1 - \epsilon_{T_1}$, and the following probabilities.

$$
\begin{aligned}
p_{\mathrm{id}} &= 1 - p_z - p_{r_0} - p_{r_1} \\
p_z &= \left(1 - p_{\mathrm{reset}}\right)\left(1 - \epsilon_{T_1}\epsilon_{T_2}^{-1}\right)/2 \\
p_{r_0} &= \left(1 - p_e\right)p_{\mathrm{reset}} \\
p_{r_1} &= p_e p_{\mathrm{reset}}
\end{aligned}
\tag{17}
$$

Here $p_{r_0}$ gives the probability that qubit resets to the ground state $|0\rangle$, $p_{r_1}$ gives the probability qubit resets to the excited state $|1\rangle$, $p_z$ gives the probability that qubit in the ground state is hit by a phase-flip operation $Z$, and $p_{\mathrm{id}}$ gives the probability that qubit in the ground state is hit by an identity gate $I$. The regimes where $T_2 \leq T_1$ and $T_2 > T_1$ have different channel representations (Blank et al., 2020). The Kraus representation for the thermal-relaxation channel (TRC) acting on the qubit $q$ when $T_2 \leq T_1$ follows the probabilities described above.

$$
\begin{array}{ll}
K_0 = \sqrt{p_{\mathrm{id}}}\, I & K_1 = \sqrt{p_z}\, Z \\[2mm]
K_2 = \sqrt{\dfrac{p_{r_0}}{2}}\,\dfrac{I + Z}{2} & K_3 = \sqrt{\dfrac{p_{r_0}}{2}}\,\dfrac{X - iY}{2} \\[2mm]
K_4 = \sqrt{\dfrac{p_{r_1}}{2}}\,\dfrac{X + iY}{2} & K_5 = \sqrt{\dfrac{p_{r_1}}{2}}\,\dfrac{I - Z}{2}
\end{array}
$$

In this regime, TRC has amplitude and phase-damping terms. Here $K_2$ ($K_5$) operator resets the qubit to $|0\rangle$ ($|1\rangle$) with probability $p_{r_0}$ ($p_{r_1}$). Similarly $K_3$ ($K_4$) operator relaxes (excites) the qubit into $|0\rangle$ ($|1\rangle$) with probability $p_{r_0}$ ($p_{r_1}$). While $K_0$ and $K_1$ implement $I$ and $Z$ with their respective probabilities. In the $T_2 > T_1$ regime, the quantum channel is represented by the following Choi

matrix (Blank et al., 2020).

$$
\Lambda = \begin{bmatrix} 1 - p_{r_1} & 0 & 0 & \epsilon_{T_2} \\ 0 & p_{r_1} & 0 & 0 \\ 0 & 0 & p_{r_0} & 0 \\ \epsilon_{T_2} & 0 & 0 & 1 - p_{r_0} \end{bmatrix} \tag{18}
$$

The eigendecomposition and "unvec" operations yield Kraus matrices $K_\lambda = \sqrt{\lambda} \Phi(v_\lambda)$. For the $T_2 > T_1$ regime, we use the analytically computed Kraus matrices given in Pennylane software library (Bergholm et al., 2018). To compute these, the $T_1$ and $T_2$ times of `IBM Mumbai` and `IBM Ourense` devices we use are the following. The ones for Mumbai devices are the median values of the data acquisition date $T_1 = 122.28\mu s$, $T_2 = 167.2\mu s$, (see Table 5). And, the ones for Ourense devices are, $T_1^0 = 75.75\mu s$, $T_2^0 = 50.81\mu s$, $T_1^1 = 78.47\mu s$, $T_2^1 = 27.56\mu s$, $T_1^2 = 101.51\mu s$, $T_2^2 = 107\mu s$, and $T_1^3 = 79.54\mu s$, $T_2^3 = 78.38\mu s$ (see (Du et al., 2022)).

## I    SOFTWARE LEVEL SIMULATION OPTIMIZATION

Our RL agent training involves billions of queries to the classical simulator, incurring substantial time and computational costs. In each episode, unless terminated by the random halting protocol or by reaching a threshold value before the final action, the RL agent executes a series of actions $n_{\mathrm{act}}$ (an integer). During each action $i$, where $i \in [n_{\mathrm{act}}]$, the agent appends a new 1-, or 2-qubit gate to a quantum circuit. At the simulator level, this corresponds to compiling $n_g^{(i)} = n_{oqg}^{(i)} + n_{tqg}^{(i)}$ gates applied by the agent until action $i$. Here, $n_g^{(i)}$ represents the total number of gates implemented during cost function evaluation (subsequent matrix-vector multiplications) at action $i$, such that at the final action $i = n_{\mathrm{act}}$, a total of $n_{\mathrm{act}} = n_g^{(n_{\mathrm{act}})}$ gates are applied. Additionally, $n_{oqg}^{(i)}$ and $n_{tqg}^{(i)}$ denote the number of 1-, and 2-qubit gates compiled during action $i$. The cost function evaluation takes $t_{\mathrm{feval}}^{(i)} \approx t_{\mathrm{feval}}$ amount of time for each action $i$. The value of $t_{\mathrm{feval}}^{(i)}$ can be approximated as an intermediate value of $t_{\mathrm{feval}}$ even though the function evaluation time cumulatively increases with $i$. Furthermore, at each action $i$, classical optimizers require $n_{\mathrm{feval}}$ cost function evaluations. Given a total of $n_e$ episodes, training an RL agent roughly consumes $n_e \times n_{\mathrm{act}} \times n_{\mathrm{feval}} \times t_{\mathrm{feval}}$ time, with an additional 20 seconds per episode for neural network training on an NVIDIA-A100 GPU.

As of the completion of this work, Qulacs remains the most efficient classical simulation framework for Python-based quantum circuit evaluations. This efficiency is derived from effective memory-access, swift linear algebra facilitated by C++ SIMD optimizations, and an underlying algorithm that handles horizontal and vertical gate fusions for named gates in noiseless scenarios (Suzuki et al., 2021). However, the advantage of using Qulacs in our experiments diminishes when transitioning to noisy simulations using the Kraus operator sum formalism, as the algorithmic benefits related to gate fusions are not as prominent in this context. The total number of dense Kraus matrix multiplications is defined as $n_{km} = 2\left(n_{oqg}n_{oqc}n_{ok} + 2n_{tqg}n_{tqc}n_{tk}\right)$. Here, $n_{oqc}$ represents the number of 1-qubit channels following 1-qubit gates, and $n_{ok}$ denotes the number of Kraus matrices in these 1-qubit channels. Similarly, $n_{tqc}$ indicates the number of 2-qubit channels following 2-qubit gates, and $n_{tk}$ specifies the number of Kraus matrices in these 2-qubit channels. It is important to note that the term for 2-qubit channels is multiplied by two, as different noise channels need to be applied at each qubit during an evolution. Additionally, the overall number is multiplied by two because the density matrix must be affected both from the left and the right by Kraus matrices. For mathematical implementations of these operations, please refer to Appendix H.

Crucially, these noise simulations are performed online (during simulation) each time. As the number of qubits and gates increases, the cost function evaluation times ($t_{\mathrm{feval}}$) grow exponentially. This results in RL training times escalating from the scale of days in the noiseless case to months in the noisy case, rendering it impractical to obtain prompt feedback from the training, even for basic unit tests and hyperparameter optimization. To address these challenges, we precomputed the Pauli-transfer matrices (PTMs) of the noisy gates before simulations and developed a GPU-based simulation framework in JAX (Python) (Bradbury et al., 2018). Due to time constraints, we were unable to implement the backward or forward propagation algorithms for sparse Paulis as prescribed in (Rall

et al., 2019; Fontana et al., 2023). However, we successfully implemented fast, dense matrix-vector multiplication-based algorithms for both noiseless and noisy cases.

To guide the simulator on the gates to be implemented, we provide it with a NumPy array (Harris et al., 2020) of dimension $n_g \times 2$. The first column enumerates the subsequent gates or PTMs in ascending order, with the first element corresponding to the first gate, and so forth. The second column contains the variational parameters to be implemented, with parameters for non-parametric gates set to zero. We also provide the positions (row numbers) of this instructions array to facilitate updating parameters in the correct order outside closed-circuit GPU computations occurring in XLA (accelerated linear algebra library) (Sabne, 2020). A just-in-time (JIT) compiled JAX function reads these instructions (in XLA) and, for each row (or instruction), generates a unitary matrix or PTM using the gate number and angle. Subsequently, another JIT-compiled function takes a vector from the previous iteration and the matrix constructed by the previous function, returning a new vector after multiplication. One level higher, another JIT-compiled function iterates through these matrix-vector multiplications (forward propagation) row by row using JAX's native *foriloop*, carrying the vector from the previous iteration until the end of the circuit. Finally, another JIT-compiled function takes over this vector and either performs an inner product operation $\psi^\dagger H \psi$ for dense Hamiltonians in noiseless simulations or a dot product $\langle H | \rho \rangle$ for Hamiltonian vectors in noisy cases. All these functions are merged into a single JIT-compiled expectation value computation subroutine. When the user queries this subroutine with a new set of angles, it first updates the variational parameters of the instructions array, then feeds this array to the JIT-compiled subroutine (running on GPU using XLA instructions) to obtain a real number output. Due to the asynchronous nature of GPU computing and the current limitations of XLA, which require static tensor geometries as input and outputs, these computational subroutines need to run concurrently with static input-output shapes to leverage computational benefits. Our implementation takes these aspects into account. These subroutines could be further integrated with sparse Pauli algorithms (Rall et al., 2019; Fontana et al., 2023) or tensor network methods (Orús, 2014) to extend the scalability of these noiseless and noisy simulations in terms of qubit numbers and gate counts. However, we leave this avenue for exploration in future work.

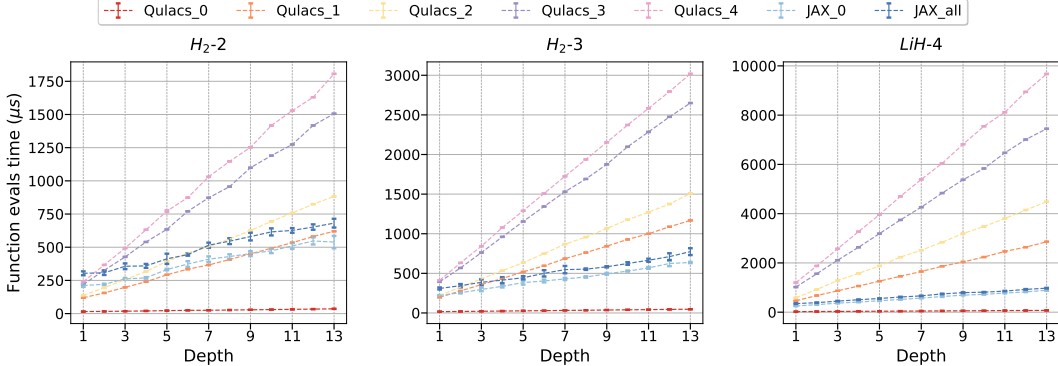

Figure 8: **Computation time per VQE function evaluation in hardware-efficient ansatz (HEA) using differing software and methods.** The x-axis shows the HEA depth for the given problem. Different traces display the implementations in different software and different noise models.

In Fig. 8, we evaluate the performance of different simulators implementing various noise models as the hardware-efficient ansatz depths increase (i.e., with growing gate counts) for different qubit system sizes. Noise models are identified by cumulative numbers that increase with the addition of noise channels after their respective gates. Initially, noise model "0" represents the noiseless simulator. Subsequently, noise model "1" introduces a 1-qubit depolarizing channel after each 1-qubit gate. Building on this, noise model "2" adds a depolarizing channel to each "ctrl" and "target" qubit of the CNOT gate after post-implementation. Further extensions include noise model "3" introducing a 1-qubit thermal relaxation channel after each 1-qubit gate, and noise model "4" which, on top of noise model "3" adds a thermal relaxation channel to each "ctrl" and "target" qubit of the CNOT gate. While Qulacs exhibits optimal evaluation times in the noiseless case for all qubit numbers, the advantage diminishes with the inclusion of any noise channel, especially as gate count and the number of dense matrix multiplications increase due to Kraus operator sum formalism (see

Appendix H). This trend becomes more pronounced with the addition of noise channels following gate implementations (i.e., an increase in the noise model number), leading to an exponential increase in the number of Kraus operations. In terms of quantitative results, for depth 13, JAX-PTM yields an improvement of up to a 2.65-, 3.9-, and 9.93-fold over Qulacs_4 noise model in 2-, 3-, and 4-qubit problems, respectively. We acquired these statistics from 20000 independent runs.

In Fig. 9, we show the RL episode times concerning different noise models and software, without implementing any random halting or early stopping of episodes, allowing each to proceed up to a gate count of 40. We ran the RL training for each of these scenarios for 200 episodes once. The neural network training takes approximately 20 seconds per episode. Again, in terms of quantitative results, we observe an improvement of up to a 1.34-, 1.59-, and 3.85-fold over Qulacs_4 noise model in 2-, 3-, and 4-qubit problems, respectively. However, excluding the neural network training time, the improvement was more pronounced. We observed a 2-, 2.9-, and 5.74-fold speed-up for the above problems, respectively.

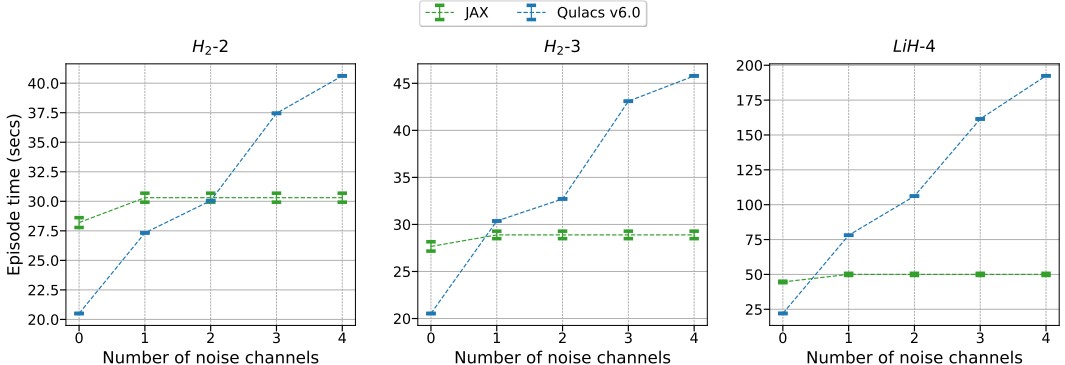

Figure 9: **Computation time per episode for RL agent using differing software and methods.** The x-axis shows the number of noise channels added after their respective gate, and $x = 0$ corresponds to noiseless simulations.

In conclusion, the use of PTMs enabled us to transform the $N$-qubit noisy simulation problem, regardless of the number of noise channels, into the $2N$-qubit noiseless simulation problem involving dense matrix-vector multiplications. Utilizing JIT-compilation and GPU computation capabilities in JAX, we achieved up to six-fold improvement in RL training while simulating noisy circuits. This enhancement allowed us to successfully train RL agents for QAS problem under the effect of quantum device noise.

## J   BEHAVIOURAL DIFFERENCE OF REAL DEVICE & CLASSICAL NOISY SIMULATION

Recent studies suggest that despite device drift, low-level hardware noise in quantum machine learning tasks yields outcomes where real hardware and simulations demonstrate comparable behavior (Rehm et al., 2023). Our comparative experiment on `IBM Lagos` device and the classical simulation of its noise profile[3] aligns with these findings in our context as well. Using the circuit generated by CRLQAS method for a 4-qubit $H_2$ molecule (see Fig. 10), we validated our hypothesis by comparing measured expectation values on the device to the classical noise model used in simulations of CRLQAS. Fig. 11 showcases that the expectation values of Pauli strings align within one-sigma standard deviation error bars. The expectation values were computed individually for each Pauli string which allowed for fine-grained analysis. It is noteworthy that the full Hamiltonian expectation values also exhibit similar trends, with a measurement of $-0.0615 \pm 0.5295$ on `IBM Lagos` and $-0.0335 \pm 0.5294$ on the noisy simulator. Our experiments were executed using the `Qiskit Runtime IBM Client` API.

---

[3]We performed the classical simulation of the `IBM Lagos` noise profile based on the calibration data of November 16, 2023.

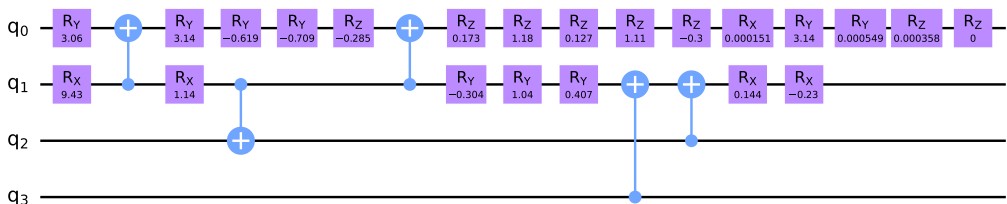

Figure 10: **The parametrized circuit generated by the CRLQAS (RH, wo-IA) method for the** $H_2 - 4$ **molecule.** 'RH' indicates the utilization of the random halting scheme, while 'wo-IA' signifies the absence of illegal actions.

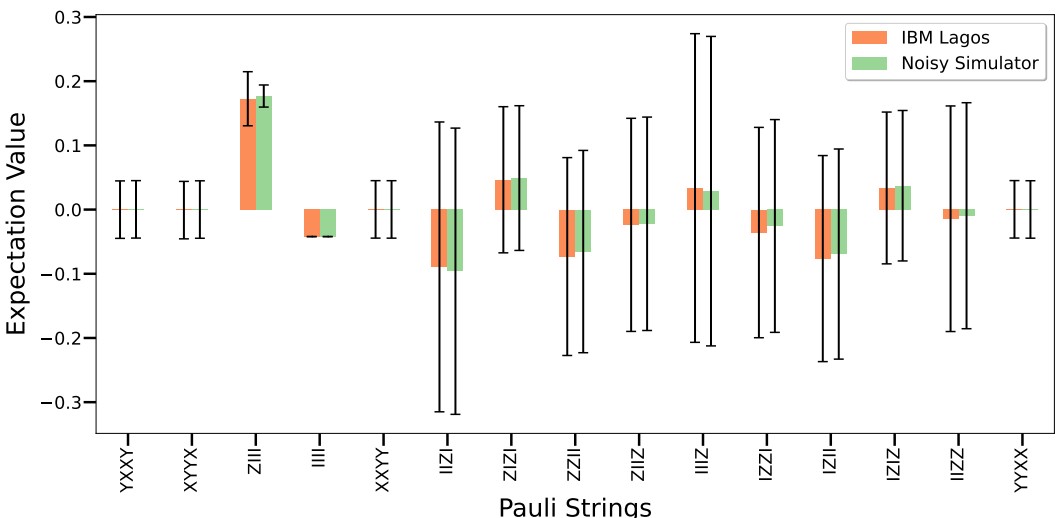

Figure 11: **Comparative experiment of real hardware to classical noisy simulator.** The bars in the plot represent the expectation values of each Pauli string for $H_2 - 4$ with the circuit depicted in Fig. 10. The colors represent the results for the `IBM Lagos` (orange) and noisy simulator (green) introduced in this work.

## K    COMPARISON TO OTHER QAS METHODS

We compare our method CRLQAS to a modified variant of qubit-ADAPT VQE (Tang et al., 2021), quantumDARTS (Wu et al., 2023) and RLQAS (Ostaszewski et al., 2021b) in noiseless experiments (see Table 1 and Table 3). For noisy experiments, we compare to QCAS (Du et al., 2022) (see Table 2). While there is a substantial amount of literature on quantum architecture search (QAS), replicating the performance of some methods is challenging due to the lack of unified experimental settings. Moreover, the purpose of this work was to establish the limits of RL methods in combination with noisy intermediate-scale quantum hardware. Nonetheless, we provide an overview of reported results and corresponding settings of the ground state estimation task for some QAS approaches.

In the QAS literature, we identified two works that include the ground state energy estimation problem in their experiments. (Rattew et al., 2019) presents numerical results for their EA-based QAS method within chemical accuracy, while (Wang et al., 2022) evaluates their one-shot super-circuit method under simulated circuit noise, significantly impacting estimation accuracy

(Rattew et al., 2019) proposed Evolutionary Variational Quantum Eigensolver (EVQE) that uses evolutionary programming techniques to dynamically generate and optimize quantum circuits (by adding or removing gates). The gate set in this algorithm is composed of 1-qubit universal gates and 2-qubit controlled universal gates, which after the optimization, are decomposed into `CNOT` gates and 1-qubit universal rotation gates. The noiseless experiments involved two molecules, a 6-qubit LiH with varying interatomic bond distances and a 7-qubit $BeH_2$ with a bond distance of 1.3Å. In

the noiseless case, examining Fig. 3 in (Rattew et al., 2019), the best-observed depth and `CNOT` gate count were approximately $35$ and $40$, respectively (exact numbers not explicitly stated in the paper). During EVQE training, the achieved error was slightly less than $1 \times 10^{-3}$ (no exact number provided). On the other hand, CRLQAS (IA, wo-RH) in median statistics achieves $8 \times 10^{-4}$ error with depth and `CNOT` gate count of $27$ and $30$, respectively (see Table 3). For the noisy experiment (where noise profiles were simulated using the Qiskit Aer QASM simulator) with a 4-qubit LiH at a bond distance of 2.2Å, referencing Fig. 7 in (Rattew et al., 2019), the averaged noisy energy error, depth, and number of `CNOT` gates over five seeds were approximately 0.036Ha, 8, and 6, respectively. In our case, we achieved a slightly improved noisy error of 0.0356Ha with an average of 1.33 `CNOT` gates and a depth of 10 across three seeds. Although the mean noisy errors and depth are comparable, CRLQAS produces a quantum circuit with fewer `CNOT` gates.

(Wang et al., 2022) introduced the TorchQuantum framework based on PyTorch (Paszke et al., 2019) and develops the QuantumNAS algorithm, a one-shot super-circuit method designed for finding noise-adaptive quantum circuits in VQA tasks. This technique involves sharing gate parameters among sub-circuits after one shot training to accelerate circuit evaluation. However, optimizing the super-circuit poses a challenge, and furthermore, there is a weak correlation in performances between sub-circuits with inherited parameters and those with optimal parameters from individual training. Our attempt to compare with (Wang et al., 2022) was hindered by the lack of essential information for a comprehensive analysis. More specifically, details about molecular geometries and quantum circuits prevented us from conducting a quantitative experiment for comparison, as it would require running their algorithm from scratch for the molecules considered in our study. Given that we already compared to the QCAS (Du et al., 2022) algorithm (see Table 2), which is similar to QuantumNAS, and considering the aforementioned reasons, we did not include this work in our comparative experiments. However, a qualitative comparison was possible for the $H_2$ molecule in a 2-qubit system. Examining Fig. 17, the best result achieved in their experiments had an energy error of 0.1Ha. In our $H_2$ simulations, we achieved a mean noisy energy error over three independent seeds of $1.116 \times 10^{-3} \pm 3.41 \times 10^{-4}$ (for `IBM Mumbai` median), which is three orders of magnitude better than their results. Additionally, we outperformed them for `IBM Mumbai` max and 10 times max noise levels, with mean noisy energy errors of $0.907 \pm 0.008$ and $0.0912 \pm 0.004$, respectively. Unfortunately, the manuscript does not provide information about the number of gates, depth, or parameters achieved for this experiment.

## L  LIST OF MOLECULES

Given the geometrical description of the molecule, and using the Born-Oppenheimer approximation, we fix a finite basis set, in our case STO-3G, to discretize the system. Subsequently, we utilize the OPENFERMION open-source electronic structure package (McClean et al., 2020), with PSI4 serving as the backend computational chemistry software, to generate the molecular Hamiltonians. We describe the essential details to generate the Hamiltonians in Table 6 below.

Table 6: List of molecules considered in our simulations. The coordinates of the configuration are given in angstrom.

| Molecule | Fermion to qubit mapping | Configuration | Number of qubits |
|---|---|---|---|
| $H_2$ | Jordan-Wigner (Jordan & Wigner, 1993) | H $(0, 0, 0)$; H $(0, 0, 0.7414)$ | 2 |
| | Jordan-Wigner | H $(0, 0, 0)$; H $(0, 0, 0.7414)$ | 3 |
| | Jordan-Wigner | H $(0, 0, -0.35)$; H $(0, 0, 0.35)$ | 4 |
| LiH | Parity (Seeley et al., 2012) | Li $(0, 0, 0)$; H $(0, 0, 3.4)$ | 4 |
| | Jordan-Wigner | Li $(0, 0, 0)$; H $(0, 0, 2.2)$ | 6 |
| $H_2O$ | Jordan-Wigner | H $(-0.021, -0.002, 0)$; O $(0.835, 0.452, 0)$; H $(1.477, -0.273, 0)$ | 8 |

