# OpenReview forum: "Curriculum reinforcement learning for quantum architecture search under hardware errors"
_ICLR.cc/2024/Conference — ICLR 2024 poster_

### Official Review · Reviewer_3EHX · 2023-10-22

**Soundness:** 3 good
**Presentation:** 3 good
**Contribution:** 2 fair
**Rating:** 5
**Confidence:** 3

**Summary:**

This paper introduces a novel quantum architecture search (QAS) algorithm based on curriculum reinforcement learning (CRL) for designing parameterized quantum circuits (PQCs) under realistic noisy environments. The main contributions include a tensor-based binary encoding scheme, a mechanism of illegal actions to prune the search space and avoid redundant gates, and a random halting technique to encourage the agent to find shorter circuits. Additionally, it presents a variant of simultaneous perturbation stochastic approximation (SPSA) algorithm with adaptive momentum and variable sample budget for faster and robust optimization, and a fast GPU simulation framework using Pauli transfer matrix formalism to fuse gates with their noise models. The effectiveness of CRLQAS is demonstrated on quantum chemistry tasks of finding the ground state energy of various molecules, such as H2, LiH, and H2O, in both noiseless and noisy settings.

**Strengths:**

- Novel QAS algorithm: The paper proposes a curriculum-based reinforcement learning QAS (CRLQAS) algorithm that can automatically construct parametrized quantum circuits for variational quantum algorithms in realistic noisy environments. The algorithm introduces several novel features, such as tensor-based binary circuit encoding, illegal actions, random halting, feedback-driven curriculum learning, and Adam-SPSA optimizer.
- Experiments on real quantum devices: The paper demonstrates the efficiency and robustness of CRLQAS under different noise models inspired by real IBM quantum devices, and provides comprehensive descriptions of the experimental setup and the source code for reproducibility.

**Weaknesses:**

1. (Major) The absence of the comparison of some key methods in terms of theory and experiments. This paper emphasizes the design for quantum hardware errors, but there is no mention or comparison of some methods of QNAS that are also designed for noise, such as QuantumNAS (a noise-adaptive quantum circuit search) [1]. In addition, as far as I know, QCAS (Du et al., 2022) is tested on depolarizing errors, and QuantumDARTS (Wu et al., 2023) is evaluated readout errors. However, these methods under noise are not compared in the experiments.

[1] Wang, H., Ding, Y., Gu, J., Lin, Y., Pan, D. Z., Chong, F. T., & Han, S. QuantumNAS: Noise-adaptive search for robust quantum circuits. In 2022 IEEE International Symposium on High-Performance Computer Architecture (HPCA).


2. (Minor) The representations of some tables are not clear enough. For example, if the best result is bolded, then ‘7.21*10^-8’ and ‘2.9*10^-4’ should be bolded in Table 1. In Table 4, CRLQAS (RH) has two rows of results but they're not marked what do they mean.

**Questions:**

1. As mentioned in the Intro, current NISQ devices are characterized by limited qubit connectivity and susceptibility to noise, but how does the proposed approach deal with qubit connectivity?
2. The proposed method is based on reinforcement learning, but only one related method is mentioned in this paper, and there are several QNSA works based on reinforcement learning that are not mentioned, such as [2,3,4]. Furthermore, compared with these methods, what is the innovation of CRLQAS, and whether it can reflect the advantage of the experiment?

[2] Kuo, E. J., Fang, Y. L. L., & Chen, S. Y. C. Quantum architecture search via deep reinforcement learning. arXiv preprint arXiv:2104.07715, 2021.

[3] Chen, S. Y. C. (2023, August). Quantum Reinforcement Learning for Quantum Architecture Search. In Proceedings of the International Workshop on Quantum Classical Cooperative (pp. 17-20), 2023.

[4] Sun, Y., Ma, Y., & Tresp, V. Differentiable quantum architecture search for quantum reinforcement learning. arXiv preprint arXiv:2309.10392, 2023.

3. In Table 4, why is CRLQAS(wo-RH) better than CRLQAS? What is the meaning of random halting (RH)?

4. What are the advantages and disadvantages of the proposed method and the Predictor-based QAS methods [5,6]?

[5] Zhimin He, Xuefen Zhang, Chuangtao Chen, Zhiming Huang, Yan Zhou, and Haozhen Situ. A gnn-based predictor for quantum architecture search. Quantum Information Processing, 22(2): 128, 2023b.

[6] Shi-Xin Zhang, Chang-Yu Hsieh, Shengyu Zhang, and Hong Yao. Neural predictor based quantum architecture search. Machine Learning: Science and Technology, 2(4):045027, 2021

---

> ### Author Response · Authors · 2023-11-19
> **Response 1 to reviewer 3EHX**
>
> We would like to thank the referee for the time spent reviewing this submission and recognizing the novelty of our CRLQAS algorithm and the introduced features within.
> We're also grateful for the referee's note regarding our utilization of GPU-accelerated Pauli transfer matrix formalism and introduction of Adam-SPSA for faster convergence.
> During the rebuttal period, we performed additional ablation studies and improved the readability of the manuscript. In the general answer, we list down the introduced changes. We hope it is sufficient, please let us know if you have any more questions or comments.
>
> > Absence of the comparison of some key methods in terms of theory and experiments. Such as QuantumNAS (a noise-adaptive quantum circuit search) [1]. In addition, as far as I know, QCAS (Du et al., 2022) is tested on depolarizing errors, and quantumDARTS (Wu et al., 2023) is evaluated readout errors. However, these methods under noise are not compared in the experiments.
>
> It's possible that due to the text structure, it was missed, but we did indeed consider these other works (or similar), and reported on it in the paper.
> For instance, we did compare our CRLQAS algorithm to QCAS (Du et al., 2022) under hardware constraints (both noise and qubit-connectivity) in Table 2 of the revised manuscript (Table 4 in the old one).
> We did not highlight it enough to the reader in the first version which is rectified now.
> We do not compare to QuantumNAS (Wang et al., 2022) as it is quite similar to QCAS algorithm (both are one-shot super-circuit methods).
> We refer the referee to our *Response 3 to reviewer WndL* for a more detailed reasoning.
>
> We respectfully disagree with the assessment of the reviewer that quantumDARTS (Wu et al., 2023) is evaluated with readout errors for quantum chemistry tasks.
> It only considers readout errors for the MNIST classification task which is not the focus of this work.
> Consequently, we only compared to quantumDARTS in the noiseless case and numerically demonstrate that our CRLQAS performs better.
> We believe that our comparison to existing methods (like QCAS, qubit-ADAPT VQE, quantumDARTS) is fair, and covers most of the existing approaches for quantum chemistry problems in both noisy and noiseless cases.
>
> > The representations of some tables are not clear enough.
>
> We appreciate the feedback on the inconsistencies within the Tables.
> We have amended such mistakes, and hope that they improve the readability of the manuscript.
> We note that the Table 4 from the older version is now Table 2 in the new version of the manuscript.
>
> > In Table 4, why is CRLQAS(wo-RH) better than CRLQAS? What is the meaning of random halting (RH)?
>
> Random Halting is one of the new techniques that we proposed and described in sec. 3.3 of the manuscript.
> By default, in CRLQAS, we utilize all three novel features namely illegal actions, tensor-based encoding and random halting.
> We denote wo-RH and RH as experiment configurations where the random halting feature is turned off and on respectively during CRLQAS training.
> As the *reviewer wt33* rightly pointed out the role of illegal actions might be pivotal within the CRLQAS algorithm.
> Consequently, we conducted an ablation study to assess the effectiveness of the new features introduced, both in noiseless and noisy scenarios. Detailed results are available in *Table 1 of Response 3 to reviewer wt33*.
> Our analysis reveals that incorporating illegal actions without random halting prompts the agent to achieve a positive signal (a successful episode) early on, albeit resulting in larger circuits.
> Conversely, introducing random halting encourages the agent to discover shorter circuits, there is a trade-off as the agent receives the positive signal at a later stage.
> We further clarify this in Table 2 in the revised manuscript (which was Table 4 in the old version of the manuscript).
>
> > As mentioned in the Intro, current NISQ devices are characterized by limited qubit connectivity and susceptibility to noise, but how does the proposed approach deal with qubit connectivity?
>
> In CRLQAS, we deal with the quantum device's hardware topology by directly limiting the agent's action space to adhere to qubit-connectivity.
> Indeed, it is also conceivable to train the agent by designing the reward function that penalizes violations of connectivity constraints.
> In our study, we opt for the former method, integrating the hardware topology of the now deprecated IBM Ourense device.
> We kindly ask the referee to refer to Table $4$ in the manuscript, illustrating an experiment conducted for the $4$-qubit $H_2$ molecule, showcasing our superior performance compared to QCAS (Du et al., 2022).
>
> [1] Wang, H., Ding, Y., Gu, J., Lin, Y., Pan, D. Z., Chong, F. T., & Han, S. QuantumNAS: Noise-adaptive search for robust quantum circuits. In 2022 IEEE International Symposium on High-Performance Computer Architecture (HPCA)

---

> > ### Author Response · Authors · 2023-11-19
> > **Response 2 to reviewer 3EHX**
> >
> > > The proposed method is based on reinforcement learning, but only one related method is mentioned in this paper, and there are several QNSA works based on reinforcement learning that are not mentioned, such as [2,3,4]. Furthermore, compared with these methods, what is the innovation of CRLQAS, and whether it can reflect the advantage of the experiment?
> >
> >
> > We would like to thank the reviewer for suggesting the papers.
> > Regrettably, we were unaware of [3,4], as they were published within a month of the ICLR paper submission deadline.
> > We respectfully disagree with the reviewer that [4] is a RL based QAS algorithm--it actually proposes a differentiable QAS approach, not geared towards RL for QAS but rather quantum reinforcement learning (not VQE).
> > Perhaps the reviewer wanted us to compare to a differentiable QAS method.
> > Indeed, we do compare to quantumDARTS (Wu et al., 2023), a state-of-the-art differentiable QAS algorithm and in fact often beat them for the molecules considered in this work across a spectrum of metrics.
> >
> > Additionally, references [2, 3] concentrate on addressing the state preparation problem specifically for $2$- and $3$-qubit GHZ states ([2] considers hardware noise) whereas our focus lies in addressing the fundamentally distinct challenge of ground state energy estimation for quantum chemistry tasks. Please refer to our *Response 2 to reviewer K92h* as well. We will add the relevant references in Related Works section in the updated manuscript.
> >
> >
> > > What are the advantages and disadvantages of the proposed method and the Predictor-based QAS methods [5,6]?
> >
> > Predictor-based QAS methods such as in [5,6] aim to predict circuit performance using a neural network. However, effectively training this neural network demands a comprehensive training set comprising diverse circuits and their actual performances. To guarantee accurate evaluations, gathering an ample amount of training samples becomes crucial, especially when navigating the combinatorially large search space of quantum circuits. Consequently, the data collection process still entails significant computational expenses.
> >
> > Furthermore, both [5,6] tackle the QAS problem in the noiseless scenario, specifically addressing Ising problems.
> > They meticulously select a gate set, including YY-even, YY-odd, ZZ-odd, ZZ-even (illustrated in fig. 4 of reference [5] and fig. 10 of reference [6]), drawing inspiration from the gate set utilized in the quantum alternating optimization algorithm (QAOA).
> > The comparison to our method is hence difficult as we allow for a much harder problem, with a set of gates that is native to quantum hardware, and not to the problem.
> >
> >
> >
> >
> >
> >
> >
> > [2] Kuo, et al., (2021)  Quantum architecture search via deep reinforcement learning.  arXiv:2104.07715.
> >
> > [3] Chen, S. Y. C. (2023, August). Quantum Reinforcement Learning for Quantum Architecture Search.
> >
> > [4] Sun, et al., (2023) Differentiable quantum architecture search for quantum reinforcement learning. arXiv:2309.10392
> >
> > [5] He et al., A gnn-based predictor for quantum architecture search. QIP 2023
> >
> > [6] Zhang et al., Neural predictor based quantum architecture search. MLST 2021

---

> > > ### Comment · Reviewer_3EHX · 2023-11-23
> > >
> > > I appreciate the author's detailed response, which addresses most of my concerns. A potential issue: The "first N+3" and "next three" in sentence "Each circuit moment is represented by a ((N + 3) × N) dimensional matrix, where the first N + 3 rows indicate the positions of control and target qubits for CNOT gates implemented within that moment. The next three rows represent the position of single-qubit rotation gates, RX, RY, RZ respectively" seem to conflict with the total dimension. I think a toy example of the  3D grid encoding of a small circuit may help readers comprehend it more easily.

---

> > > > ### Author Response · Authors · 2023-11-23
> > > > **Suggested Revisions**
> > > >
> > > > We thank reviewer 3EHX for bringing to light the potential confusion in our previous description of the tensor encoding of circuits. In response to this valuable feedback, we have rephrased the paragraph to enhance clarity and readability.
> > > >
> > > > Additionally, as recommended by the reviewer, we have included an illustrative example of a quantum circuit represented through this tensor encoding in Appendix B.3. This addition aims to provide readers with a more intuitive understanding. Please note that all these modifications are clearly highlighted in green within the revised manuscript.

---

### Official Review · Reviewer_WndL · 2023-10-29

**Soundness:** 2 fair
**Presentation:** 2 fair
**Contribution:** 2 fair
**Rating:** 5
**Confidence:** 4

**Summary:**

In the era of noisy intermediate-scale quantum computing, selecting effective circuits in line with current device constraints is pivotal. This study accelerates quantum circuit simulations using the Pauli transfer matrix method and introduces a novel Curriculum-based Reinforcement Learning Quantum Architecture Search (CRLQAS) approach for optimal circuit structure determination.

**Strengths:**

1. A detailed analysis of the error rate.
2. Using the Pauli transfer matrix method to accelerate simulation and consider the noise in the training process.

**Weaknesses:**

1. The paper does not specify the function used to fit the spin orbital of the molecule. If the 'sto-3g' function is used, the claim of achieving chemical accuracy might be an overstatement. While it's not impossible, further clarification is necessary.

2. I'm concerned about the computational cost and efficiency of the reinforcement learning approach, especially when adding the noise injection method. Even with GPU acceleration, reinforcement learning is inherently expensive. Given that the noise model is derived from "current" calibration data and the gate-level noise model is only an approximation, there are concerns about its accuracy. For instance, although noise might seem stable over a short time frame, over a more extended period (e.g., a week), one can observe significant shifts. If the framework has a high computational cost, it may yield unreliable results if the calibration data changes rapidly during training.

3. Some parts of the paper could be clearer. For example, Table 4 should specify which molecule task is being discussed. There are several sections in the paper with similar clarity issues, making it harder to understand.

4. I observed that only one quantum architecture search work is considered in the comparative experiments. There are many other relevant works in the quantum architecture search for VQE, such as:

[1] Wang, Hanrui, et al. "Quantumnas: Noise-adaptive search for robust quantum circuits." 2022 IEEE International Symposium on High-Performance Computer Architecture (HPCA). IEEE, 2022.

[2] Rattew, Arthur G., et al. "A domain-agnostic, noise-resistant, hardware-efficient evolutionary variational quantum eigensolver." arXiv preprint arXiv:1910.09694 (2019).

[3] Liu, Xiaoyuan, et al. "Layer VQE: A variational approach for combinatorial optimization on noisy quantum computers." IEEE Transactions on Quantum Engineering 3 (2022): 1-20.

[4] Cheng, Jinglei, et al. "TopGen: Topology-Aware Bottom-Up Generator for Variational Quantum Circuits." arXiv preprint arXiv:2210.08190 (2022).

While there is a brief comparison with previous reinforcement learning-based architecture search papers, the fact that they don't consider noise models means it's expected for your approach to perform better. However, the increased computational cost is also a concern.

5.There's a notable absence of results on real quantum hardware. Testing only on a noisy simulator might not be entirely convincing, especially when using reinforcement learning, which is assumed to achieve the upper bound of performance. Given that you utilize a noise model from the current noisy environment and consider only one seed, real hardware testing is crucial.

**Questions:**

1. What is the function you used to fit the spin orbital of the molecule?

2. What is the computational cost for your framework? Say, based on the GPU acceleration, on what specific classical hardware, how long does it take? Please specify in detail.

3. Have you tried to implement your final result on real hardware? - I believe it should be easy for you since you already get your optimal arch and parameters from noisy simulation. You can just run this circuit on real quantum hardware and obtain the result.

---

> ### Author Response · Authors · 2023-11-19
> **Response 1 to reviewer WndL**
>
> We thank to the reviewer for appreciating the uniqueness of this work: Pauli transfer matrix noisy simulations and a curriculum-based reinforcement learning to build quantum circuits.
> In particular, performing noisy simulation is a computationally heavy task even at a small number of qubits, thus building the framework that allows users to run such simulations is key to further improve the algorithms for noisy hardware.
> In our work, we take state-of-the-art error rates from publicly available devices such that our results are as close as one can expect if they were performed on real hardware. During the rebuttal period, we performed additional ablation studies and improved the readability of the manuscript. In the general answer, we list down the introduced changes. We hope it is sufficient, please let us know if you have any more questions or comments.
>
> > The paper does not specify the function used to fit the spin-orbital of the molecule. What is the function you used to fit the spin orbital of the molecule?
>
> We are not 100\% certain we know what the reviewer means by the following sentence:
> `the function used to fit the spin-orbital'.
> In our experiments, we construct the molecular electronic structure Hamiltonians by standard methods using *OpenFermion* and *Psi4* packages.
> In short, from the geometrical description of the molecule, and using the Born-Oppenheimer approximation, we fix a finite basis set, in our case STO-3G, to discretize the system.
> Based on the subsequent remarks of the reviewer , it is likely they refer to the basis set used to discretize the system.
> We have added a more elaborated description of the molecules, and the basis set used to compute their Hamiltonians at the beginning of Section 4 Experiments in the revised manuscript.
>
> > If the 'STO-3G' function is used, the claim of achieving chemical accuracy might be an overstatement. While it's not impossible, further clarification is necessary.
>
> While we think we understand the referee's point, we respectfully disagree that talking about chemical accuracy is not appropriate, but do agree that a clarification helps, and one is added in the revision.
> In computational chemistry, chemical accuracy (CA) takes an exact value,
> \begin{equation}
>     \text{CA} = 1 \frac{\text{kcal}}{\text{mol}} = 0.0016 \text{Hartree},
> \end{equation}
> and it defines the precision such that a computational chemistry method is capable of making realistic predictions.
> Our system thus achieves chemical accuracy with respect to the Hamiltonian that defines our cost function, and this is how the term is used in literature on quantum algorithms for chemistry.
> Of course, the Hamiltonian we use is not the exact molecular second-quantized Hamiltonian, as it has been derived through numerous approximations, including the use of the STO-3G basis to describe a molecular system.
> However, this concern is about computational chemistry methods, which is far beyond the scope of this manuscript.
>
> > Some parts of the paper could be clearer. For example, Table 4 should specify which molecule task is being discussed. There are several sections in the paper with similar clarity issues, making it harder to understand.
>
> We agree with the reviewer regards to Table 4 needing more clarity. We fixed the clarity issues, and added comparisons in the revised text. Additionally, we note that the Table 4 in the older version of the manuscript is now Table 2 in the new version of the manuscript.
>
> > While there is a brief comparison with previous reinforcement learning-based architecture search papers, the fact that they don't consider noise models means it's expected for your approach to perform better. However, the increased computational cost is also a concern.
>
> We thank the reviewer for remarking that CRLQAS is stronger than previous methods, even under noisy conditions.
> It is likely that any method that aims at handling hardware noise will also suffer a significant increase in computational cost. Despite the increasing computational cost with the addition of noise, training-free approaches remain feasible. However, heretofore training-free methods suffer a huge performance drawback. As discussed above, we believe that studying systems smaller in qubit size using various noise models and without any constraints on the training (i.e., training-based) is necessary to leapfrog into developing approaches (e.g., training-free, transfer learning, etc.) where computational cost is not increasing exponentially with the qubit size.

---

> > ### Author Response · Authors · 2023-11-19
> > **Response 2 to reviewer WndL**
> >
> > >  I'm concerned about the computational cost and efficiency of the reinforcement learning approach, especially when adding the noise injection method. Even with GPU acceleration, reinforcement learning is inherently expensive.
> >
> > We agree with the reviewer that the computational cost of the method is high. We have to make a distinction between the computational cost of the reinforcement learning algorithm for the VQE problem, and the computational cost of the noise simulation backend. For the former, the computational cost of RL does not stem from the training of the underlying neural network, but from the search space of all the possible quantum circuits getting combinatorially large with an increase in the system size. Due to this both the episode lengths (number of actions in the episode) and the total number of episodes need to be increased to efficiently explore the search space.
> > In such a large space, any algorithm that is not based on human intuition [1] or domain knowledge (e.g., physics models)  has to pay this cost to explore the subspaces that can not be explored by the algorithms based on those [2, 3]. Yet the exploration of these subspaces, and their interpretation, could allow new physics-based models. For RLQAS we can claim our method is the best one so far for the task of from-scratch-circuit generation (i.e., starting from an empty circuit and adding gates one by one).
> >
> > For the latter, the computational cost of the noisy backend stems both due to the training-based (i.e., optimizing after each action) approach making too many queries to the backend and due to growing cost in qubit numbers and noise models. To tackle the first, we significantly decreased the function evaluations by introducing two variants of Adam-SPSA.
> > To tackle the second, we developed the fastest noisy quantum circuit simulator (for RL-based QAS methods) to mitigate the computational cost of the noisy backend. We discuss the details of the computational cost of the noisy backend in Appendix G (of the revised manuscript) and answer the question below.
> >
> > > What is the computational cost for your framework? Say, based on the GPU acceleration, on what specific classical hardware, how long does it take? Please specify in detail.
> >
> > As explained in the appendix G (of the revised manuscript), the computational cost stems from the numerous queries to the noisy simulation backend that implements subsequent dense matrix-vector multiplications between $4^n$ dimensional Pauli-Transfer matrices and density matrices in Pauli-Liouville representation. Here $n$ denotes the number of qubits. The training times scale linearly with the total number of function evaluations (i.e., queries to the simulator) as $n_{train} = n_{feval} t_{feval}$. Here $n_{feval}$ denotes the total number of function evaluations, and $t_{feval}$ denotes the average value of a function evaluation. The total number of function evaluations is given by $n_e \times n_a \times n_f$. Here $n_e$ denotes the number of episodes, $n_a$ denotes the number of actions per episode, and $n_f$ denotes the number of function evaluations per action (utilized in optimization).  Each evaluation takes $t_{feval}^{(i)} \approx t_{feval}$ amount of time at each action $i$, which can roughly be taken as a constant around all actions for a given qubit number. We used a NVIDIA-A100 GPU for these simulations. In that hardware, $t_{feval}$ is roughly $450\mu s$ for four-qubit circuits, $430\mu s$ for three-qubit circuits, and (counter-intuitively) $480\mu s$ for two-qubit circuits. We apply $n_a = 40$ actions (when random halting is not applied) for each number of qubits, while $n_f = 500$ for two \& three-qubit circuits and $n_f = 1675$ (or $3300$ for some experiments) for $4$-qubit circuits when Adam-SPSA is used at each action (finite-sample noise is on). Also, we use a rather lightweight neural network, so we spend $20$ secs at each episode on neural network training for each number of qubits. Combining each of these constituents roughly fits our empirical tests on RL training time that yields an average $30$ seconds per episode for two & three qubit problem, and $50$ seconds per episode for four qubit problem on NVIDIA-A100 GPU. We used Weights \& Biases (WandB) platform to streamline our experiments. The existing statistics of WandB indicate that we roughly used $100$ days of GPU compute time for our experiments. These experiments involved using multiple random seeds (typically 3 and 5 for noisy and noiseless experiment, respectively) for each RL agent, and each training corresponded to a different noise model and/or qubit number.
> >
> >
> > [1] Melnikov et al., "Active learning machine learns to create new quantum experiments." PNAS 115, no. 6 (2018)
> >
> > [2] Peruzzo et al., "A variational eigenvalue solver on a photonic quantum processor." Nature communications 5, no. 1 (2014)
> >
> > [3] Wecker et al., "Progress towards practical quantum variational algorithms." Physical Review A 92, no. 4 (2015)

---

> ### Author Response · Authors · 2023-11-19
> **Response 3 to reviewer WndL**
>
> > I observed that only one quantum architecture search work is considered in the comparative experiments. There are many other relevant works in the quantum architecture search for VQE, such as: \
> [1] Wang, Hanrui, et al. "Quantumnas: ..." 2022 IEEE HPCA. IEEE, 2022. \
> [2] Rattew, Arthur G., et al. "A domain-agnostic, ...." arXiv:1910.09694 (2019). \
> [3] Liu, Xiaoyuan, et al. "Layer VQE: ...." IEEE Transactions on Quantum Engineering 3 (2022). \
> [4] Cheng, Jinglei, et al. "TopGen...." arXiv:2210.08190 (2022).
>
> We respectfully disagree with the reviewer's statement the we only compare to one QAS method.
> We compare to qubit-ADAPT VQE (Tang et al., 2021) and quantumDARTS (Wu et al., 2023) for all the noiseless experiments and QCAS (Du et al., 2022) for noisy experiments (also with qubit-connectivity constraints).
> Although there exists a significant amount of literature about quantum architecture search (QAS),
> the purpose of this work was to establish the limits of RL methods in combination with noisy intermediate-scale quantum hardware.
> However, in view that other reviewers have pointed out similar works to compare, we are give explanations for each of the references, providing qualitative comparison wherever possible.
>
> Regarding ref. [1], our understanding of the paper suggests that QuantumNAS (Wang et al., 2022) and QCAS (Du et al., 2022) fall within the family of one-shot super-circuit methods.
> We investigated QuantumNAS manuscript in an attempt to facilitate either a fully-fledged quantitative or qualitative comparison.
> Unfortunately, the work lacks essential information for such an analysis. Specifically, details about the molecular geometries and quantum circuits are absent, preventing us from conducting any type of comparison.
> Consequently, this would require us to run their algorithm from scratch for the molecules considered in this study.
> Hence, due to the fact that we already compared to an algorithm (QCAS) which is akin to QuantumNAS and the reasons mentioned above, we did not consider this work in the comparative experiments.
> However, a qualitative comparison is possible on the $H_2$ molecule for the 2-qubit system.
> By looking at fig. 17 in ref. [1] we observe that the best ever result achieve in those experiments have an energy error 0.1 Hartree.
> Our $H_2$ simulations achieve the mean noisy energy error over three independent seeds of $1.116 \times 10^{-3} \pm 3.41 \times 10^{-4}$ (for IBM Mumbai median), which is three orders of magnitude better than their results.
> Moreover, we still beat them for IBM Mumbai max and 10max noise levels with the mean noisy energy error of $0.907 \pm 0.008$ and $0.0912 \pm 0.004$ respectively.
>
> Similarly, we can conduct a quantitative comparison in noiseless scenario, while limiting our assessment to a qualitative comparison for noisy case of $LiH$ molecule in reference to [2].
> For the noiseless case, on closer inspection of fig. 3 in ref [2], best observed depth and CNOT gate is $35$ and $40$ approximately (these exact numbers are not explicitly stated anywhere in the paper but are rough estimations).
> During the training of EVQE [2], it achieves error which is slightly less than $1 \times 10^{−3}$ (again, no exact number in the paper).
> Conversely, in the noisy case (where noise profiles were simulated using the Qiskit Aer QASM simulator), referencing fig. 7 in ref. [2], we observe that the averaged noisy energy error, depth, and number of CNOT gates over five seeds are approximately \(0.036\) Ha, \(8\), and \(6\) respectively for the $4$-qubit $LiH$. Once again, these are estimated values derived from the figure. In our scenario, we achieve a slightly improved noisy error of \(0.0356\) Ha with an average of \(1.33\) CNOT gates and a depth of \(10\) across three seeds. Although the mean noisy errors and depth are comparable, our algorithm produces a quantum circuit with fewer CNOT gates.
>
> In regards to ref. [3], after studying the work, we have realized that the method is inspired by ADAPT-VQE (Tang et al., 2021) which we have covered with noiseless simulation.
> Additionally, the method is designed to build circuits for combinatorial optimization problems as a target, which is not in the same spirit of our work.
>
> Ref. [4] explores a random selection of quantum gates, aligning with device constraints such as device topology.
> Unfortunately, comparison with ref. [4] isn't feasible since it doesn't involve any molecular problems.
> We anticipate that integrating topological constraints into the reward function of CRLQAS method, penalizing violations of connectivity constraints, could enhance transferability across quantum devices compared to ref. [4].
> The reason behind this is the fact the ref. [4] picks available gates at random, while an RL learns to generalize better, at the cost of much larger training.

---

> ### Author Response · Authors · 2023-11-19
> **Response 4 to reviewer WndL**
>
> > Given that the noise model is derived from "current" calibration data and the gate-level noise model is only an approximation, there are concerns about its accuracy. For instance, although noise might seem stable over a short time frame, over a more extended period (e.g., a week), one can observe significant shifts. If the framework has a high computational cost, it may yield unreliable results if the calibration data changes rapidly during training.
>
> We agree with the reviewer that our noise simulations do not model the dynamical shift of noise parameters due to drift, nor the unmodeled noisy phenomena such that our findings are not absolute. We highlight this fact in our manuscript. Recent literature suggests the simulated and actual hardware outcomes demonstrate comparable behaviour when confronted with low levels of hardware noise for quantum machine learning tasks [1]. In the said work, given the low levels of noise, the authors observed the difference between the real hardware and the pure simulations was statistically insignificant despite the device drift, with a notable exception of occasional sudden spikes. To this end, we will provide a comparative experiment on the real hardware and the noisy simulators in the revised manuscript in coming days. Using a single set of parameters and $20,000$ shots, we evaluated the VQE cost function of the four qubit $H_2$ problem with the circuit constructed by the CRLQAS agent on IBM-Lagos device and its noise models based on the calibration data of November $16$, 2023.
>
>
> > Notable absence of results on real quantum hardware. Testing only on a noisy simulator might not be entirely convincing, especially when using reinforcement learning, which is assumed to achieve the upper bound of performance.
>
> We agree that there may be added value of testing RL based QAS methods, and any other small experiment, on real quantum devices to fully assess the algorithm performance.
> However, as we elaborate, we do not see it as crucial as the referee suggests, atop of being often impossible in practice for most groups, as we explain next.
> But before going into these explanations, we point out that we did not quite understand what the referee means by reinforcement learning being an 'upper bound of performance'.
> One expects that a well-trained reinforcement learning agent is capable of finding much better results than any other method (i.e. ADAPT-VQE or UCCSD) by being able to explore a much larger combinations of gates, even those that one might not consider useful by a human.
> Regarding further testing on real quantum hardware, we would like to emphasize that this is a non-trivial task.
> In our view, doing so makes sense only in two cases,
> 1. when the differences between simulation and the real system make a systematic important impact on the conclusions we draw about our method.
> 2. when one wants to understand the performance of the quantum device itself.
>
> We are not concerned about the topic 2. Regarding topic 1, the likely main issues in learning have to do with inconsistent evaluations of the energy, which simulations capture well.
> Since this is the first paper to systematically evaluate RL in noisy simulations of this type, we believe that considering this source of differential behaviour is sufficient to highlight the potential and limitations of our methods.
> Further, as mentioned earlier, our parallel yet unpublished studies along with other works such as [1] suggest that indeed the ML systems do not have a significant differential performance in real vs. simulated noise.
>
> There is however another important issue we see with the suggestion of the referee that incorporating tests on real quantum devices is *crucial*, meaning for the results of the paper, and thus consequently for the potential of papers to be accepted to top conferences like ICLR.
>
> The issue is that real devices are exclusively proprietary, and free use does not give anywhere near enough access to do experiments of type we want. For instance, IBM has a *monthly limit of $10$ minutes for a free account*). And we believe the reviewer will agree that demanding that a necessary criterion for acceptance of a paper to top conferences is that they have funds to pay for extensive real QC runs would not be fair, and put smaller groups at a significant disadvantage, especially in the scenario where it is not even clear that these real runs reveal anything systematic about the method, and not just the specific *temperament* of a **particular** quantum computer.
>
>
>
>
>
> [1] Rehm, Florian et al., "Precise image generation on current noisy quantum computing devices." Quantum Science and Technology 9, no. 1 (2023): 015009.

---

> ### Author Response · Authors · 2023-11-19
> **Response 5 to reviewer WndL**
>
> > Given that you utilize a noise model from the current noisy environment and consider only one seed, real hardware testing is crucial.
>
> Going back to the potential use of free access, a typical VQE function evaluation (query) on IBM Lagos device with $20,000$ shots takes roughly $0.3$ seconds and training even a single circuit needs more than a thousand evaluations.
> Having said this, we can only barely manage to train just one episode of our method with the free access.
> On the other hand, one could potentially look for an experimental group interested on running such methods.
> However, in the vast majority of cases, experimental labs have dedicated hardware and software to run experiments which is really difficult to integrate for RL.
> Nonetheless, the goal of running noisy classical simulations with realistic noise models is to bring evidence to experimental groups that RL can be used in real hardware.
> We hope that our work motivates simultaneously theorists and experimentalists to run RL methods on real hardware for quantum architecture search.
>
>
> > Have you tried to implement your final result on real hardware? - I believe it should be easy for you since you already get your optimal arch and parameters from noisy simulation. You can just run this circuit on real quantum hardware and obtain the result.
>
> The first part of the question has been addressed in the *Response 4 to reviewer WndL*.
> We have not attempted to run any of the experiments (training CRLQAS agent) in real hardware due to time and access limitations, even for the minimal examples of running the optimal circuits for all the molecular systems considered in this study.
> While this experiment is interesting in its own right, it will not make the results more or less interesting: either the results are close, and our noise model is accurate, or the results are significantly different and we must adjust our noise model.
> The really interesting experiment is to run the full CRLQAS pipeline such that one can study the behaviour of the RL agent, assess the actual computational cost, and study the differences with respect to noiseless and noisy classical simulations.
> Unfortunately, to perform such an experiment we need access to a dedicated machine with no time limits.
> Moreover, public access to IBM Quantum devices is limited to 10 minutes per month, as of November 2023, with exceedingly high waiting times in the queues. A typical VQE function evaluation (query) on these devices with $20,000$ shots takes roughly $0.3$ seconds and training even a single circuit needs more than a thousand evaluations.
>
> However, we do perform a small comparative experiment on the real hardware and the noisy simulators (exactly modelled as in our work) for $H_2$ molecule with a circuit produced by a trained CRLQAS agent.
> We will update the manuscript in the coming days with the results of this experiment.
> In future, we hope to engage with experimental labs interested in attempting such a challenging computation.

---

> > ### Comment · Reviewer_WndL · 2023-11-21
> >
> > I really appreciate the author's efforts to address my questions. However, I still have two last concerns:
> > 1. The scalability of the CRLQAS. I understand the authors provide the data for the small qubit circuits, but what if we want to implement CRLQAS for the 1000 qubit-scale task? I think the noisy simulation could have an exponential cost with the increase in the number of qubits.
> >
> > 2. Based on my understanding, the noise model is an approximate graph of the real condition since the noise is time-dependent, which means this is actually a dynamics model. And I used to track the dynamics of the noise model directly on IBM devices. I observed that there was almost a sharp change in the noise model per week; it is not a serious problem for the QML tasks that the authors mentioned since such tasks can be finished in one day. However, for the CRLQAS, what I think is that if we implement maybe 100 qubits, it could take a couple days to process, which introduces a probability: the noise model input to the CRLQAS could be away from the current condition on the real machine.
> >
> > I understand the limited access to the cloud machines; then, I have personal suggestions for authors:
> > 1. Theoretically prove how you speed up your framework to avoid such a scalability problem.
> > 2. May come up with some interesting reward functions that can capture the dynamics of the noise.

---

> ### Author Response · Authors · 2023-11-22
> **Response to Concerns of Official Comment 2 by reviewer WndL**
>
> > I really appreciate the author's efforts to address my questions. However, I still have two last concerns:
>
> We thank the reviewer for the comments on our submission, we believe that by addressing them we have substantially improved our manuscript.
>
> > The scalability of the CRLQAS. I understand the authors provide the data for the small qubit circuits, but what if we want to implement CRLQAS for the 1000 qubit-scale task? I think the noisy simulation could have an exponential cost with the increase in the number of qubits.
>
> The reviewer is correct when stating that our noisy simulator has an exponential cost in the number of qubits. In fact, no faithful noisy simulator can avoid an exponential cost (as it has to be able to evaluate noiseless circuits as well), unless the whole of quantum computing can be efficiently simulated. However, we would like to reiterate that the noisy simulation was *only* needed to showcase the effectiveness of the learning process of our CRLQAS method (the core contribution of our paper). That being said, the cost of the framework simulating noise becomes  less relevant but its development was *only* necessary due to the lack of unrestricted access to a real device. Ultimately, even with a moderate number of qubits (e.g.,~30) the experiments will need to be performed on quantum hardware, and no simulator will be able to fully capture the complexity of these experiments.
>
> > Based on my understanding, the noise model is an approximate graph of the real condition since the noise is time-dependent, which means this is actually a dynamics model.
> And I used to track the dynamics of the noise model directly on IBM devices.
> I observed that there was almost a sharp change in the noise model per week; it is not a serious problem for the QML tasks that the authors mentioned since such tasks can be finished in one day.
> However, for the CRLQAS, what I think is that if we implement maybe 100 qubits, it could take a couple days to process, which introduces a probability: the noise model input to the CRLQAS could be away from the current condition on the real machine.
>
> Assuming the referee meant that the noise model is not a fully accurate simulation of a real device, they are correct when stating that the noise of a quantum device changes with time.
> For instance, superconducting qubit systems of IBM and Google have a drift time-scale of approximately a day.
> Of course it is not possible for us to study exact noise processes, and observing what happens in this *time-independent* noise model is a natural starting point. Claims for the noise model we have is the only claim we can make rigorously.
> However, we briefly discuss the likely outcomes of the use of our algorithm in other scenarios, namely the time drift aspect.
> Even though the system drifts, the quantum circuits generated by CRLQAS are still valid because after tuning the device, the noise in the system will have not changed massively. For example, superconducting qubits are dominated by T1 and T2 errors. These errors might vary within a day, yet they stay within the same order of magnitude even after re-calibration. The changes in these parameters remain within $\mu$ seconds for superconducting qubits [1].
> In a simulation, one could take the best and worst noise profiles of a quantum chip, creating a lower and an upper bound on what one could expect in a real experiment.
> Having a lower and an upper bound on the performance of the device one considers using before the start of an experiment is incredibly insightful because one can explore to what extent the quantum hardware can be used.
>
> &ensp;
>
> [1] Baheri, Betis, et al. "Quantum Noise in the Flow of Time: A Temporal Study of the Noise in Quantum Computers." In 2022 IEEE 28th International Symposium on On-Line Testing and Robust System Design (IOLTS), pp. 1-5. IEEE, 2022.

---

> ### Author Response · Authors · 2023-11-22
> **Response to Suggestions of Official Comment 2 by reviewer WndL**
>
> >I have personal suggestions for authors:
> >
> > 1. Theoretically prove how you speed up your framework to avoid such a scalability problem.
> > 2. May come up with some interesting reward functions that can capture the dynamics of the noise.
>
> We thank the reviewer for the suggestions and find them interesting; but as we will explain they are not really feasible in reasonable time frames, and we do not believe they should be a deciding factor in whether our work fits ICLR.
>
> 1. **Scalability of CRLQAS method:** Note that our method is *inherently heuristic since it employs deep RL methods*. By definition, it is thus not possible to provide analytic proofs, nor have any proofs of a similar kind ever been successfully provided in any related setting to our knowledge. Even beyond heuristics, predicting and reliably extrapolating the performance of quantum systems would be a clear holy grail of the field. However, also likely completely impossible, as the differential performance between quantum and classical systems itself must be intractable for a classical computer. It is not difficult to construct cases where getting information on the deviation between a classical and quantum model also allows us to approximate the outcome of the quantum model. \
> \
> **Scalability of classical noisy simulation of real device:** Any quantum simulation under noise takes exponential classical memory as the system size increases. The largest noisy simulations with a density-matrix simulation are of the order of 20 qubits [1, 2]. Therefore, we can not prove that we have a simulation method that scales up the noisy simulation for the speeds we need. Furthermore, if such a method were to exist, there would be no need for quantum computers at all.
>
> &ensp;
>
> 2. We would like to reiterate that capturing the drift aspect of the real device is not a key focus of our work, and that it would also not allow us to capture all the relevant noise anyways, as we explain next. Understanding and tracking down the actual drift processes (needed to make a simulator that reliably captures drift) is an active field of research in quantum computing, quantum control and even condensed matter physics. There has been some progress in predicting the drift of superconducting qubits (via Floquet dynamics [3, 4]). Nevertheless, theoretically, it only works on short time scales, on the order of hours after the drift. Drift, stemming from non-Markovian $1/f$ noise in quantum computers, remains challenging to accurately model for individual qubits within open quantum systems, let alone complex hardware [5]. First-principle approaches like the Nakajima-Zwanzing master equation are generally unsolvable or numerically complex, especially for the system sizes of present-day devices [6].  Even analogues like the Time-Convolutionless master equation rely on a perturbation expansion that might not fully capture the underlying physics of these devices [7]. Developing an RL method that learns to capture the dynamics of an open quantum system particularly with drifts is far from the scope of this work. Therefore, we can not fulfil this suggestion as it poses a scientific question that might not have an answer.
>
> \
> \
> \
> \
> [1] O’Brien, Thomas E., et al. "Density-matrix simulation of small surface codes under current and projected experimental noise." npj Quantum Information 3.1 (2017): 39
>
> [2] Jones, Tyson, et al. "QuEST and high performance simulation of quantum computers." Scientific reports 9.1 (2019): 10736.
>
> [3] Arute, Frank, et al. "Observation of separated dynamics of charge and spin in the fermi-hubbard model." arXiv preprint arXiv:2010.07965 (2020).
>
> [4] Google Quantum AI. (2023, May 18). Calibration FAQ. Retrieved November 22, 2023, from https://quantumai.google/cirq/noise/calibration_faq
>
> [5] Rol, Michiel Adriaan. "Control for programmable superconducting quantum systems." (2020).
>
> [6] Zwanzig, Robert. "Ensemble method in the theory of irreversibility." The Journal of Chemical Physics 33, no. 5 (1960): 1338-1341.
>
> [7] Vacchini, Bassano, and Heinz-Peter Breuer. "Exact master equations for the non-Markovian decay of a qubit." Physical Review A 81, no. 4 (2010): 042103.

---

### Official Review · Reviewer_wt33 · 2023-11-01

**Soundness:** 3 good
**Presentation:** 3 good
**Contribution:** 2 fair
**Rating:** 6
**Confidence:** 4

**Summary:**

The paper presents a new combined RL algorithm tailored for the domain of quantum architecture search (QAS). AN empirical study shows benefits over state-of-the-art approaches in this very early field of research.

**Strengths:**

The paper does a reasonable job introducing the issue of quantum architecture search and, from the presented difficulties of the domain, manages to derive its adaptations to standard RL employed to tackle this domain. The adaptations presented are sound and straight-forward. The results give a nice guideline for future research in that direction.

**Weaknesses:**

As with most approaches in the field of quantum computing, the evaluation is rather weak does not allow to draw too strong a conclusion (which, luckily, the authors did not draw). The presentation of the sequence of experiments is rather convoluted (and not helped by putting the plots very late in the document). More clear-cut scenario descriptions would be helpful.

The discussion lacks an explanation how adaptations which were largely motivated by the accommodation of noise models then manage to outperform the state-of-the-art in the noiseless case.

Also, I would have liked a singled-out study on the impact of the "illegal actions" adaptations. It appears that it might do a lot of the heavy lifting, which would then change the overall story of the paper.

Many discussions are left to the Appendix and several forward-references point to it. The structure should be improved here to make a clearer distinction between important and less important points.

The writing still has several problems including:
- p1, second paragraph, last sentence: no verb
- p2, fig1, caption: "(i)" without "(ii)"
- p3: "Methods like this" should read "Methods like these"
- p4, eq2: parameters/variables are not sufficiently explained in the text, at least not in a concise manner
- p4, last paragraph before 3.1, first sentence: It is unclear how the "environment" is deterministic/stochastic. Refer to transition function, reward function etc.
- p5, section3.3, throughout: Typo "T_{s^e}" instead of "T _s^e"?
- p4, section3.4: do not put comma after "where"
- p6, section3.6: Use "it is", not "it's"
- p6, section3.6: "the Appendix" randomly appears in a sentence

**Questions:**

see above...

What is the distinct impact of the "illegal actions" adaptation?

Why is the result even better upon "closer inspection"? (p7, very bottom)

---

> ### Author Response · Authors · 2023-11-19
> **Response 1 to reviewer wt33**
>
> Thank you for your thoughtful and constructive feedback on our paper, especially on the remark of performing a *singled-out study on illegal actions*.
> We appreciate your acknowledgement of the introduced adaptations to standard RL and the insights provided on the domain's challenges.
> During the rebuttal period, we performed additional ablation studies and improved the readability of the manuscript. In the general answer, we list down the introduced changes. We hope it is sufficient, please let us know if you have any more questions or comments.
>
>
> > As with most approaches in the field of quantum computing, the evaluation is rather weak and does not allow to draw too strong a conclusion (which, luckily, the authors did not draw)
>
> The reviewer is of course correct that it is difficult to draw strong conclusions form the volume and scale of simulations and especially real QC runs which are broadly possible today, but as the rest of the community, we put in maximal efforts to push the boundary.
> In our case, nearly a year's worth of work was invested in transitioning our simulation methods. We experimented with various off-the-shelf SDKs, such as Tensorflow Quantum, Qulacs, Qibo, and TorchQuantum, before settling on the relatively faster JAX framework.
>
> As a result of this transition, we've developed an in-house JAX package specifically tailored to reinforcement based QAS.
> This transition enabled us to perform slightly bigger simulations with more types of noise.
>
> > The presentation of the sequence of experiments is rather convoluted (and not helped by putting the plots very late in the document). More clear-cut scenario descriptions would be helpful.
>
> We are sorry to hear that the ordering as presented affected the readability for the reviewer.
> Our approach followed the convention of our research group, focusing on presenting important contributions upfront, followed by results and detailed descriptions in the appendix.
> Due to page limitations, we moved some descriptions to the appendix.
> To further enhance readability, we tried to improve on those and to ensure clarity, we have added more details in the appendix wherever applicable.
> If the reviewer has further suggestions for improving readability, we are more than happy to try to implement them in our work.
>
> > Many discussions are left to the Appendix and several forward-references point to it. The structure should be improved here to make a clearer distinction between important and less important points.
>
> We are thankful to the reviewer for pointing out the problems and typos in the text. In the revised version of the manuscript, we incorporated the changes and typing mistakes.
>
> By "environment" is deterministic/stochastic we meant that the reward function can be stochastic because it depends on the cost function evaluation.
> The nature of the CRLQAS reward function depends on the cost function evaluation.
> The cost function evaluation can be deterministic or stochastic based on both the classical optimizer and quantum noise.
> Stochastic optimizers, like SPSA, tend to converge toward different parameters, resulting in different function values across multiple trials, even when initialized identically.
> Moreover, the quantum noise may also lead to different function values even with the same parameters.
> We have clarified this aspect in the text by including this description in the revised manuscript.

---

> ### Author Response · Authors · 2023-11-19
> **Response 2 to reviewer wt33**
>
> > The discussion lacks an explanation how adaptations which were largely motivated by the accommodation of noise models then manage to outperform the state-of-the-art in the noiseless case. Also, I would have liked a singled-out study on the impact of the "illegal actions" adaptations. It appears that it might do a lot of the heavy lifting, which would then change the overall story of the paper.
>
> We agree that additional ablation studies would strengthen the paper.
> To this end, we ran simulations to conduct a detailed study to find out which of the newly proposed features stands out.
> These simulations implement a controlled study of each feature by training a different CRLQAS agent for a given problem with and without each feature.
> Our analysis reveals that incorporating illegal actions without random halting prompts the agent to achieve a positive signal (a successful episode) early on, albeit resulting in larger circuits.
> Conversely, introducing random halting encourages the agent to discover shorter circuits, but there is a trade-off as the agent receives the positive signal at a later stage.
> We provide a table (see *Table 1*) of these studies.
> We will update the manuscript and visualize the data from the *Table1* more tactfully in the coming days.
>
>
> > What is the distinct impact of the "illegal actions" adaptation?
>
> As previously noted, to answer this question, we initiated a new set of ablation studies.
> These studies not only center on assessing the impact of "illegal actions" but also delve into the effects of implementing "random halting".
> Our analysis reveals that incorporating illegal actions without random halting prompts the agent to achieve a positive signal (a successful episode) early on, albeit resulting in larger circuits.
> Please refer to  *Table 1 in Response 3 to reviewer wt33* where we provide the statistics of experiments.
>
>
>
> In more detail, the distinct impact of "illegal actions" adaptation is the constraining of the combinatorially large search space of the possible gate combinations such that a more efficient search can be conducted.
> We see this trend in our ablation studies via the early observation of circuits that can achieve good energy estimates, such as estimates within chemical accuracy (please refer to the revised manuscript).
> The simple rule of not allowing the addition of two similar gates subsequently on the same qubits (i.e., not allowing two CNOTs on the same control \& target qubits in a row, or not allowing two single qubit rotation in the same direction on the same qubit in a row) led the agent to get a good signal at the random exploration stage early on such that it could further learn new patterns leading to even better circuits.
> In the absence of illegal actions, the agent is deprived of such a signal during the random exploration stage (i.e., when the randomness of the actions parametrized by $\epsilon$-greedy agent for higher values of $\epsilon$ ) so that it can never or can barely learn to construct good circuits.
> This trend is quite apparent in the case of the 4-qubit $H_2$ molecule solved with limited qubit connectivity, and IBM Ourense device noise model - a problem we derived from the QCAS (Du et al., 2022).
> Here, with illegal actions and without random halting, CRLQAS agents that are trained for 3 random seeds were able to achieve chemical accuracy in roughly $1200$ (median of seeds) out of $15000$ episodes.
> Similarly, with illegal actions but with random halting, they achieve chemical accuracy (CA) in roughly $300$ (also median) out of $15000$ episodes.
> In contrast, without illegal actions and without random halting, the agents achieve CA in only $10$ (median) out of $15000$ episodes.
> Lastly, without illegal actions and with random halting, the agents achieve CA in only $6$ (median) out of $15000$ episodes.
> Training with and without illegal actions leads up to a three-order-of-magnitude difference in the number of successful episodes for this scenario.
> We refer the reviewer to the *Table1 in Response 3 to reviewer wt33* where we provide in-depth statistics of our ablation study in both noisy and noiseless scenarios.
> We will update the manuscript and visualize the data from the *Table 1 in Response 3 to reviewer wt33* more tactfully in the coming days.

---

> ### Author Response · Authors · 2023-11-19
> **Response 3 to reviewer wt33**
>
> > Ablation studies on different adaptations of CRLQAS method
>
>
> ### Table 1
> We performed a detailed study to find out which of the newly proposed features of CRLQAS method stands out in both noisy (for 4-qubit $LiH$ and $H_2$) and noiseless (for 6-qubit $LiH$) settings. Below we first find an episode with the best noiseless error and acquire circuit statistics (i.e., depth, gates, etc.) as well as the noisy error of that episode. The last two columns quantifies the learning performance of the CRLQAS method. We describe the median statistics over three and five random seeds for noisy and noiseless experiments, respectively. We emboldened the noiseless errors which pass the chemical accuracy. Below, $\text{wo-X}$ denotes switching off the feature $\text{X}$ where $\text{X}$ $\in$ {$\text{IA}, \text{RH}$}.
>
> | Molecule                                               |   Environment & Agent |   Noisy Error |   Noiseless Error |   Depth |   No. of Gates |   Parameters |   Actions to First Successful Episode |   No. of Successful Episodes |
> |:---------------------------------------------------------:|:-------------------------------:|-------------------------:|-------------------:|--------:|------------:|-----------:|------------------------:|----------------:|
> | $LiH-4$ |     (wo-IA, wo-RH, IBM Mumbai Median)     |              0.0581266   |        0.00257938  |      24 |          40 |         33 |                       N.A. |               N.A. |
> | $LiH-4$   |           (IA, wo-RH, IBM Mumbai Median) |              0.081803    |        **0.00150412**  |      23 |          40 |         30 |                   66221 |               1 |
> | $LiH-4$   |           (wo-IA, RH, IBM Mumbai Median) |              0.0535075   |        0.00288408  |      15 |          29 |         21 |                       N.A. |               N.A. |
> | $LiH-4$     |           (IA, RH, IBM Mumbai Median) |              0.0669863   |        0.00242298  |      15 |          29 |         23 |                       N.A. |               N.A. |
> | $LiH-4$               |           (IA, RH, IBM Mumbai Median & shot noise) |              0.073456    |        0.00376099  |      17 |          28 |         18 |                       N.A. |               N.A. |
> | $H_2-4$                         |           (IA, wo-RH, IBM Ourense) |              0.238207    |        **0.000115746** |      26 |          40 |         14 |                   59311 |            1209 |
> | $H_2-4$                       |           (wo-IA, wo-RH, IBM Ourense) |              0.311764    |        **0.000422948** |      29 |          40 |         23 |                   58157 |              10 |
> | $H_2-4$                          |           (IA, RH, IBM Ourense) |              0.137199    |        **0.000222438** |      27 |          32 |          5 |                   42331 |             217 |
> | $H_2-4$                         |           (wo-IA, RH, IBM Ourense) |              0.152238    |        **0.000308149** |      19 |          27 |         22 |                   35593 |               6 |
> | $LiH-6$                               |           (Ostaszewski et al. 2021, noiseless) |              N.A.  |        0.00242028  |      24 |          55 |         29 |                       N.A. |               N.A. |
> | $LiH-6$                               |           (IA, wo-RH, noiseless) |              N.A. |        **0.000832465** |      27 |          56 |         26 |                  356604 |              30 |
> | $LiH-6$                                 |           (IA, RH, noiseless) |              N.A.  |        **0.00121936**  |      30 |          59 |         37 |                  641593 |               3 |
> | $LiH-6$                               |           (wo-IA, RH, noiseless) |              N.A.  |        0.00372759  |      23 |          45 |         25 |                       N.A. |               N.A. |
> | $LiH-6$                             |           (wo-IA, wo-RH, noiseless) |              N.A. |        0.00163461  |      31 |          62 |         31 |                       N.A. |               N.A.

---

### Official Review · Reviewer_sjih · 2023-11-05

**Soundness:** 2 fair
**Presentation:** 2 fair
**Contribution:** 2 fair
**Rating:** 6
**Confidence:** 5

**Summary:**

This paper tackles the challenge of identifying optimal quantum circuits within the constraints of noisy intermediate-scale quantum (NISQ) devices. It introduces a novel curriculum-based reinforcement learning algorithm for quantum architecture search (CRLQAS), which innovatively employs a 3-D architecture encoding, an episode halting mechanism, and an enhanced simultaneous perturbation stochastic approximation optimizer for more efficient exploration and convergence. By incorporating Pauli transfer matrix formalism to improve simulation times, the proposed CRLQAS demonstrates superior performance in automating the design of quantum circuit architectures, outstripping existing methods in both noiseless and noisy conditions, specifically for quantum chemistry applications.

**Strengths:**

The paper excels in its strategic approach to quantum computing in the NISQ era by giving paramount importance to the role of quantum noise and its effect on circuit architecture. It introduces a pioneering curriculum-based reinforcement learning (RL) algorithm, CRLQAS, which specifically addresses the challenges posed by noise in quantum systems. The use of the Pauli transfer matrix formalism in this algorithm marks a substantial improvement in the simulation of noisy quantum circuits, significantly reducing computational time and increasing simulation fidelity.

In addition to the quantum-specific advancements, the paper contributes to the field of RL by implementing a 3-D architecture encoding and an episode halting mechanism within its curriculum-based framework, enhancing the RL agent's ability to discover shorter and more hardware-efficient circuit architectures rapidly. The introduction of a novel optimization technique further refines the learning process, facilitating faster convergence and potentially leading to more robust quantum computing solutions. These combined strengths showcase the paper's dual focus on developing an efficient simulation that carefully accounts for quantum noise and on refining RL techniques to ensure that the architecture search yields practical, hardware-efficient designs.

**Weaknesses:**

One area for improvement in the paper is the absence of the promised source code, which limits the ability of others to replicate and build upon the work. It is essential for the authors to provide the source code to enhance the transparency and applicability of their research. Therefore, it is recommended that the authors include a link to the open-sourced code or an appendix containing the code in any subsequent revision of the paper. This addition would be a valuable resource for the community and would greatly facilitate further research and validation of the proposed methods.

The paper could be enhanced by broadening the scope of its comparative analysis with existing methodologies. While the current work focuses on a specific approach to quantum circuit optimization, there are alternative methods worth considering, such as those based on SMT (Satisfiability Modulo Theories) solvers. For instance, the technique presented by B. Tan and J. Cong in "Optimal layout synthesis for quantum computing," (ICCAD, 2020) could serve as a valuable benchmark. It would be beneficial for the authors to include a comparison with these kinds of potentially optimal circuit compilers to contextualize the effectiveness of their proposed method when applied to analogous problems.

Furthermore, the terminology used in the paper, specifically the term "architecture," may lead to ambiguity within the broader scientific community. In the realm of computer architecture, "architecture" typically refers to the low-level hardware design, as discussed in sources like the one available on IEEE Xplore. Since this work is centered around the optimization of parameterized quantum circuits, and not the hardware itself, clarity could be improved by selecting a term that more accurately describes the subject of optimization. A term like "circuit design" or "circuit configuration" may better capture the essence of the research without the potential for misinterpretation.

**Questions:**

1. Can the authors provide the source code that was mentioned as a part of the paper's contributions to facilitate replication and further research by peers?

2. Would it be possible to expand the comparative analysis section of the paper to include methodologies based on SMT solvers, such as the approach detailed by B. Tan and J. Cong in ICCAD 2020, to provide a broader context and demonstrate the relative effectiveness of the proposed algorithm?

3. Considering the potential for confusion within the interdisciplinary community, could the authors consider using a more precise term than "architecture" to describe the optimization of parameterized quantum circuits, thereby avoiding conflation with low-level hardware design?

---

> ### Author Response · Authors · 2023-11-19
> **Response to reviewer sjih**
>
> We would like to thank the referee for the time spent reviewing this submission and for its appreciation of the timeliness and importance of this work to accelerate the current state of quantum computation with near-term quantum hardware. During the rebuttal period, we performed additional ablation studies and improved the readability of the manuscript. In the general answer, we list down the introduced changes. We hope it is sufficient, please let us know if you have any more questions or comments.
>
> > Can the authors provide the source code that was mentioned as a part of the paper's contributions to facilitate replication and further research by peers?
>
> We thank the reviewer for bringing this issue to our notice. Currently, we are in the process of refining and polishing the codebase.
> An anonymous version of the code will be made available before November 22, 2023. Thank you for your understanding and patience.
>
>
> > Would it be possible to expand the comparative analysis section of the paper to include methodologies based on SMT solvers, such as the approach detailed by B. Tan and J. Cong in ICCAD 2020, to provide a broader context and demonstrate the relative effectiveness of the proposed algorithm?
>
> We appreciate the comment of the reviewer. However, there are subtleties in what the objectives of what these other papers achieve compared to our own which makes it hard drawing conclusion from such comparisons, and consequently we opted not to do them.
>
> In particular, (Tan et al., 2020) deals with the quantum compilation or transpilation problem. This problem involves decomposing a given static unitary (a quantum program) in terms of subsequent native gates (unitaries) of certain quantum hardware. This compilation process may account for hardware constraints such as noise, or the timing \& queues of pulse sequences implementing these gates.
> In fact our introduction of illegal actions scheme was inspired from quantum compilation and transpilation techniques.
> And the effectiveness of this simple scheme is highlighted by the ablation studies we performed (see *Table 1 from Response 3 to reviewer wt33*).
>
> The quantum architecture search (QAS) problem we are dealing with is more general. First of all, QAS problem does not deal with finding decompositions of a static circuit unitary but deals with finding families of circuit unitaries that solves certain type of problems such as state preparation, machine learning (classification, regression, etc.), or quantum chemistry (VQE). Second, our RL-based QAS method builds circuits out of primitive gate sets while being constrained by the noise of such gate sets. In that regard, the reviewer is right to suggest that we can embed the quantum compilation or transpilation problem for certain hardware within our QAS method.
> Incorporating CRLQAS with analogous problems, as suggested, holds potential interest, but it's a direction we intend to explore in future endeavors rather than within the context of this particular paper.
>
>
> > Considering the potential for confusion within the interdisciplinary community, could the authors consider using a more precise term than "architecture" to describe the optimization of parameterized quantum circuits, thereby avoiding conflation with low-level hardware design?
>
> We understand the reviewers concern about the use of 'architecture' to describe a quantum circuit, as this concept exists with broad meanings depending on the field.
> Unfortunately, the term 'quantum architecture search' has been used in the quantum computing community since its introduction in 2020 by (Du et al. npj Quantum Inf 8, 62, 2022) and (Zhang et al. Quantum Science and Technology 7 045023, 2022).
> The concept is an extension to that of neural architecture search (NAS) in classical machine learning.
> We agree with the reviewer that there exist alternative ways of describing the process of building a quantum circuit, but it would require a community effort to get rid of the 'quantum architecture search' concept.
> Furthermore, the 'architecture' is integral to the title of our paper, and per conference guidelines, we are unable to modify it.
> We apologize for any confusion this concept might cause, and hope that the reviewer understands our position.

---

> > ### Comment · Reviewer_sjih · 2023-12-01
> > **feedback**
> >
> > I appreciate the authors' perspective, however, I respectfully differ in opinion regarding the statement that 'QAS problem does not deal with finding decompositions of a static circuit unitary.' In my understanding, QAS typically begins with a static circuit unitary. Additionally, I feel that there was a missed opportunity to address the use of terminology in quantum architecture, which is a concern of mine. Due to these reasons, I feel compelled, though with reluctance, to adjust the score accordingly.

---

### Official Review · Reviewer_K92h · 2023-11-08

**Soundness:** 2 fair
**Presentation:** 1 poor
**Contribution:** 3 good
**Rating:** 6
**Confidence:** 4

**Summary:**

Variational quantum algorithms (VQAs) stand for a promising candidate of practical quantum algorithms in the NISQ era. Nevertheless, the performance of VQA is significantly influenced by the choice of ansatz and noise effects.

In this paper, the authors adopt curriculum reinforcement learning for quantum architecture search (QAS), aiming to select ansatz with good performance under hardware errors. Specifically, they devise a CRLQAS algorithm with tensor-based 3-D encoding of quantum circuits, a random halting scheme to reduce the circuit length, and an variant of Adam-SPSA for better convergence. In addition, this work accecelerate simulation of noisy quantum circuits by employing pauli transfer matrix formalism.

Numerical experiments on quantum chemistry tasks demonstrate the performance of the proposed method under noiseless and noisy circumstances.

**Strengths:**

- Originality: As far as I know, this is the first work to consider noise models from real quantum machines in RL-based QAS. Previous work focus on noiseless scenarios or only touch on simple noisy configurations. The setting of this work is closer to real applications in NISQ era, hence making it an advance over the previous state-of-the-art.
- Quality: The paper proposes several techniques which are effective and easy to realize. The tensor-based binary encoding is straightforward but efficient and compact. Pruning of illegal actions successfully narrow the search space. Random Halting is introduced to reduce the training episodes, which is a major drawback of previous work. The offline computation with Pauli-transfer matrices and JAX achieves considerable acceleration for noise simulation.
- Clarity: The main ideas and techniques in the paper are mostly clear, but many issues remain (see Weaknesses & Questions).
- Significance: This paper seeks to tackle an important problem by considering noise of current devices in QAS for VQA, which is well-motivated. Literature discussing the negative impacts of noise on VQA has been thoroughly cited. The numerical demonstration of achieving chemical accuracy and outperforming previous methods is impressive.

**Weaknesses:**

- The writing is poor, full of mistakes in grammar and notation. For example, the first sentence in the second paragraph of Section 3.6 is incomplete. There are erroneous notations such as  $T_{s}^{e}$ v.s $T_{s^{e}}$ in Section 3.3, wrong format of citation like (Wang et al., 2021) in introduction, inconsistent proper nouns such as "ansatz" v.s. "ans{\"a}tze", and numerous syntax errors. These issues impair the quality of the paper.
- Originality: This paper extends the ideas from previous work (Ostaszewski et al., 2021). Some techniques such as feedback driven curriculum learning and Adam-SPSA seem to be the direct adaptation of existing methods.
- Although several techniques are proposed, the paper may lack insights into which component principally improves the results, especially under noisy circumstances. It is unclear to me which technique is specialized for facilitating noise adaptations, rather than general improvement over existing methods.
- Related to the previous question, in the noisy simulation section, the paper conducts no comparison with other RL-based QAS methods. Therefore it is unclear what is the improvement of this algorithm over previous methods in noisy environments. Moreover, for molecules such as LiH-4, the experiments only consider shot and 1-qubit depolarizing noise, which is far from real noise models. This undermines the persuasiveness of this work.
- Important related work not discussed. Despite little consideration of noise in previous reinforcement learning methods for QAS, there have been many attempts in traditional architecture field. The paper may need more comparison with other QAS algorithms under hardware erors, especially those targeting noise-aware search, for example Wang et al. QuantumNAS 2022.

**Questions:**

- Please proofread the manuscript and correct the numerous mistakes in grammar and notation. Also there is a repetition in the reference section, where "Abhinav Kandala, Antonio Mezzacapo, Kristan Temme, Maika Takita, Markus Brink, Jerry M Chow, and Jay M Gambetta. Hardware-efficient variational quantum eigensolver for small molecules and quantum magnets. nature, 549(7671):242–246, 2017" is listed twice.
- Some descriptions are unclear to me. In Section 3.3, what does "$n_s$ is the number of successes" mean? Details in Section 3.4 could also be better stated. Moreover, from a reader's point of view, the organization of Section 3 is scattered and lacks emphasis on major contributions.
- Does the model need retraining for different noise models to deliver competitive results? I am curious about the efficiency for adaptation to new noise models. Also I believe experiments on real quantum machines comparing the proposed method with other algorithms would make the work more solid.
- Although the authors claim to tackle the general employment of VQAs, numerical experiments on applications are actually limited to VQE for quantum chemistry tasks. The authors should avoid overstatement, or apply the method to general VQA settings and tasks.
- There is almost no discussion of limitations. What is the weakness of this method and what are the potential future directions?

---

> ### Author Response · Authors · 2023-11-19
> **Response 1 to reviewer K92h**
>
> We thank the referee for the time spent in this review, and for remarking the significance of this work in order to assess the viability of reinforcement learning strategies in combination with noisy quantum hardware.
> Additionally, we appreciate the referee's comment about our implementation of Pauli transfer matrix formalism accelerated with GPU.
> During the rebuttal period, we performed additional ablation studies and improved the readability of the manuscript. In the general answer, we list down the introduced changes. We hope it is sufficient, please let us know if you have any more questions or comments.
>
> > This paper extends the ideas from previous work (Ostaszewski et al., 2021). Some techniques such as feedback driven curriculum learning and Adam-SPSA seem to be the direct adaptation of existing methods.
>
> We agree with the reviewer that the paper treats (Ostaszewski et al., 2021) as a baseline to improve upon. On the other hand, we respectfully disagree with the reviewer in regard to the comment that this work is a mere adaptation of existing methods.
> In particular, we numerically demonstrate that the newly developed tensor-based binary encoding, the illegal actions, and random halting all improve the previous work *quantitatively* by orders of magnitude in metrics such as the first successful episode number, the total number of successful episodes, circuit depth, etc. Similarly, a version of Adam-SPSA we developed in this paper helps the RL agent tackle finite-sampling noise in a query way.  Moreover, we improved the existing works *qualitatively* by making such training, and their simulation, possible under a realistic noise model.
> Without the curation, creation and adaptation of such elements \& methods training a CRLQAS agent would simply be not possible under realistic noise.
> Our goal was to demonstrate that further improvements can be taken to combine reinforcement learning methods with near-term quantum hardware. And, such studies are contingent on the techniques developed in this paper.
>
> > Although several techniques are proposed, the paper may lack insights into which component principally improves the results, especially under noisy circumstances. It is unclear to me which technique is specialized for facilitating noise adaptations, rather than general improvement over existing methods.
>
> We agree with the reviewer on the lack of ablation studies on which component improves the result in noiseless and noisy instances with the highest impact. Such a study is valuable to the scientific community who will build upon our results. We are providing a detailed ablation study, which involves numerical experiments that are not included in the manuscript, and its visualization in the revised manuscript.
> For now, we please bear with us and refer to the *Table 1 of Response 2 and Response 3 to reviewer wt33* to the reviewer on the question about the same.
>
> > Moreover, for molecules such as LiH-4, the experiments only consider shot and 1-qubit depolarizing noise, which is far from real noise models. This undermines the persuasiveness of this work.
>
> We respectfully disagree with the reviewer as we provided results in that matter, but we agree that we should have highlighted them more.
> We would like to point to the reviewer that in section 4.2 we studied two more noise models for LiH-4, even though they did not achieve chemical accuracy.
> Namely, a model where 1-qubit depolarizing noise of strength $0.001$ is combined with 2-qubit depolarizing noise of strength $0.005$, and the median noise profile of IBM Mumbai device as described in the paper.
> We conducted further experiments since the submission of the manuscript.
> We will elaborate more about these results in the revised version in the coming days.
> For now, please refer to *Table 1 of Response 3 to reviewer wt33*.
>
>
> > Although the authors claim to tackle the general employment of VQAs, numerical experiments on applications are actually limited to VQE for quantum chemistry tasks. The authors should avoid overstatement, or apply the method to general VQA settings and tasks.
>
> We agree with the reviewer that within the scope of this paper, we are only dealing with a particular variant of VQAs, namely the VQEs in the quantum chemistry context. We will make the corrections in the revised manuscript. On the other hand, we kindly disagree with the statement that our method is limited to VQEs. No component of our method imposes such a restriction or is limited to quantum chemistry tasks.
>
> > There is almost no discussion of limitations. What is the weakness of this method and what are the potential future directions?
>
> Thank you for pointing this out. We have added a new section in the Appendix A of the revised manuscript on the topics of computational demands, evolution of RL methods, validation on real hardware, scalability challenges and transfer learning exploration.

---

> ### Author Response · Authors · 2023-11-19
> **Response 2 to reviewer K92h**
>
> > In the noisy simulation section, the paper conducts no comparison with other RL-based QAS methods. Therefore it is unclear what is the improvement of this algorithm over previous methods in noisy environments.
>
> We respectfully disagree with the reviewer. To our knowledge, this work is the first study of classical RL-based quantum architecture search for quantum chemistry problems in the presence of quantum hardware noise.
> Additionally, ref. [1] concentrate on addressing the state preparation problem specifically for $2$- and $3$-qubit GHZ states under hardware noise whereas our focus lies in addressing the fundamentally different challenge of ground state energy estimation for quantum chemistry tasks.
> It is important to note that in the state preparation problem, the algorithm receives the target state as an input.
> In contrast, for ground state energy estimation problem, the algorithm does not have any prior information about the ground state, inherently rendering the it a notably challenging problem.
>
> > Important related work not discussed. Despite little consideration of noise in previous reinforcement learning methods for QAS, there have been many attempts in the traditional architecture field. The paper may need more comparison with other QAS algorithms under hardware errors, especially those targeting noise-aware search, for example, Wang et al. QuantumNAS 2022.
>
> We kindly disagree with the reviewer.
> It may be the case it was missed due to the text structure, but we in fact did consider these other works (or similar), and reported on it in the paper.
> We have referred to QAS algorithms employing search strategies such as sampling-based, evolutionary algorithms (EA) and RL in sec. 2 of our manuscript.
> Furthermore, our study includes a comparison with QCAS (Du et al., 2022), a one-shot super-circuit method, under both hardware noise and qubit-connectivity constraints for the $4$-qubit $H_2$ molecule.
>
> Our understanding of the paper suggests that QuantumNAS (Wang et al., 2022) falls within the family of one-shot super-circuit methods.
> This technique involves sharing gate parameters among sub-circuits after one shot training to accelerate circuit evaluation.
> However, optimizing the super-circuit poses a challenge, and furthermore, there is a weak correlation in performances between sub-circuits with inherited parameters and those with optimal parameters from individual training.
>
> We investigated QuantumNAS manuscript in an attempt to facilitate either a fully-fledged quantitative or qualitative comparison.
> Unfortunately, the work lacks essential information for such an analysis. Specifically, details about the molecular geometries and quantum circuits are absent, preventing us from conducting a quantitative experiment for comparison.
> Consequently, this would require us to run their algorithm from scratch for the molecules considered in this study.
> Hence, due to the fact that we already compared to an algorithm (QCAS) which is akin to QuantumNAS and the reasons mentioned above, we believe that our comparison to existing methods (like QCAS, qubit-ADAPT VQE, quantumDARTS) is fair, and covers most of the existing approaches for quantum chemistry problems.
>
> Nevertheless, a qualitative comparison is only feasible in the case of 2-qubit $H_2$ molecule.
> Examining fig. 17 in ref. [2], we note that the best result achieved in those experiments exhibits an energy error of $0.1$ Hartree. In contrast, our $H_2$ simulations, conducted over three independent seeds (for median of IBM Mumbai device), yield a mean noisy energy error of $1.116 \times 10^{-3} \pm 3.41 \times 10^{-4}$—a performance three orders of magnitude superior to their outcomes. Furthermore, we outperform their results for IBM Mumbai at max and 10max noise levels, with mean noisy energy errors of $0.907 \pm 0.008$ and $0.0912 \pm 0.004$, respectively.
> Regrettably, there is no available information regarding the number of gates, depth, or parameters achieved by the method presented in ref. [2].
>
> [1] Kuo, En-Jui, Yao-Lung L. Fang, and Samuel Yen-Chi Chen. "Quantum architecture search via deep reinforcement learning." arXiv preprint arXiv:2104.07715 (2021).
>
> [2] Wang, Hanrui, Yongshan Ding, Jiaqi Gu, Yujun Lin, David Z. Pan, Frederic T. Chong, and Song Han. "Quantumnas: Noise-adaptive search for robust quantum circuits." In 2022 IEEE International Symposium on High-Performance Computer Architecture (HPCA), pp. 692-708. IEEE, 2022.

---

> ### Author Response · Authors · 2023-11-19
> **Response 3 to reviewer K92h**
>
> > Please proofread the manuscript and correct the numerous mistakes in grammar and notation. Also there is a repetition in the reference section, where "Abhinav Kandala, Antonio Mezzacapo, Kristan Temme, Maika Takita, Markus Brink, Jerry M Chow, and Jay M Gambetta. Hardware-efficient variational quantum eigensolver for small molecules and quantum magnets. nature, 549(7671):242–246, 2017" is listed twice.
>
> Our sincerest apologies for any inconvenience caused by grammatical mistakes and notation.
> Again, we would like to thank the reviewer for spotting typos and notation errors.
> In response, we have rectified those highlighted by the reviewer and have also addressed additional errors discovered internally in the updated manuscript. These modifications have significantly enhanced the readability and overall consistency of the text.
>
> > Some descriptions are unclear to me. In Section 3.3, what does "is the number of successes" mean? Details in Section 3.4 could also be better stated. Moreover, from a reader's point of view, the organization of Section 3 is scattered and lacks emphasis on major contributions.
>
> We agree with the reviewer regarding the inconsistencies within the text. We conducted a thorough proofreading and introduced several revisions in the updated manuscript.
> For instance, as highlighted by the reviewer, we restructured the text. Instead of the 'number of successes' of the negative binomial distribution we refer to 'number of failed Bernoulli trials' of this distribution in the text.
> Additionally, we've refined the method descriptions in sec. 3, focusing solely on the presentation of new methods. The detailed explanation of the modified Feedback-driven curriculum, initially included in sec. 3, has been relocated to Appendix B.1.
>
> > Does the model need retraining for different noise models to deliver competitive results? I am curious about the efficiency for adaptation to new noise models.
>
> The reviewer raises a fascinating point that we are also deeply interested in; i.e., the matters of transfer \& active learning in the context of different noise models are indeed relevant questions both for practical concerns (e.g., leveraging training time by pre-training one and using it on the other), and for the fundamental understanding of the RL methods.
> But, this is not the core of our question in this work.
> Nonetheless, to lay the foundations for such a study, we trained different CRLQAS agents in different noisy environments for different Hamiltonians of the same system size. Also, we conducted two naive experiments where the neural network weights \& the training memory buffer of a CRLQAS agent were transferred to different problems without any alterations. In the first one, we transferred agents trained in noiseless environments for the chemistry problem of different bond dimensions of the same molecule. Here we observed an agent trained within the scope of a higher bond-dimension problem (more difficult) to perform well in an easier problem, but not the other way around. In the second one, we transferred agents trained in the noiseless environment of the same molecule to a noisy environment. Here we also found that the agent was able to react to the new environment by synthesizing circuits that achieve chemical accuracy. On the other hand, we found that after a few thousand episodes the agents were not able to perform their task as adequately anymore due to catastrophic forgetting. We deemed a more systematic, elaborate future study is needed for this task. We believe such a study is *contingent* on the results \& techniques developed in our study.
>
> > Also I believe experiments on real quantum machines comparing the proposed method with other algorithms would make the work more solid.
>
> We kindly disagree with the reviewer on the necessity of real hardware experiments, especially at this stage.
> Since similar question was brought up by reviewer WndL, we have addressed this concern in our response to them. Please refer the answer there in *Response 4 and Response 5 to reviewer WndL*.

---

> > ### Comment · Reviewer_K92h · 2023-11-23
> >
> > I would like to thank the authors for the detailed response and clarification. I really appreciate your efforts for addressing the concerns and the improvement over the rebuttal period. I have no further questions.

---

### Author Response · Authors · 2023-11-20
**General Response**

We are grateful for the invaluable feedback provided by the reviewers. Their comments have not only motivated us but have significantly contributed to enhancing our manuscript.

In the revised version, we carefully addressed their comments and the changes made are highlighted in blue for ease of identification. Specifically, we improved the Introduction, Section 3 (describing CRLQAS and its novel features), Section 4 (Experiments), re-shuffled and rewrote the some appendices for better readability. We have added a limitations and future works section in the appendix A and added appendix J which indicates comparable behaviour of real device and classical noisy simulation.


## Strengths

1. Our work's novelty in developing features that enhance the learning process and reduces the search space and training episodes in the CRLQAS method, was appreciated by multiple reviewers (*K92h, sjih, 3EHX*).

2. Acknowledgment of our framework's efficiency in simulating noise profiles akin to real quantum devices was highlighted as a notable strength (*K92h, sjih, 3EHX, WndL*).

3. Demonstrating the achievement of chemical accuracy and superiority over previous methods in both noisy and noiseless settings was well received (*K92h, WndL*).



## Weaknesses and Responses

>1. Absence of Ablation study of proposed features of CRLQAS (wt33, K92h).

Addressing the impact of newly proposed CRLQAS features in noisy and noiseless settings has been thoroughly explored in our response to *reviewer wt33*. Our analysis highlights the trade-offs between incorporating illegal actions without random halting and introducing random halting, influencing circuit size and episode success (*see our [Response 2](https://openreview.net/forum?id=rINBD8jPoP&noteId=hxBQJf3CDY) and  [Response 3](https://openreview.net/forum?id=rINBD8jPoP&noteId=3W3ECnUUXJ) to reviewer wt33*) .

>2. Absence of comparison to other relevant works (K92h, 3EHX, WndL).

While other relevant works were suggested, limitations in these references hindered a comprehensive quantitative comparison. Within the suggested references, QuantumNAS [1] appeared frequently. We did not consider [1] as we already compared to QCAS method [2] and performed better. Both QCAS and QuantumNAS fall within the same family of one-shot super-circuit methods. We made qualitative comparisons wherever feasible and established fair comparative analyses with methods like QCAS, qubit-ADAPT VQE, and quantumDARTS (the state-of-the-art method) (*see our [Response 2 to reviewer 3EHX](https://openreview.net/forum?id=rINBD8jPoP&noteId=Lx1B4hqwbb) and [Response 3 to reviewer WndL](https://openreview.net/forum?id=rINBD8jPoP&noteId=vQyN28PAiI)*).

>3. Absence of experiments on real quantum device (K92h, WndL).

Acknowledging the importance of experiments on real quantum devices, we conducted a comparative experiment verifying simulated outcomes against actual hardware, aligning with recent studies (which was studied in different context) indicating comparable results despite device drifts [3] (*see Appendix J in the revised manuscript*).
We ask the reader to also refer to our [Response 4](https://openreview.net/forum?id=rINBD8jPoP&noteId=vw4nbGHtGP) and [Response 5](https://openreview.net/forum?id=rINBD8jPoP&noteId=7yqThAeBSi) *to reviewer WndL* for larger context.

We've made significant efforts to address these points, striving to strengthen our manuscript in line with the reviewers' feedback.








[1] Wang, Hanrui, et al. "Quantumnas: Noise-adaptive search for robust quantum circuits." 2022 IEEE International Symposium on High-Performance Computer Architecture (HPCA). IEEE, 2022.

[2] Du, Yuxuan, et al. "Quantum circuit architecture search for variational quantum algorithms." npj Quantum Information 8.1 (2022): 62.

[3] Rehm, Florian et al., "Precise image generation on current noisy quantum computing devices." Quantum Science and Technology 9, no. 1 (2023): 015009.

---

> ### Author Response · Authors · 2023-11-22
> **Anonymized Source Code is now Available Online**
>
> Dear reviewers,\
> \
> We would like to draw your attention that the source code used to conduct experiments in our manuscript is now available here: <https://anonymous.4open.science/r/CRLQAS>. We have included it in the Reproducibility Statement of the manuscript. \
> \
> Additionally, we have also now included the results from our *Ablation Study* to identify the features that standout within the CRLQAS
> method in both noiseless and noisy settings in Appendix K of the revised manuscript and consequently modified the description in  Section 4 (Experiments).\
> \
> With kind regards, \
> the Authors

---

### Comment · Area_Chair_F9eG · 2023-11-23

Dear Reviewers K92h, sjih, wt33, 3EHX,

As the discussion period is coming to an end, please acknowledge that you have read the rebuttals written by the authors. Thanks.

Best wishes,
AC

---

### Meta-Review · Area_Chair_F9eG · 2023-12-06

**Metareview:**

This paper studies ansatz design in variational quantum algorithms (VQAs), a counterpart of classical neural networks for quantum computing that is implementable on near-term noisy intermediate-scale quantum devices. The main contribution is a curriculum-based reinforcement learning QAS (CRLQAS) algorithm optimized to tackle the challenges of realistic VQA deployment by introducing (i) a 3-D architecture encoding and restrictions on environment dynamics to explore the search space of possible circuits efficiently, (ii) an episode halting scheme to steer the agent to find shorter circuits, and (iii) a novel variant of simultaneous perturbation stochastic approximation algorithm as an optimizer for faster convergence. Numerical experiments focusing on quantum chemistry tasks demonstrate that CRLQAS outperforms existing QAS algorithms across noiseless and noisy environments.

The initial version of the paper has the following strengths:
- The idea of developing features that enhance the learning process and reduces the search space and training episodes in the CRLQAS method has novelty.
- The CRLQAS algorithm has efficiency in simulating noise profiles akin to real quantum devices.
- For the experiments on quantum chemistry, the results achieve notable chemical accuracy and superiority over previous methods in both noisy and noiseless settings.

It also has the following disadvantages, as pointed out by the reviewers:
- Absence of Ablation study of proposed features of CRLQAS.
- Absence of comparison to other relevant works.
- Absence of experiments on real quantum device.
- Absence of codes.
- Writing and presentation has notable space to improve.

During the rebuttal period, the authors have devoted significant efforts to improve upon these perspectives. In particular, the writing (including typos) and the presentation style has been polished more, the source codes are (anonymously) released, comparison to existing references is added, and more experiments are conducted for ablation study and simulation of real machine outcomes.

Although the reviewers are not very involved into the rebuttal period and subsequent discussions, the AC carefully checked the discussions, in particular the posts by the authors. From the perspective of the AC, the quality of the paper has been significantly improved, and the story that reinforcement learning can be applied to searching ansatzes of variational quantum algorithms with improvement is overall compelling. On the one hand, the technique is a demonstration of AI for Science as it applies RL to solving a quantum problem, and on the other hand, ansatz design is also an important problem in quantum computing while it shares similarity to neural network design in machine learning. These two perspectives should be beneficial for the general audiences for ICLR 2024, and hence the AC decides to vote for acceptance.

For the final version, the authors should carefully check if all points discussed in the rebuttals are merged into the paper. Many points from the rebuttal are valuable and can provide a more complete story for this work.

**Justification For Why Not Higher Score:**

The scores of this paper are on the borderline, and it's probably not competitive enough for a spotlight or an oral.

**Justification For Why Not Lower Score:**

As explained above, the story that reinforcement learning can be applied to searching ansatzes of variational quantum algorithms with improvement is overall compelling. On the one hand, the technique is a demonstration of AI for Science as it applies RL to solving a quantum problem, and on the other hand, ansatz design is also an important problem in quantum computing while it shares similarity to neural network design in machine learning. These two perspectives should be beneficial for the general audiences for ICLR 2024, and hence the AC decides to vote for acceptance.

---

### Decision · Program_Chairs · 2024-01-16

Accept (poster)